# Ramping dynamics and theta oscillations reflect dissociable signatures during rule-guided human behavior

Jan Weber [1,2], Anne-Kristin Solbakk [3,4,5,6], Alejandro O. Blenkmann [3,4], Anais Llorens [3,4,7], Ingrid Funderud [3,4,6], Sabine Leske [3,4,8], Pål Gunnar Larsson [5], Jugoslav Ivanovic [5], Robert T. Knight [7,9], Tor Endestad [3,4,10] & Randolph F. Helfrich [1,10] ✉

Contextual cues and prior evidence guide human goal-directed behavior. The neurophysiological mechanisms that implement contextual priors to guide subsequent actions in the human brain remain unclear. Using intracranial electroencephalography (iEEG), we demonstrate that increasing uncertainty introduces a shift from a purely oscillatory to a mixed processing regime with an additional ramping component. Oscillatory and ramping dynamics reflect dissociable signatures, which likely differentially contribute to the encoding and transfer of different cognitive variables in a cue-guided motor task. The results support the idea that prefrontal activity encodes rules and ensuing actions in distinct coding subspaces, while theta oscillations synchronize the prefrontal-motor network, possibly to guide action execution. Collectively, our results reveal how two key features of large-scale neural population activity, namely continuous ramping dynamics and oscillatory synchrony, jointly support rule-guided human behavior.

Human decisions depend on available prior evidence and contextual cues. A long-standing question in models of top-down guided behavior is how prior evidence is incorporated to guide subsequent action[1–3]. The active sensing framework postulates that the brain utilizes its inherent rhythmic structure as an energy-efficient mechanism to implement temporal predictions[4,5]. This framework further predicts that the brain switches from a rhythmic to a continuous energy-costly processing mode when less prior evidence is available. Active sensing also implies that synchronization of endogenous oscillations is instrumental for inter-areal information transfer, as suggested by the communication-through-coherence hypothesis[6]. Active sensing has

mainly been studied in the context of sensory selection[7–9] and to date it remains unknown whether similar principles apply when context is signaled by abstract cues. Recent work in non-human primates (NHP) has demonstrated that sensorimotor cortex as well as adjacent premotor areas, such as the frontal eye fields, encode high-level contextual information in neural population codes[10–15]. Whereas the active sensing framework relies on univariate features (i.e., oscillatory power, phase, and neural firing), the population doctrine emphasizes that information is encoded in the entire population response that can be conceptualized as a trajectory passing through a high-dimensional neural state space[16]. To date, population coding and neural oscillations, two key signatures of

[1]Hertie Institute for Clinical Brain Research, Center for Neurology, University Medical Center Tübingen, Tübingen, Germany. [2]International Max Planck Research School for the Mechanisms of Mental Function and Dysfunction, University of Tübingen, Tübingen, Germany. [3]Department of Psychology, University of Oslo, Oslo, Norway. [4]RITMO Centre for Interdisciplinary Studies in Rhythm, Time and Motion, University of Oslo, Oslo, Norway. [5]Department of Neurosurgery, Oslo University Hospital, Oslo, Norway. [6]Department of Neuropsychology, Helgeland Hospital, Mosjøen, Norway. [7]Helen Wills Neuroscience Institute, UC Berkeley, Berkeley, CA, USA. [8]Department of Musicology, University of Oslo, Oslo, Norway. [9]Department of Psychology, UC Berkeley, Berkeley, CA, USA. [10]These authors jointly supervised this work: Tor Endestad, Randolph F. Helfrich. ✉e-mail: randolph.helfrich@gmail.com

coordinated population activity, have mainly been studied in isolation. Consequently, it remains elusive how both features interact to guide goal-directed behavior.

In this study, we addressed how high-level contextual information is flexibly integrated into ensuing actions in humans. We specifically tested if principles of the active sensing framework also apply to prefrontal-motor interactions when contextual information is rule-based and not sensory-driven[9]. Furthermore, we aimed to determine the population correlates of rhythmic and continuous processing modes. Population activity has mainly been studied using single- and multi-unit recordings in NHPs. Here we recorded intracranial electroencephalography (iEEG) from prefrontal and motor cortex in patients with epilepsy who underwent invasive monitoring for localization of the seizure onset zone. We specifically studied high-frequency band activity (HFA; 70–150 Hz) as a proxy of population firing to address if coding principles that have previously been identified in NHP also apply in the human brain. All participants engaged in a cue-guided motor task, where they were instructed to continuously track a moving target and release a button once it reached a predefined spatial location. A contextual cue determined the probability of a premature and abrupt stop when participants had to withhold their ongoing response. Here, we defined context as the currently active rule, which exhibited predictive information about a subsequent action.

We describe a functional dissociation between population activity and network oscillations where human PFC encodes the current context (active rule) and the current action plan in distinct subspaces using a continuous processing regime, while theta oscillations mediate the inter-areal communication between PFC and motor cortex to mediate context-dependent actions. Collectively, we identified computationally distinct roles of continuous and rhythmic brain activity at the population level that jointly support context-dependent, goal-directed human behavior.

## Results

We recorded intracranial EEG (iEEG) from 19 pharmaco-resistant patients with epilepsy (33.73 years ±12.52, mean ± SD; 7 females) who performed a predictive motor task (Fig. 1a). Participants had to closely track a moving target and respond (go trial) as soon as the target reached a predefined spatial location (hit lower limit; HLL). They were instructed to withhold their response if the target stopped prematurely (stop trial). A predictive cue defined the context for the current trial by signaling the likelihood of a stop trial (green circle = 0%, orange circle = 25%, red circle = 75%). We refer to the stop likelihood as behavioral uncertainty or predictive context and use these terms interchangeably. We simultaneously recorded from prefrontal cortex (PFC) and motor cortex to study how the human prefrontal-motor network converts predictive context into concrete actions (Fig. 1b).

### Neural and behavioral signatures of context-dependent computations

We confirmed that participants used the predictive cue to guide behavior using reaction time, accuracy and signal detection theory.

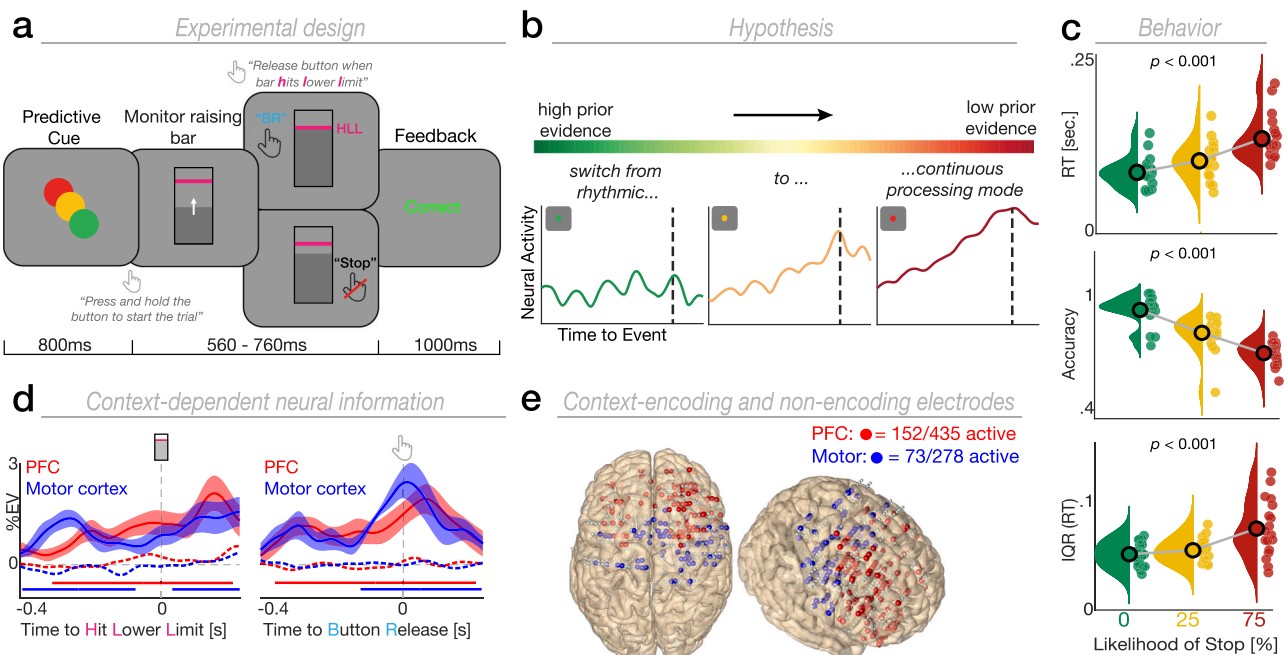

**Fig. 1 | Task design, hypothesis, behavioral results and electrophysiological signatures of context-dependent neural information. a** Participants were presented with a predictive cue indicating the likelihood that a moving target (self-initiated via space bar press) would stop prior to a predefined lower limit (HLL; pink horizontal line). Participants were asked to release the space bar as soon as the target hit the lower limit (go trial) or withhold the response if the target stopped before reaching the lower limit (stop trial). Afterwards, participants received feedback. **b** Schematic illustration of our key hypothesis. States of high behavioral uncertainty should introduce a switch towards stronger ramping dynamics. **c** Behavioral results. Scattered dots represent single grand averages, black outlined dots depict the group level average and histograms illustrate the probability distribution. Upper: RTs gradually scales with behavioral uncertainty ($F(2,36) = 58.99$, $p < 0.001$, $n = 19$; one-way RM-ANOVA). Middle: Accuracy gradually decreases as a function of behavioral uncertainty ($F(2,36) = 81.53$, $p < 0.001$, $n = 19$; one-way RM-ANOVA). Lower: Interquartile range also increases from trials with no to high behavioral uncertainty ($F(2,36) = 11.36$, $p < 0.001$, $n = 19$; one-way RM-ANOVA). **d** ROI-specific time course of context-dependent neural information (percent explained variance, %EV) for context-encoding (solid lines) and non-encoding electrodes (dashed lines) aligned to the HLL (left; PFC: $n = 16$; motor cortex: $n = 11$) or the button release (right; PFC: $n = 17$; motor cortex: $n = 11$). The lower horizontal lines show the temporal extent of significant cluster differences between context-encoding and non-encoding electrodes for the respective ROI (cluster test; two-tailed). Shading represents the standard error of the mean (SEM) across participants. **e** Context-encoding (red spheres = PFC; blue spheres = motor cortex) and non-encoding (white spheres) electrodes overlaid on a standardized brain in MNI space for our two regions of interest. Source data are provided as Source Data file.

Here, reaction time was quantified as the time interval between the moving target reaching the predefined lower limit and the participants' response. Accuracy was defined as the percentage of correct responses relative to the number of trials. Trials in which participants released the button within the time interval between the lower and upper limit (Fig. 1a) were considered as correct trials whereas trials in which they released the button either before the lower limit or after the upper limit were considered as incorrect. We found that reaction times (RT) gradually increased as a function of uncertainty (Fig. 1c; 0% = 79.92 ± 19.66 ms; 25% = 94.98 ± 23.86 ms; 75% = 123.81 ± 29.34 ms; mean ± SD; $F_{(2,36)} = 58.99$, $p < 0.001$, $\eta_\rho^2 = 0.77$; one-way RM-ANOVA; see also Supplementary Fig. 1). Participants were also less accurate in trials with high uncertainty (Fig. 1c; 0% = 92.08 ± 6.99%; 25% = 81.1 ± 8.33%; 75% = 71.39 ± 5.46%; mean ± SD; $F_{(2,36)} = 81.53$, $p < 0.001$, $\eta_\rho^2 = 0.82$). We then quantified how predictive context modulated participants' sensitivity $d'$ (d-prime) and decision criterion $c$ (Methods).

We observed that sensitivity $d'$ decreased (Supplementary Fig. 2; $p = 0.002$, Cohen's $d = 1.03$; Wilcoxon signed-rank test; two-tailed) and criterion $c$ increased (Supplementary Fig. 2; $p = 0.008$, Cohen's $d = -0.7$) with uncertainty, indicating a more conservative response strategy as stop trials became more likely. Furthermore, linear ballistic accumulator modeling (Methods) revealed that prior evidence caused a shift in the starting point (Supplementary Fig. 2; $p < 0.001$; Friedman test). In contrast, no statistically significant difference was observed for the drift rate of the decision process ($p = 0.229$). To quantify trial-by-trial variability, we assessed the interquartile range (IQR) as a measure of dispersion (Fig. 1c). We found that RTs were more consistent for predictive trials (IQR 0.05 s ± 0.01 s; mean ± SD) and more variable under high uncertainty (IQR 0.08 s ± 0.03 s; mean ± SD; $F_{(2,36)} = 11.36$, $p < 0.001$, $\eta_\rho^2 = 0.39$; one-way RM-ANOVA). In sum, these results demonstrate that states of high behavioral uncertainty are detrimental for the speed, accuracy, and sensitivity of action-linked decisions. Furthermore, they demonstrate that participants altered their response strategy, thereby providing evidence that they used the predictive cue to guide their decisions.

We assessed the neural dynamics using HFA as a proxy for local population activity[17–19]. The initial quantification of percent variance[20–24] explained by context revealed significant context-dependent neural information in both PFC and motor cortex when time-locked to the HLL (Fig. 1d; PFC: $t_{(15)} = 985.91$, $p < 0.001$, Cohen's $d = 0.83$; motor cortex: first cluster, $t_{(10)} = 761.78$, $p < 0.001$, Cohen's $d = 1.24$; second cluster, $t_{(10)} = 351.6$, $p < 0.001$, Cohen's $d = 0.96$; cluster test). A comparable pattern was observed when time-locked to action execution (button release, BR; PFC: $t_{(16)} = 1144.2$, $p < 0.001$, Cohen's $d = 0.88$; motor cortex: $t_{(10)} = 941.25$, $p = 0.002$, Cohen's $d = 1.51$). Neural information evolved similarly in both regions over time (no statistically significant differences were observed; all $p > 0.09$; cluster test). To ensure that the difference in the ratio between correct and incorrect trials did not confound our analysis, we orthogonalized the factors context and accuracy using an unbalanced ANOVA (Supplementary Fig. 3). This confirmed the existence of context-dependent neural information, precluding spurious effects as driven by the ratio of correct/incorrect trials. Note that percent variance explained is an unsigned estimate of neural information and does not indicate the direction of the association (positive or negative). Hence, no inference on the sign of context-dependent effects can be drawn based on this analysis. While this approach allows for the extraction of context-encoding electrodes, it does neither impose any bias nor provide any information with respect to the direction of the effect. Thus, any randomly distributed effect across time and/or conditions would result in an inconsistent context-dependent modulation in the grand average HFA traces (Supplementary Fig. 4; Methods).

Overall, we found that 35% ($N = 152$) of all electrodes in PFC and 27% ($N = 73$) of all electrodes in motor cortex significantly encoded context (Fig. 1e; Methods). We used context-encoding electrodes for subsequent univariate analyses unless stated otherwise. We found a context-dependent HFA modulation in both PFC (Fig. 2a; first cluster: $F_{(2,30)} = 699.15$, $p = 0.009$; second cluster: $F_{(2,30)} = 496.57$, $p = 0.018$; cluster test) and motor cortex (Fig. 2b; $F_{(2,20)} = 326.6$, $p = 0.036$). The strongest context-dependent modulation was observed for the PFC ~ 300 ms prior to the HLL (Fig. 2a).

We next assessed HFA strength (peak amplitude) and peak timing to quantify neural dynamics on a trial-by-trial basis. We observed that HFA in PFC gradually scaled with behavioral uncertainty (Fig. 2a; $F_{(2,30)} = 9.77$, $p < 0.001$, $\eta_\rho^2 = 0.39$; one-way RM-ANOVA). We did not observe a statistically significant modulation as a function of uncertainty in motor cortex (Fig. 2b; $F_{(2,18)} = 2.37$, $p = 0.122$, $\eta_\rho^2 = 0.21$). A significant context x ROI interaction confirmed the local specificity of this effect ($F_{(2,18)} = 4.67$, $p = 0.046$, $\eta_\rho^2 = 0.34$; two-way RM-ANOVA). Furthermore, PFC population activity peaked later in trials with high as compared to no uncertainty (Fig. 2a; $F_{(2,30)} = 9.07$, $p < 0.001$, $\eta_\rho^2 = 0.38$; one-way RM-ANOVA). We found no evidence for a context-dependent temporal dissociation in motor cortex ($F_{(2,20)} = 1.97$, $p = 0.165$, $\eta_\rho^2 = 0.16$). However, the direction of the effect did not differ significantly between the two regions (context x ROI interaction; $F_{(2,18)} = 1.25$, $p = 0.299$, $\eta_\rho^2 = 0.12$; two-way RM-ANOVA). Collectively, these findings indicate that context-dependent computations are pronounced in PFC. In contrast, contextual information only marginally modulates activity in motor cortex.

Single-trial associations between HFA amplitude, timing, and behavior were investigated using linear regression. This analysis revealed that HFA in PFC and motor cortex predicted RT on a trial-by-trial basis (Fig. 2e, f; PFC; $F_{(2,2903)} = 40.09$, $R^2 = 0.026$, $p < 0.001$; motor cortex; $F_{(2,2017)} = 34.93$, $R^2 = 0.032$, $p < 0.001$; linear regression; see Supplementary Table 1 for partial linear regression). In summary, larger HFA peak amplitudes and slower peak latencies predicted slower RTs. These results highlight delayed and increased HFA responses in states of low predictive context that can be directly mapped to behavior on single trials.

## Ramping dynamics, but not oscillatory signatures dissociate states of uncertainty

We directly tested whether different processing modes implement predictive context (Fig. 1b) by disentangling oscillatory and ramping dynamics. We computed the HFA slope on single trials (Fig. 2c, d). In line with our main predictions, we found that ramping dynamics were modulated by predictive context in PFC (Fig. 3a; $F_{(2,30)} = 4.49$, $p = 0.019$, $\eta_\rho^2 = 0.23$; one-way RM-ANOVA). Importantly, we did not find a statistically significant ramping effect in PFC during trials with no uncertainty ($t_{(15)} = -0.2$, $p = 0.419$, Cohen's $d = 0.07$; one-tailed $t$ test vs. zero). However, we found significant ramping in PFC during trials with moderate ($t_{(15)} = 3.34$, $p = 0.002$, Cohen's $d = 1.18$) and high uncertainty ($t_{(14)} = 2.15$, $p = 0.024$, Cohen's $d = 0.79$). There was no statistically significant effect for context-dependent ramping dynamics in motor cortex, (Fig. 3b; $F_{(2,20)} = 0.36$, $p = 0.698$, $\eta_\rho^2 = 0.035$; one-way RM-ANOVA). These results support our prediction that ramping dynamics in PFC are modulated by predictive context.

Prior studies have argued that ramping dynamics reflect the sequential activation of neural populations with recurrent excitation[25,26]. We examined whether ramping dynamics directly index neural excitability using three surrogate markers of population-level neural excitability (low-frequency desynchronization, spectral exponent, and sample entropy[27–30]). While high-frequency synchronization was evident in both PFC (Fig. 3c; $F_{(2,30)} = 849.3$, $p = 0.029$; cluster test) and motor cortex (Fig. 3d; $F_{(2,20)} = 743.3$, $p = 0.007$), low-frequency desynchronization was only apparent in PFC (Fig. 3c; $F_{(2,30)} = 922.1$, $p = 0.024$). To quantify a context x ROI interaction effect, we contrasted the difference between the two extreme context conditions (75% and 0%

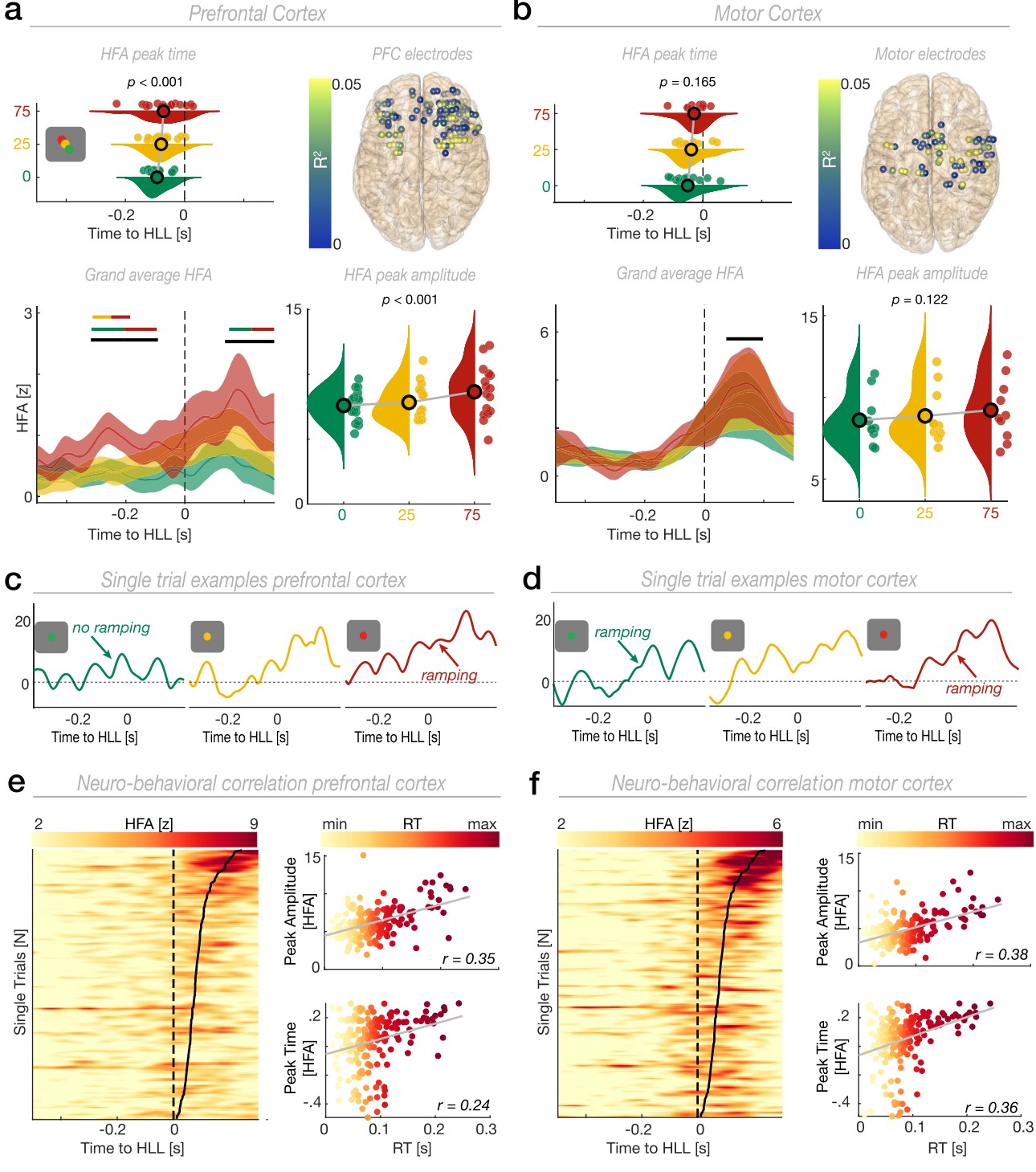

**Fig. 2 | HFA encodes prior evidence and predicts behavior on a trial-by-trial basis. a** Changes in amplitude (lower right) and peak timing (upper left) of the HFA with behavioral uncertainty in PFC (amplitude: $F(2,30) = 9.77$, $p < 0.001$, $n = 16$; timing: $F(2,30) = 9.07$, $p < 0.001$, $n = 16$; one-way RM-ANOVA). Lower left: Grand average HFA time courses per context condition (mean ± SEM). The single-colored horizontal lines show the temporal extent of significant context-dependent processing (cluster test). Two-colored horizontal lines indicate the temporal extent of significant clusters obtained from pairwise comparisons (cluster test; two-tailed). Upper right: Topographical depiction of the neuro-behavioral linear regression. All electrodes are color-coded according to the coefficient of determination ($R^2$). **b** Changes in amplitude (lower right) and peak timing (upper left) of the HFA with behavioral uncertainty in motor cortex (amplitude: $F(2,18) = 2.37$, $p = 0.122$, $n = 10$; timing: $F(2,20) = 1.97$, $p = 0.165$, $n = 11$; one-way RM-ANOVA). Lower left: Grand

average HFA time courses per context condition ($n = 10$; mean ± SEM). Upper right: Topographical depiction of the neuro-behavioral linear regression. Same conventions as in (**a**). **c**, **d** Single HFA trials. **e** Representative single participant example for the neuro-behavioral regression in PFC. Left: Vertically stacked single trials sorted by RT (black line) and color coded according to the z-score. For visualization, panels were smoothed using a 4 trial-wide boxcar function after sorting. Upper right: Relationship between RT and HFA strength ($r = 0.35$, $p < 0.001$, $n = 188$; Spearman's rank correlation; two-tailed). Lower Right: Relationship between RT and HFA peak timing ($r = 0.24$, $p < 0.001$, $n = 188$). **f** Same as (**e**), but for motor cortex. Upper right: Relationship between RT and HFA strength ($r = 0.38$, $p < 0.001$, $n = 188$). Lower Right: Relationship between RT and HFA peak timing ($r = 0.36$, $p < 0.001$, $n = 188$). Same conventions as in (**e**). Source data are provided as Source Data file.

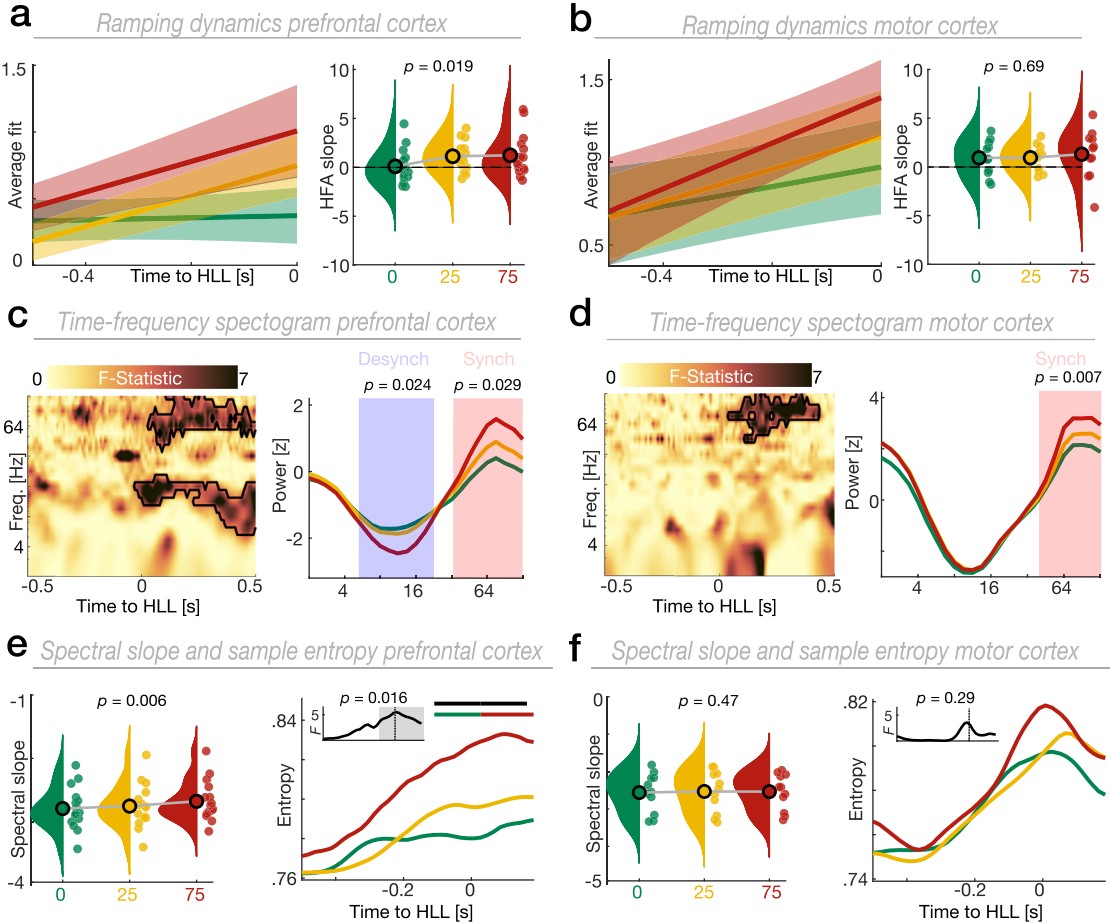

**Fig. 3 | Ramping dynamics dissociate states of behavioral uncertainty and reflect neural excitability. a** Left: Grand average linear fit (mean ± SEM) obtained by fitting a linear regression to HFA single trials in PFC. Right: Group-level results depicting the context-dependent modulation of ramping dynamics in PFC ($F(2,30) = 4.49$, $p = 0.019$, $n = 16$; one-way RM-ANOVA). **b** Same as (**a**), but for motor cortex ($F(2,20) = 0.36$, $p = 0.698$, $n = 11$). Same conventions as in (**a**). **c** Time-frequency dynamics in PFC were modulated by predictive context. The black outline indicates the extent of the significant cluster across time and frequency (left panel). Higher frequencies synchronized whereas lower frequencies desynchronized as a function of behavioral uncertainty (cluster test; $n = 16$; right panel). Traces were smoothed for visualization purposes using a 5 Hz running average.

**d** Same as (**c**), but for motor cortex ($n = 11$). Same conventions as in (**c**). **e** Left: The aperiodic spectral slope in PFC flattened with increasing uncertainty ($F(2,30) = 5.97$, $p = 0.006$, $n = 16$; one-way RM-ANOVA). Right: Time-resolved PFC sample entropy ($F(2,30) = 68.32$, $p = 0.016$, $n = 16$; cluster test). The single-colored horizontal lines show the temporal extent of significant main effects. Two-colored horizontal lines indicate the temporal extent of significant clusters obtained from pairwise comparisons (cluster test; two-tailed). The small inset depicts the temporal evolution of context-dependent sample entropy. **f** Same as (**e**), but for motor cortex (left: spectral slope; $F(2,20) = 0.77$, $p = 0.476$, $n = 11$; one-way RM-ANOVA; right: sample entropy; $F(2,20) = 4.01$, $p = 0.291$, $n = 11$; cluster test). Same conventions as in (**e**). Source data are provided as Source Data file.

likelihood of stop) obtained per ROI using cluster-corrected paired t-tests (two-tailed; Methods). This analysis confirmed that low-frequency desynchronization during states of high uncertainty was specific to PFC (2–19 Hz; $t(10) = −548.72$, $p = 0.006$). The spectral slope has been shown to closely track the excitability in neural circuits flatter slopes indicate more excitability[29,31,32]. We found that the spectral slope flattened with behavioral uncertainty in PFC (Fig. 3e; $F(2,30) = 5.97$, $p = 0.006$, $\eta_\rho^2 = 0.28$; one-way RM-ANOVA). We did not observe a statistically significant difference in motor cortex (Fig. 3f; $F(2,20) = 0.77$, $p = 0.476$, $\eta_\rho^2 = 0.07$; context x ROI interaction effect; $F(2,20) = 14.2$, $p < 0.001$, $\eta_\rho^2 = 0.59$; two-way RM-ANOVA). We computed time-resolved excitability fluctuations in using sample entropy[30]. Time-resolved sample entropy was context-dependent in PFC and showed the strongest increase in trials with high uncertainty (Fig. 3e; $F(2,30) = 68.32$, $p = 0.016$; cluster-test). In contrast, we found no statistically significant difference for context-dependent time-resolved entropy in motor cortex (Fig. 3f; $F(2,20) = 4.01$, $p = 0.291$). No statistically significant context x ROI interaction was observed (no cluster at $p < 0.05$; two-tailed, paired $t$ test). Collectively, this set of findings demonstrates that

predictive context initiates a shift in ramping dynamics and neural excitability. These shifts are most pronounced in PFC (Fig. 3a, c, e) in comparison to motor cortex (Fig. 3b, d, g) and gradually scale with behavioral uncertainty.

Next, we investigated how oscillatory dynamics were modulated by predictive context. We extracted all HFA peaks (Fig. 4a) and performed peak-triggered averaging (PTA; Fig. 4b). We found that HFA is nested in a theta oscillation (~5 Hz; unconstrained sine fit; Fig. 4b). In order to quantify this on a group level and assess context-dependent modulations, we spectrally decomposed the PTA and separated oscillatory from aperiodic background activity by means of irregular-resampling auto-spectral analysis (IRASA)[33]. A cluster-based permutation test revealed reduced oscillatory power in PFC during trials with high uncertainty (Fig. 4c; first cluster; $F(2,30) = 58.16$, $p < 0.001$; second cluster; $F(2,30) = 56.54$, $p < 0.001$; cluster test). Importantly, this context-dependent modulation was not driven by changes in the peak frequency of the theta oscillations (Fig. 4c). Pronounced theta peaks were present irrespective of the contextual cue. We also determined the instantaneous peak frequency directly on the HFA signal by computing the interval between adjacent HFA peaks (Fig. 4d)[34]. The

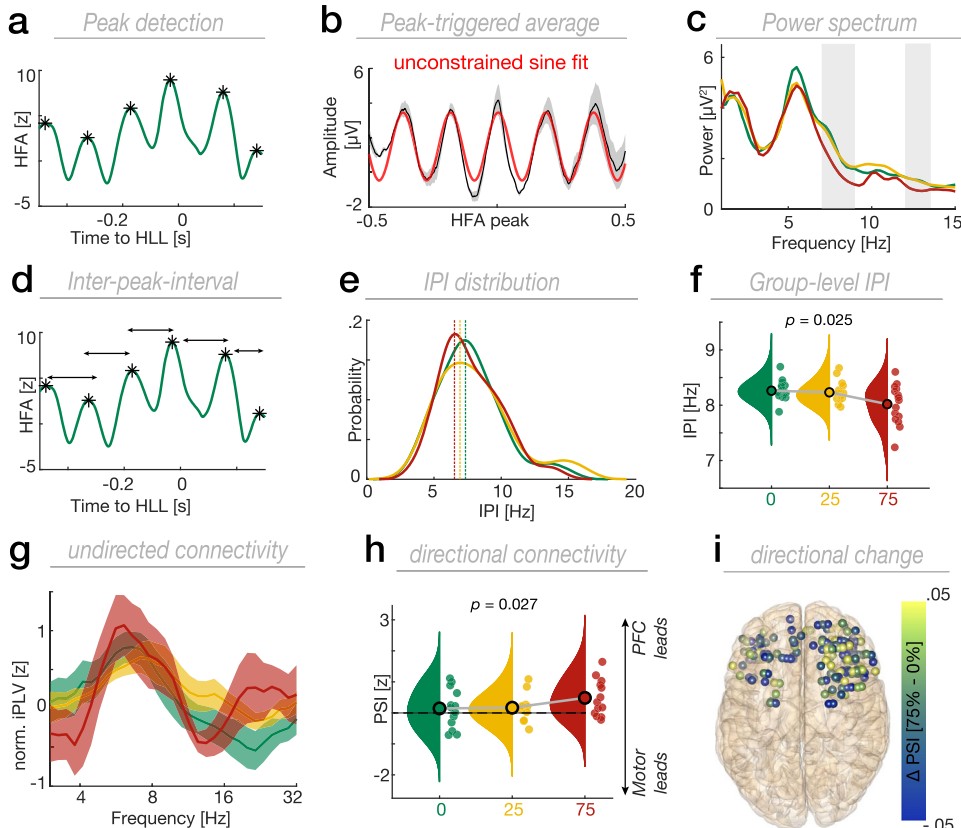

**Fig. 4 | Theta oscillations modulate HFA and mediate context-dependent information flow from PFC to motor cortex. a** Example of the peak detection on single trial HFA traces (black asterisk). Note the waxing and waning pattern in single trials. **b** Peak-triggered average (PTA; mean ± SEM; ±0.5 s from HFA peak) in a representative single participant across PFC electrodes. HFA was nested into a ~ 5 Hz theta oscillation (red line depicts a sine fit to the PTA). **c** Grand average 1/f-corrected power spectrum computed on the PTA time-series using IRASA. Shaded gray areas depict the extent of significant context-dependent power modulation (n = 16; cluster test). Pronounced theta peaks were present in all predictive context conditions. **d** Example trace depicting the quantification of the inter-peak interval (IPI) as a time length between two contiguous peaks. **e** Single electrode example showing the IPI distribution across conditions. Vertical dashed lines represent the peaks of the distributions. **f** Reduced IPI with increasing behavioral uncertainty (F(2,30) = 4.14, p = 0.025, n = 16; one-way RM-ANOVA). **g** Grand-average (mean ± SEM) prefrontal-motor undirected connectivity. Undirected connectivity was not modulated by states of uncertainty, but showed pronounced peak connectivity in the theta band. **h** Directional prefrontal-motor connectivity in the theta band. Directional information flow from PFC to motor cortex was enhanced during states of high uncertainty (F(2,24) = 4.2, p = 0.027, n = 13; one-way RM-ANOVA). **i** Topographical depiction of the directional change in information flow from PFC to motor cortex between distinct states of uncertainty. Source data are provided as Source Data file.

instantaneous HFA peak frequency decreased with uncertainty (Fig. 4e, f; F(2,30) = 4.14, p = 0.025, $\eta_\rho^2$ = 0.22; one-way RM-ANOVA). While theta oscillatory peaks were equally present in motor cortex, we did not observe a statistically significant context-dependent modulation in either the oscillatory power of the PTA (Supplementary Fig. 5; F(2,20) = 7.61, p = 0.368; cluster test) or the instantaneous frequency (Supplementary Fig. 5; F(2,20) = 2.24, p = 0.132, $\eta_\rho^2$ = 0.18; one-way RM-ANOVA) of the HFA signal. Taken together, we did not find strong evidence for a context-dependent modulation of oscillatory power. In contrast to the presumed switch from an oscillatory to a continuous processing regime (Fig. 1b), we found that neural oscillations are ubiquitous across all predictive contexts.

This set of findings raised the question which role neural oscillations play in processes where evidence needs to be converted into an action. Based on the well-established role of neural oscillations in mediating inter-areal communication[6,35], we tested whether oscillations synchronize the prefrontal-motor network. We computed the imaginary phase-locking value (iPLV) between prefrontal-motor electrode pairs to assess network connectivity. We observed strong prefrontal-motor synchrony in the theta band (Fig. 4g; 6.4 ± 1.3 Hz; mean ± SD), but we did not observe a statistically significant difference between context conditions (F(2,26) = 3.82, p = 0.39; cluster test). To assess directional interactions, we computed the phase-slope index

(PSI)[36]. We first identified the individual iPLV peak frequency for every prefrontal-motor electrode pair prior to computation of the PSI. We found that directional theta connectivity from PFC to motor cortex was context-dependent (Fig. 4h, i; F(2,24) = 4.2, p = 0.027, $\eta_\rho^2$ = 0.26; one-way RM-ANOVA) and strongest in trials with high behavioral uncertainty (t(12) = 2.96, p = 0.012, Cohen's d = 1.16; two-tailed t-test vs. zero). Collectively, this set of findings demonstrates that ramping dynamics in PFC dissociate states of behavioral uncertainty while neural oscillations might dynamically coordinate the prefrontal-motor network interaction in a context-dependent manner.

Having established that oscillatory and ramping dynamics reflect dissociable signatures of large-scale population activity, we next tested whether ramping dynamics reflect coordinated population activity using a multivariate state-space approach (Supplementary Note and Supplementary Figs. 6–12). Collectively, this set of analyses indicated that behaviorally-relevant information about the current rule and ensuing action are encoded in distinct, low-dimensional subspaces. Hence, we finally characterized how state-space dynamics interact with neural oscillations to support rule-guided behavior. We specifically focused on theta oscillations as a potential candidate mechanism for the temporal synchronization and generalization of action plans. This step was chosen in a data- and theory-driven way. We did not observe a statistically significant modulation of theta oscillations as a function of

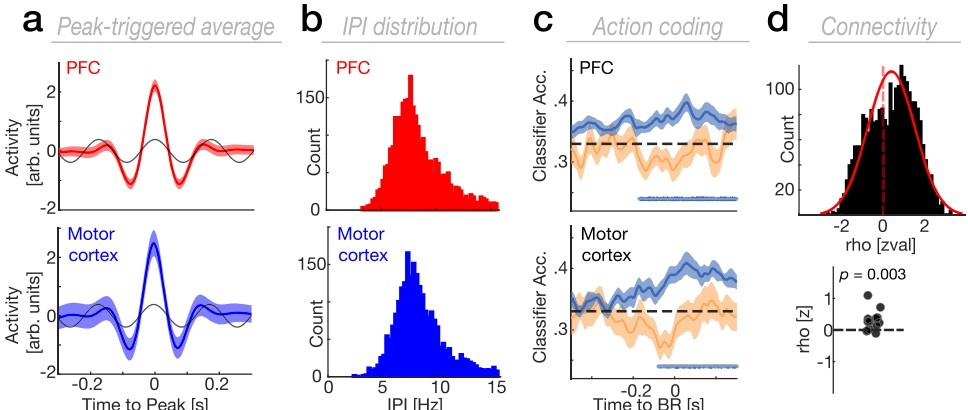

**Fig. 5 | Theta oscillations temporally coordinate action-encoding subspaces in the prefrontal-motor network. a** Peak-triggered average (PTA; mean ± SEM) across participants obtained from the PC with the strongest theta power in PFC (upper panel; *n* = 17) and motor cortex (lower panel; *n* = 14). The black lines depict a sine fit to the PTAs. **b** The instantaneous frequency of the identified theta PC as computed by the inter-peak-interval (upper panel shows the distribution for PFC, the lower panel for motor cortex). Both PFC and motor cortex showed strong oscillatory peaks in the theta-frequency band. **c** Grand average decoding accuracy (mean ± SEM) for context and action within the identified theta PC in PFC (upper panel) and motor cortex (lower panel). The colored horizontal line shows the temporal extent of significant action-decoding (cluster test; two-tailed). **d** Upper panel: Histogram depicting z-normalized power correlation coefficients between the theta PCs in PFC and motor cortex. Lower panel: Significant coupling between the theta PCs in PFC and motor cortex (*p* = 0.003; Wilcoxon signed-rank test; two-tailed). Single dots represent the z-normalized (permutation) power correlation coefficients (*n* = 14). Source data are provided as Source Data file.

context in either PFC or motor cortex. Instead, the results indicated that neural activity was rhythmically structured in both regions (cf. Fig. 4c; Supplementary Fig. 5). Hence, we reasoned that theta oscillations might synchronize the prefrontal-motor network for inter-areal communication (cf. Fig. 4g). These data-driven findings were further supported by the communication-through-coherence framework[6] and a seminal study on theta-oscillatory, prefrontal-motor interactions during cognitive control[22]. Consequently, we aimed to test whether theta oscillations temporally synchronize low-dimensional subspaces between PFC and motor cortex for the transfer of action plans from PFC to motor cortex.

To accomplish this, we first extracted the dimension with the strongest oscillatory theta power in every participant (Fig. 5a, b). Next, we employed LDA classifiers to assess the coding features of the theta component. In both PFC and motor cortex, we found that the dimension with the strongest theta power significantly encoded action (Fig. 5c; PFC; $t(13) = 740.72$, $p < 0.001$, Cohen's $d = 0.82$; motor cortex; $t(9) = 816.03$, $p = 0.002$, Cohen's $d = 1.32$; cluster test), but we did not observe a statistically significant modulation for context (PFC and motor cortex; no cluster at $p < 0.05$). We also observed no statistically significant difference for the consideration that the theta dimension and the previously determined action dimensions were embedded in distinct subspaces ($p = 0.18$; Binomial test). Finally, we observed that neural dynamics embedded in the dimensions with strongest theta power are functionally coupled within the prefrontal-motor network (Fig. 5d; $p = 0.003$, Cohen's $d = 0.69$; Wilcoxon signed-rank test), indicating a possible functional role of theta oscillations to mediate the cross-regional generalization of action plans from prefrontal to motor cortex. Coupling strength was not statistically significantly different between predictive contexts ($F(2,26) = 1.36$, $p = 0.273$, $\eta_\rho^2 = 0.09$; one-way RM-ANOVA).

Taken together, these findings are in accordance with the idea that structured population activity in PFC encodes and integrates predictive information into ensuing action plans that are executed in motor cortex. Our results imply that the transformation from PFC-dependent context integration to goal-directed action execution in motor cortex is mediated by directed theta-band connectivity (cf. Fig. 4g–i).

## Discussion

Rule-guided decision-making is a hallmark of flexible human behavior. To date, it remains unknown how rules or contextual priors are encoded to guide decision processes in humans. Previous work in NHP indicated that adjacent premotor structures, such as frontal eye fields[10,12] or dorsal premotor cortex[14,37], might mediate context-dependent decision-making. While earlier theories, such as the active sensing framework[4,5], emphasized that neural coding is mainly reflected in local activity profiles (i.e., neural firing or oscillatory desynchronization), novel population-based theories now suggest that context-dependent processing is distributed across large-scale neural populations[10,12].

Thus far, the population doctrine had its greatest impact on understanding movement-related computations in the non-human primate motor system[38–41]. We posited it might also provide a powerful framework to understand higher cognitive processes in humans[42]. Using a predictive motor task, we demonstrate that (I) behavioral uncertainty is reflected in neural indices of uncertainty as quantified by uni- (Figs. 2, 3) and multivariate analyses (Supplementary Fig. 6). In line with the active sensing framework, we show that (II) behavioral uncertainty introduces a shift from a presumably energy-efficient oscillatory to a likely more energy-costly processing mode with mixed oscillatory and ramping dynamics (Figs. 3, 4). Using population-based analysis strategies, our results demonstrate that (III) oscillatory and ramping dynamics reflect dissociable population signatures that likely contribute to distinct aspects of encoding and transfer of context-dependent action plans (Supplementary Fig. 6). Specifically, our results support the view that (IV) prefrontal population activity encodes predictive information and ensuing action plans in distinct and serially unfolding subspaces, while motor cortex is primarily involved in action execution (Supplementary Fig. 6). Furthermore, our results indicate that (V) theta synchrony might temporally coordinate action-encoding population subspaces, thereby mediating the cross-regional generalization of action plans (Fig. 5). Collectively, our results provide evidence for the idea that two hallmarks of large-scale population activity, namely continuous ramping dynamics and oscillatory synchrony, are dissociable and possibly fulfill distinct operations to guide context-dependent human behavior.

### Oscillatory and ramping dynamics reflect distinct population signatures of context-dependent behavior

The influential active sensing framework postulates that the brain switches from an energy-efficient oscillatory processing mode during states of high predictability to an energy-consuming continuous

ramping processing mode in states of low predictability. Evidence for this theory has mainly been obtained in NHP auditory cortex[7,8], but it had been argued that similar principles apply to higher-order cortical areas[5]. In line with this framework, we found that the transition from high to low prior evidence increased ramping dynamics in the human PFC, but not in motor cortex. Contrary to the theory, we did not find evidence for a modulation of local oscillatory dynamics as a function of predictability. In addition, a related line of inquiry argued that frontal theta activity constitutes a mechanism of cognitive control, especially in states of high uncertainty[22,43]. Using direct brain recordings in humans, we found that directional theta synchrony is inversely related to predictability. We found stronger directional theta synchrony from prefrontal to motor cortex in states of high uncertainty, indicating a flexible recruitment and network engagement when limited predictive information is available. Using population-based decoding, we found that theta oscillations were not associated with the encoding of predictions per se, but that theta activity was confined to the action subspaces of the population activity. This finding is in line with the communication-trough-coherence hypothesis[6]. This observation could be interpreted as a sequence, where the PFC first encodes the current rule, then devises the appropriate action, which is then executed in the motor cortex. In this scenario, theta synchrony could possibly play an important role for the inter-areal transfer of information about the subsequent action. However, due to the correlative nature of our study we cannot draw any causal inference or directionality nor exclude the possibility that the observed effects could be driven by a third structure that we did not record from. This consideration also possibly explains why we did not observe a continuous representation of the currently active rule in PFC. Future studies that also consider e.g., parietal of medial temporal areas might be able to observe such a sustained response. Furthermore, information about action only culminated after the movement onset. Thus, we cannot preclude that these processes mainly capture post-movement, rather than preparatory dynamics. In our analyses, we observed a build-up of action-specific information (cf. Supplementary Fig. 6g) that emerged prior to the HLL (Supplementary Fig. 9). This observation is compatible with the view that the observed dynamics track the internal transition from planning to the final movement execution, in line with a recent human iEEG-study[44]. However, we cannot completely dismiss the possibility that action-related information preceding the lower limit might merely reflect different mental states, such as discrete phases of movement (planning vs. execution). Therefore, future studies should simultaneously record (sub)cortical activity and electromyography to fully untangle the spatiotemporal gradient between movement planning and movement execution from higher-order association to sensorimotor areas. While we focused on theta oscillations as a possible mechanism for the transfer of motor plans via inter-areal synchronization of action subspaces, other mechanisms related to the temporal control of action have been demonstrated, such as single neuron ramping activity in the lateral intraparietal area[45], medial PFC[46], or in frontal-striatal circuits[47]. In these studies, ramping dynamics mirrored the temporal integration of time (e.g., by representing the hazard rate of reward probabilities[45]). Yet, in our study, ramping dynamics in PFC dissociated between distinct states of uncertainty, while the temporal dynamics in the task were kept constant across trials. This suggests that ramping dynamics in the human PFC possibly encode latent variables in addition to timing[25]. Future studies that are designed to disentangle timing from abstract rule-guided activity are therefore necessary to address the impact of task timing. Moreover, it is conceivable that ramping dynamics could reflect other latent variables, such as engagement, alertness or selective attention. This possibility could ideally be tested using behavioral tasks that are designed to isolate the constructs, possibly combined with other physiologic readouts, such as pupil size, skin conductance, electrocardiogram or eye-tracking to quantify their contribution to ramping dynamics.

Critically, we did not observe a statistically significant context-dependent modulation of ramping activity in motor cortex. Instead, ramping dynamics in motor cortex were largely preserved across all trials, likely reflecting a non-specific and context-independent growing urgency signal[13,48] driven by the necessity of rapid motor decisions in this task. This may be reconciled by the idea that ramping dynamics may subserve specialized functions in different regions. Neural activity in motor cortex exhibited large, context-independent activity changes preceding movement initiation, a pattern that is consistent with prior studies reporting abrupt shifts in activity shortly before movement onset[49]. This context-independent activity, previously also referred to as a condition-invariant signal (CIS)[50], typically reflects the largest response component in motor cortex[50]. The present findings are compatible with a CIS in human motor cortex. The first principal component of the HFA-signal in motor cortex is (1) context-independent and (2) explains the largest variance. It is also possible to consider a scenario where the condition-invariant increase in motor cortex activity prior to reaching the lower limit (cf. Supplementary Fig. 7) foreshadows parallel planning for both movement inhibition and execution[51], the two possible behavioral responses in the task. An unresolved question is whether other areas might drive this sudden change in motor cortex activity. Previous studies have identified a large-scale network that might provide input to motor cortex, including subcortical[52,53] and cortical structures[54,55]. The present findings demonstrate that human PFC also modulates neural activity in motor cortex. Collectively, these results demonstrate that several hallmarks of predictive processing that have primarily been captured using univariate metrics reflect coordinated population-wide activity patterns.

Translating our findings to previous observation in the non-human primate motor system is hampered by the fact that signals from different recording modalities are typically being compared (e.g., single unit vs. EEG activity). Here, we analyzed HFA in humans as a proxy of multiunit activity firing[17–19]. HFA offers the advantage that it already constitutes an aggregate metric that summarizes the underlying population activity. Recent work demonstrated that HFA contains more behaviorally relevant information than single-/multi-unit activity or EEG activity, constituting a suitable level of abstraction to study population-wide activity[56]. Furthermore, theta oscillations rhythmically structure HFA through phase-amplitude cross-frequency coupling. Thus, our results complement previous findings in animal models demonstrating that neural firing is linked to network oscillations[57,58].

### The population doctrine and cognitive processing
The population doctrine is an emerging concept highlighting that population activity, and not the single unit per se, reflects the essential unit of computation in the brain[16,59]. Population activity has mainly been studied in NHP (pre-)motor cortex where distinct movement trajectories are represented by unique neural trajectories of the population[38,41]. While previous evidence in NHP indicated that adjacent premotor structures, such as frontal eye fields[10,12] or dorsal premotor cortex[13,37] may perform context-dependent computations, we found that neural trajectories in prefrontal, but not motor cortex, dissociated the current predictive context. Critically, we observed that large-magnitude neural states within PFC indexed behavioral uncertainty. We found that PFC settled into a low-energy state (smaller magnitude, only covering a limited subspace of the entire state space) during states of high predictability. Critically, these patterns could only be observed using multivariate analysis strategies (Supplementary Fig. 6; cf. Fig. 2 for the univariate approach) that take coordinated variability across different recording sites into account. Previous work in NHP demonstrated that motor cortex exhibits a low-dimensional structure[59,60]. Here, we replicate this finding in humans, but in contrast to prior work in NHPs that has revealed context-dependent

computations in adjacent premotor structures[10,12,13,37], we found no evidence for this consideration in the human motor cortex. However, it is worth noting that we cannot rule out some alternative hypotheses due to inherent limitations of human intracranial recordings and the currently employed experimental design. First, due to the inherent limited coverage in intracranial recordings, we cannot preclude that specialized sub-regions within the human motor cortex (e.g., anterior or posterior parts of the supplementary motor area, frontal eye fields, premotor cortex) might also encode contextual information. Moreover, we did not record from various other brain regions that might maintain context-dependent representation throughout, e.g., the parietal cortex or hippocampus. Second, based on the current experimental design, we cannot fully dissociate between preparatory- and movement-related computations. Furthermore, while we did not employ a characteristic go-cue (i.e., sudden appearance of a sensory go-cue) in our experimental paradigm, the moment at which the bar reaches the lower limit (cf. Fig. 1a) still resembles a go-cue and could potentially trigger condition-invariant activity changes. Consequently, based on the current experimental paradigm, we cannot fully disentangle neural activity that reflects context-dependent processing from neural activity potentially triggered by the lower limit. However, the fact that context-dependent dynamics already evolved (cf. Fig. 2a & Supplementary Fig. 7a) and neural dynamics mainly ramped-up prior to lower limit suggests that the observed neural activity patterns in PFC were not solely triggered by the lower limit, but reflected intrinsic, context-dependent dynamics prior to the lower limit. However, changes in neural activity both before as well as after the lower limit might also be explained by the lower limit itself. Hence, the activity ramp-up in PFC prior to the lower limit might reflect a mixture of coexisting context- and go-cue-dependent activity. Future studies might resolve this limitation by simultaneously recording both brain and muscle activity. However, it is also possible that distinguishing context- and movement-dependent activity in the PFC cannot be fully disentangled given the involvement of the PFC in multiple operations. Future studies that employ experimental designs that are geared towards disentangling context- and go-cues are necessary to separate purely cognitive representation from motor preparation and execution. Finally, prefrontal population activity is high-dimensional in nature, where different operations are encoded in distinct subspaces. Yet, our analytical approach does not allow to draw any inference on the dimensionality of the decoding latent space, only on the overall dimensionality of the neural data. However, our findings support the notion that the high-dimensional prefrontal functional architecture constitutes a substrate for flexible goal-directed behavior and that simultaneous processing in separate coding dimensions maximizes information-coding capacity of the underlying population[11,12,61].

In the present study, our results are compatible with the view that high-dimensional prefrontal population dynamics encode predictive context and action plans in distinct and serially unfolding subspaces. We demonstrate that a lack of prior evidence comes at a behavioral (increased response time/error rates) as well as a neural (large magnitude neural states) cost. Moreover, our results support the interpretation that population trajectories and oscillatory synchrony are dissociable signatures, which support distinct functional roles in the prefrontal and motor cortex. Specifically, our results imply that prefrontal population trajectories encode the current rule, while oscillatory synchrony mediates the transfer of action plans from prefrontal to motor cortex. The view of a division-of-labor between both regions is supported by the observation of low-dimensional neural dynamics in human motor cortex, which did not encode predictive context, but relied on theta-mediated input from higher-order prefrontal areas. In sum, we studied context-dependent motor behavior with univariate as well as multivariate analyses, which collectively shed new light on the role of ramping and oscillatory activity during predictive processing. These findings imply that population dynamics and oscillatory synchrony interact in concert to jointly guide flexible human behavior.

## Methods

### Patients and implantation procedure

We obtained intracranial recordings from a total of 19 pharmaco-resistant epilepsy patients (33.73 years ± 12.52, mean ± SD; 7 females) who underwent presurgical monitoring and were implanted with intracranial depth electrodes (DIXI Medical, France). No statistical methods were used to pre-determine sample size, but the sample size reported in this study is similar or exceeds the sample size reported in comparable previous studies[22,44]. Data from one patient were excluded from neural analyses because a low-pass filter was applied at 50 Hz during data export from the clinical system, thus, precluding analyses focusing on HFA. All patients were recruited from the Department of Neurosurgery, Oslo University Hospital. Electrode implantation site was solely determined by clinical considerations and all patients provided written informed consent to participate in the study. Patients were not compensated for their participation in this study. All procedures were approved by the Regional Committees for Medical and Health Research Ethics, Region North Norway (#2015/175) and the Data Protection Officer at the Oslo University Hospital as well as the University Medical Center Tuebingen (049/2020BO2) and conducted in accordance with the Declaration of Helsinki.

### iEEG data acquisition

Intracranial EEG data were acquired at the Oslo University Hospital at a sampling frequency of 512 Hz using the NicoletOne (Nicolet, Natus Neurology Inc., USA) or at a sampling frequency of 16 KHz using the ATLAS (Neuralynx) recording system.

### CT and MRI data acquisition

We obtained anonymized postoperative CT scans and pre-surgical MRI scans, which were routinely acquired during clinical care.

### Electrode localization

Two independent neurologists visually determined all electrode positions based on individual scans in native space. For further visualization, we reconstructed the electrode positions as outlined recently[62]. In brief, the pre-implant MRI and the post-implant CT were transformed into Talairach space. Then we segmented the MRI using Freesurfer 5.3.0[63] and co-registered the T1 to the CT. 3D electrode coordinates were determined using the Fieldtrip toolbox[64] on the CT scan. Then we warped the aligned electrodes onto a template brain in MNI space for group-level analyses.

### Task

Participants performed a predictive motor task where they had to continuously track a moving target and respond as soon as the target hits or withhold their response if the target stops prior a predefined spatial position using their dominant hand (Fig. 1a). Prior to the main experiment, participants were familiarized with the task by means of a short practice session. Each trial started with a baseline period of 500 ms followed by a cue (presented for 800 ms centered) that informed participants about the likelihood that the moving target would stop prior to the lower limit (hit lower limit; HLL; Fig. 1a). Thus, the predictive cue could be directly translated into the probability that either of two possible action scenarios will occur: button release (BR) vs. withhold response (Bernoulli distribution). Participants were instructed to either release the button as soon as the target hits (Go trials) or withhold their response if the target stops prior to the HLL (Stop trials). The timing of a premature stop was normally distributed prior to the HLL (Supplementary Fig. 13). We parametrically modulated the likelihood of stopping. A green

circle indicated a 0% likelihood, an orange circle indicated a 25% likelihood and a red circle indicated a 75% likelihood that the moving target would stop prior to the HLL. Hence, participants were able to fully predict the outcome on trials with a 0% likelihood and already prepare the motor response. However, in trials with a 25% or 75% likelihood of stopping, they continuously had to accumulate evidence in order to decide whether to release the button or withhold the response. Upon receiving the predictive cue, participants were able to start the trial in a self-paced manner by pressing the space bar on the keyboard (average time to start the trial: 1.76 s ± 0.55 s; mean ± SD). By pressing the space bar, the target would start moving upwards and reach the HLL after 560–580 ms. The upper boundary was reached after 740–760 ms, thus, leaving 160 ms between the HLL and the upper boundary. If participants released the button within this 160 ms interval, the trial was considered as correct. Trials in which the button was released either before or after this interval were considered as incorrect. Feedback on trial performance was provided upon each trial for 1000 ms.

## Behavioral data analysis

We quantified reaction time (RT) as the time passed between the moving target reaching the HLL and the participants' response. We considered both correct and incorrect trials in our analyses on RT. Accuracy was quantified as the average number of correct responses relative to the number of trials. We used the interquartile range (IQR) as a measure for behavioral trial-by-trial variability[65]. We also considered the signal detection theoretic measures $d'$ (d-prime) and $c$ (criterion)[66]. While $d'$ quantifies the distance between the signal (e.g., go trials) and noise distribution (e.g., stop trials), $c$ reflects a participant's propensity to choose yes or no (decision criterion). Due to the nature of the task (absence of noise distribution in the 0% condition), we were only able to quantify $d'$ and $c$ for conditions with a 25% or 75% likelihood of stopping.

**Linear ballistic accumulator model.** For each subject, we fitted a linear ballistic accumulator model[67] for the reaction time using a convolutional method to estimate the non-decision time[68]. We fixed the variance of the drift and the non-decision time at 0.5. We estimated the drift and offset, allowing them to differ between conditions. Model fit was performed in R v. 4.1.3 (R Core Team, 2016), using the DstarM package[69]. It is important to acknowledge that the environment of intracranial EEG recordings precludes long experiments with many control conditions. Based on our design, we cannot fully disentangle behavioral uncertainty from overall task difficulty. We directly addressed this limitation by calculating the SDT as well as linear ballistic accumulator models to quantify the participants' response strategy. However, the observed shift in criterion as a function of uncertainty could also be explained by an overall shift of the signal and noise distribution along the internal response axis that would not involve any change in participants' response strategy. Furthermore, because there were no supplementary eye tracking recordings in the present study, we are unable to comment on the potential variances in attentional states that may be associated with task difficulty.

## Intracranial EEG analysis

**Preprocessing and artifact rejection.** Intracranial EEG data were demeaned, linearly de-trended, locally re-referenced (bipolar derivations to the next adjacent lateral contact) and if necessary down-sampled to 512 Hz. To remove line noise, data were notch-filtered at 50 Hz and all harmonics. Subsequently, a neurologist visually inspected the raw data for epileptic activity. Channels or epochs with interictal epileptic discharges (IEDs) and other artifacts were removed.

**Trial definition.** We extracted 10 s long, partially overlapping trials to prevent edge artifacts in subsequent filtering. We excluded all stop trials and focused subsequent analyses on go trials. Trials were event-locked to the HLL unless otherwise stated. We considered both correct and incorrect trials for all following neural analyses in order to maximize the number of trials. Specifically, we included both trials in which participants released the button within the lower and upper limit (correct trials; 94.7%) and trials in which participants released the button after the upper limit passed (incorrect trials; 4.8%). Collectively, we excluded trials in which participants released the button prior to the lower limit.

**Definition of regions of interest.** The pre-selection of electrodes was guided by our question on how the human prefrontal-motor network is engaged during context-dependent computations. Electrodes were classified into discrete PFC and motor ROIs based on surface anatomy using the Anatomical Automatic Labeling atlas (ROI_MNI_V4.nii[70]; accessed via FieldTrip[64]). Electrodes in the following areas were considered to be in the PFC ROI (equal for both hemispheres): superior frontal gyrus (orbital, medial and dorsolateral part), medial frontal gyrus, inferior frontal gyrus (opercular, triangular and orbital part). Electrodes in the following areas were considered to be motor electrodes (equal for both hemispheres): precentral gyrus, supplementary motor area, paracentral lobule. In total, 17 patients were implanted with clean, artifact-free electrodes in PFC, 14 patients in motor cortex and 14 patients were implanted with clean, artifact-free electrodes in both ROIs.

**HFA extraction.** The extraction of the high-frequency activity time series was conducted in a three-step process. In the first step, we bandpass-filtered the raw data epochs (10 s) between 70 and 150 Hz into eight, non-overlapping 10 Hz wide bins. We then applied the Hilbert transform to obtain the instantaneous amplitude of the filtered time series. In a last step, we normalized the high-gamma traces using a bootstrapped baseline distribution[71,72]. This involved randomly resampling baseline values (from −0.2 to −0.01 s relative to cue onset) 1000 times with replacement and normalizing single high-gamma traces by subtracting the mean and dividing by the standard deviation of the bootstrap distribution. The high-gamma traces were finally averaged across the eight bins. This procedure mitigates the effect of the 1/f power drop-off and enables comparable estimates across different conditions by minimizing the influence of different baseline distributions onto task-related activity.

**Context-dependent neural information.** We identified context-encoding electrodes using a well-established information theoretical approach that has been used in both human and non-human primate studies[22–24,73]. We employed a one-way analysis of variance (ANOVA) to quantify the percentage of HFA variance that could be explained by our behavioral regressor predictive context. The amount of percent explained variance was quantified using $\omega^2$ as

$$\omega^2 = \frac{SS_{between-groups} - (df \times MSE)}{SS_{total + MSE}} \tag{1}$$

where $SS_{total}$ reflects the total sum of squares across $n$ trials,

$$SS_{total} = \sum_{i=1}^{n} (x_i - \bar{x})^2 \tag{2}$$

$SS_{between-groups}$ the sum of squares between $G$ groups (e.g., factor levels),

$$SS_{between-groups} = \sum_{group}^{G} n_{group} \left( \bar{x}_{group} - \bar{x} \right)^2 \tag{3}$$

MSE the mean square error,

$$\text{MSE} = \sum_{i=1}^{n} (x_i - \bar{x}_{group})^2 \tag{4}$$

and $df$ the degrees of freedom specified as $df = G - 1$. In order to obtain a time series of context-dependent neural information, we estimated $\omega^2$ using a sliding window of 50 ms that was shifted in steps of 2 ms. Electrodes that exhibited a significant main effect of predictive context for at least 10% of consecutive samples across the trial segment were defined as context-encoding electrodes[22,71,74,75]. Note that this approach was blind with respect to both direction and timing of the effect. Finally, to minimize inter-individual variance and maximize the sensitivity to identify a temporally consistent pattern that accounts for most of the variance explained by predictive context within the context-encoding electrodes across participants, we employed principal component analysis (PCA)[24,75]. PCA was applied to the $F$ value time series concatenated across participants (channel x time matrix)[24,76]. In order to define PCs that explain a significant proportion of variance in the data, we used non-parametric permutation testing to determine the proportion of variance that can be explained by chance (Supplementary Fig. 14). We randomly shuffled the $F$ value time series 1000 times to test the null hypothesis that there is no temporal structure present in the data. Electrodes that exhibited a strong weight (75th percentile) on any of the high variance-explaining PCs as determined by their coefficients were defined as context-encoding. This analytical approach classified electrodes to be context-encoding for 16 patients in PFC (time-locked to HLL; 17 patients showed context-encoding electrodes when time-locked to the behavioral response) and for 11 patients in motor cortex.

We used context-encoding electrodes for univariate analyses (Figs. 1–4f). Instead, we used all available electrodes (context-encoding and non-encoding electrodes) for multivariate analyses (Figs. 4g, 5; Supplementary Figs. 6–12).

**HFA peak analyses.** HFA peak amplitude and timing were estimated on a trial-by-trial basis and used as a proxy of strength and timing of the neural responses, respectively. Amplitude and latencies below the 2.5th or above the 97.5th percentile per channel were considered as outliers and removed from further statistical analysis.

**HFA single trial regression to behavior.** Peak amplitude and latency were computed as described above (see HFA peak analyses) and regressed against behavior (RT) via linear regression. We quantified the neuro-behavioral relationship using both full (peak amplitude + latency ~ behavior) and partial linear models (peak amplitude/latency ~ behavior).

**Estimation of ramping dynamics.** To estimate ramping dynamics on a trial-by-trial basis, we quantified the slope of single trial HFA traces using robust linear regression. The slope was estimated from trial start to the HLL.

**Time-frequency decomposition.** We decomposed the raw data into the time-frequency domain using the multitaper method based on discrete prolate spheroidal Slepian sequences in 33 logarithmically spaced bins between 0.5 and 128 Hz. Temporal and spectral smoothing was adjusted to approximately match a 200 ms time window and ¼ octave frequency smoothing. To avoid edge artifacts and allow for resolving low frequency activity, decomposition was performed from ±2 s. surrounding the HLL. As for the HFA analysis (see HFA extraction), we normalized the time-frequency data per frequency bin using a bootstrapped baseline distribution (from −0.4 to −0.1 s relative to cue onset). Power values were z-transformed according to the means and standard deviations of the bootstrapped distribution.

**Spectral slope estimation.** Spectral estimates were obtained by means of a fast Fourier transform (FFT) for linearly spaced frequencies between 1 and 45 Hz after applying a Hanning window and zero padding the data to obtain a fine-grained frequency resolution of 0.25 Hz to improve subsequent background activity estimation. In order to get an estimate of the aperiodic background activity of the power spectrum, we utilized irregular-resampling auto-spectral analysis (IRASA)[33]. IRASA takes advantage of the fact that resampling the original time series by a non-integer resampling factor will leave the 1/f background activity unchanged while systematically shifting the peak frequency at the scale of resampling. Thereby, IRASA disentangles the spectrum into oscillatory (periodic) and 1/f (aperiodic) components. We used the original resampling parameters 1.1 to 1.9 in steps of 0.05[33] that have also been used in a variety of previous studies[31,71,77]. In a next step, we quantified the spectral slope by means of applying a linear fit to the aperiodic power spectrum in log-log space between 30 and 45 Hz as suggested previously[29].

**Time-resolved sample entropy.** Sample entropy reflects an information-theoretic measure and captures the complexity of natural time series data[78]. Sample entropy is defined as the negative natural logarithm of the conditional probability that two sequences similar for $m$ data points will still match when another data sample ($m + 1$) is added to the sequence:

$$\text{SampEN}(m, r, N) = -\log\left(\frac{p^{m+1}(r)}{p^m(r)}\right) \tag{5}$$

where $m$ defines the sequence length, $r$ the similarity criterion and defines the tolerance with which two points are considered similar, and $N$ the length of the time series to be considered for analysis ($m = 2$ and $r = 0.2$[78,79]). In order to obtain a time series of sample entropy, we estimated sample entropy using a sliding window of 100 ms that was shifted in steps of 20 ms. Resulting sample entropy time series were smoothed using a 5 ms boxcar window to attenuate trial-by-trial variability.

**HFA peak-triggered average.** We conducted a peak-triggered average analysis in order to test (1) whether the HFA is nested into ongoing oscillatory activity, and (2) whether the strength of oscillatory activity is context-dependent. This approach is conceptually similar to spike-triggered averaging used in single unit electrophysiology[80]. Therefore, we detected peaks in the single-trial HFA traces and re-aligned the raw unfiltered data to the detected peak events (segmented ±0.5 s surrounding the peaks). To assess the spectral content of the underlying raw traces, we obtained spectral estimates by means of a FFT for linearly spaced frequencies between 1 and 30 Hz after applying a Hanning window and zero padding the data to obtain a frequency resolution of 0.25 Hz. We used IRASA (same parameters and settings as for spectral slope estimation) to discount the aperiodic component. Oscillatory residuals were extracted by subtracting the aperiodic spectral component from the original power spectrum.

**HFA inter-peak-interval.** The speed of the HFA traces was quantified by means of computing the interval between two adjacent peaks. We estimated the inter-peak-interval (IPI) on single trials and transformed the distance into frequencies (sampling frequency divided by the time interval between two adjacent peaks). The instantaneous frequency of the HFA amplitude modulation was inferred by the mean of the distribution.

**Connectivity estimates.** We calculated phase-based connectivity metrics between PFC and motor cortex electrodes to infer inter-areal interactions. We first established the presence of undirected phase-based connectivity between PFC and motor cortex by means of the

imaginary phase-locking value (iPLV). The iPLV was computed for center frequencies between 3 and 32 Hz (± center frequency/4), logarithmically spaced in steps of $2^{1/8}$ after band-pass filtering and applying the Hilbert transform[81,82]. Only considering the imaginary part of the phase-locking value removes zero-phase lag contributions[83]. The iPLV was computed as:

$$\text{iPLV}_f = \left| \text{imag}\left( n^{-1} \sum_{t=1}^{n} e^{i(\phi_{xt} - \phi_{yt})} \right) \right| \qquad (6)$$

where $n$ is the number of time points and $\phi$ reflects the phase angles from electrode $x$ and $y$ at time $t$ and frequency $f$. We first identified the electrode in motor cortex that explained most behavioral variance using linear regression (regressor = HFA timing; response variable = RT). This substantially reduced the degrees of freedom in terms of prefrontal-motor electrode combinations. We then quantified the iPLV between all PFC electrodes and the motor cortex electrode explaining most of the behavioral variance. We have chosen to use behavioral variance explaining electrodes in motor cortex, and not in PFC as motor cortex reflects the final cortical output station to direct behavior[84]. To normalize undirected connectivity, we obtained a permutation distribution by randomly shuffling trial vectors and re-computing the iPLV for every random partition. We further randomly resampled the permutation values 1000 times to approximate a normal distribution. The resulting mean and standard deviations of the bootstrapped permutation distribution were then used to z-normalize the iPLVs. Having established the presence of inter-areal connectivity, we used the phase-slope-index (PSI)[36] to infer directional connectivity between PFC and motor cortex. We focused our PSI analysis on the low-frequency range given that we observed true oscillatory activity within the low-frequency theta band (Fig. 4c). We employed an individualized measure of the PSI using participant-specific peak iPLV frequencies between 2 and 13 Hz (computed separately per prefrontal-motor electrode pair and using the grand average across all trials) in order to maximize sensitivity and prevent spurious inference on directional prefrontal-motor connectivity[72]. Channel-pairs without a distinct iPLV peak between 2 and 13 Hz were discarded from the analysis. We computed the PSI between prefrontal-motor electrode pairs on segmented data (zero-padded by 2 s on every side) using the corresponding peak iPLV frequency (±3 Hz frequency boundary; linearly spaced). PSI values were z-normalized by means of a permutation distribution that was created by randomly shuffling the frequencies in one vector and recomputing the PSI (1000 iterations)[72]. Note that we used both context-encoding and non-encoding electrodes for undirected and directed connectivity estimates to sample the entire network population.

## Population dynamics

**Multidimensional distance.** The activation state of the full neural population at time $t$ can be represented as a point in a $n$-dimensional coordinate system where $n$ reflects the number of electrodes (state space). The neural dynamics between the activation state at time $t$ and time $t + t_n$ can then be represented as a trajectory through this $n$-dimensional state space[10,16,85]. We quantified the population dynamics by means of the HFA as a proxy for local population activity[17–19]. To investigate whether neural trajectories in the state space are context-dependent, we computed the Euclidean distance between pairwise neural trajectories (e.g., 0% and 75% likelihood of stopping) and then summed the pairwise distances. We used a sliding window of 50 ms that was shifted by 20 ms in time to obtain a time series of multidimensional distances (Supplementary Fig. 6a, b). We smoothed the time series using a 25 ms boxcar window to attenuate trial-by-trial variability.

**Euclidean state transitions.** We also quantified transitions within neural trajectories separately per context condition (Supplementary Fig. 6a, b). Thus, we computed the Euclidean distance on single trial trajectories between two adjacent 50 ms time windows that were overlapping for 20 ms.

**Dimensionality reduction (PCA).** We used principal component analysis (PCA) to identify linearly uncorrelated population activity patterns and construct a low-dimensional manifold that is embedded in the neural state space spanned by the recorded depth electrodes. We performed PCA on a two-dimensional data matrix (channel x time, trial) locked to either the HLL or to the movement onset. All trials were used to construct the PC-space. The resulting matrix (component x time, trial) was then reshaped into a three-dimensional matrix (trial x component x time) which allowed us to perform single trial analysis in PC space.

**Identification of coding dimensions.** While the top PCs reflect a set of orthogonal dimensions that are optimized to capture maximum variance, they might not always reflect the computationally-relevant subspaces. We used linear discriminant analysis (LDA) to identify the dimensions that carry maximal information about neural dynamics linked to context-integration and action planning. We therefore trained two linear classifiers on the PC data. The first classifier was trained to discriminate the type of predictive context, and the second one was trained to discriminate behavioral performance (RT; split into terciles; referred to as action). This procedure allowed us to dissociate neural dynamics linked to the integration of predictive context from subsequent dynamics linked to the planning of motor actions. We split the data into training and testing sets using tenfold cross-validation. Because results obtained from cross-validation are stochastic by nature (due to the random assignment of trials into folds), we repeated the analysis five times and then averaged across the repetitions. We applied the LDAs to all PCs in order to identify the dimension that carries most information about our latent variable of interest (note that we only considered PCs that cumulatively explained 99% of the variance and discarded the remaining PCs from the decoding analysis). Decoding traces were then smoothed via application of a 25 ms boxcar window. We applied a threshold at chance level to the resulting decoding time series (~33% for both context and action) and set values below chance level to zero. Next, we identified clusters in the decoding time series (adjacent non-zero values) and summed the classification accuracies within each cluster. We defined the PC with the maximum decoding accuracy (largest cluster) as the dimension coding for the latent variable (context or action; referred to as action or context coding dimension). We further created a permutation distribution of classification accuracies by randomly shuffling the trial labels and re-computing the largest cluster from the resulting decoding time series 50 times. We then contrasted the classification accuracies (cluster values) of the identified coding dimension with the generated permutation distribution. We only considered the coding dimension to be valid if the true cluster exceeded the 95th percentile of the permutation distribution. Importantly, we further constrained the dimensions coding for context and action to be orthogonal (distinct PCs). This, however, was empirically the case without adding constraints in 11/14 participants in PFC ($p = 0.057$; Binomial test) and 10/10 participants in motor cortex ($p = 0.002$).

**Cross-regional pattern analysis.** To quantify whether a discriminative action-specific pattern present in PFC is equally present in motor cortex, we trained a linear classifier on every time point in the action subspace (principal component maximally discriminating action) in PFC and subsequently applied it on every time point in the action subspace in motor cortex. Cross-validation was not necessary since training and testing datasets were independent. Finally, classification

values were tested against chance level and corrected for multiple comparison using cluster-based permutation statistics.

**Cross-correlation analysis.** We computed the cross-correlation of neural activity between the action subspaces in PFC and motor cortex in order to examine their temporal relation on a trial-by-trial basis. The time lags were then averaged across trial for each participant. Negative time lags indicate that neural activity within the motor cortex action subspace temporally lags neural activity within the PFC action subspace and vice versa for positive time lags.

**Decoding control analyses.** We performed additional control analyses to ensure that our two identified subspaces (context and action) capture dissociable processes (Supplementary Fig. 10). We tested whether decoding performance was still above chance level when the action dimension was used to predict context and vice versa. Non-significant classification performance would imply that these two subspaces capture distinct processes.

**Determination of oscillatory components in PC space.** We obtained spectral estimates for all PCs using IRASA (see Spectral slope estimation) for linearly spaced frequencies between 2 and 13 Hz. We then identified the PC with the strongest power in this frequency range.

**PC-based functional connectivity.** We determined the functional connectivity between PCs using power correlations. We computed the correlation coefficient between PC single-trials in PFC and motor cortex from $-0.5$ s to 0.3 s with respect to movement onset. This ensured that all trials contained an equal number of data samples, thereby avoiding potential confounds in the correlation value simply due to variable reaction times across trials. To compare the power correlation across conditions, we normalized the correlation coefficients based on a permutation distribution. We generated the permutation distribution by a random block swapping procedure. This procedure was repeated 1000 times on a trial-by-trial basis to obtain a permutation distribution. Correlation coefficients were then z-transformed using the mean and standard deviation of the permutation distribution.

### Statistical analysis

**Analysis of variance (ANOVA).** Data were aggregated into ROIs (averaged across electrodes) for statistical testing. We performed a one-way repeated-measures ANOVA using predictive context as a within-subject factor to analyze behavior (Fig. 1c), HFA peak latency/ amplitude (Fig. 2a, b), HFA ramping activity (Fig. 3a, b), aperiodic slope (Fig. 3e, f), inter-peak interval (Fig. 4f) and phase-slope index (Fig. 4h). Since not every participant was implanted with electrodes in both PFC and motor cortex, we computed the ANOVA separately for both cortices to estimate the main effect of context onto our latent variable. Significant ANOVA effects were followed by post-hoc testing (two-tailed and corrected for multiple comparisons using the Benjamini-Hochberg procedure[86]). We computed the interaction effect between context x region of interest (PFC, motor cortex) using only a subset of participants that were implanted with electrodes in both regions ($N = 11$). We considered participant data where z-scores exceeded 3$^{rd}$ standard deviation as outliers. In cases where data normality or equal variances assumptions were violated, non-parametric tests were performed in those cases. Specifically, Friedman tests were applied to repeated measures (Supplementary Fig. 2c), Kruskal-Wallis test to one-way, between-region contrasts (Supplementary Fig. 12), Wilcoxon signed-rank tests to paired samples (Fig. 5d; Supplementary Fig. 2a, b) and Wilcoxon rank-sum tests to unpaired samples.

**Linear mixed effect models.** We confirmed the ANOVA results using linear mixed effect models. Participants were treated as random

effects while context and ROI were treated as fixed effects in our model. This approach has been used in previous studies involving human intracranial EEG recordings[72,87]. Model testing was obtained by likelihood ratio tests to compare the models with and without an interaction term (context x ROI). Linear mixed effect models largely confirmed the ANOVA results and are reported in Supplementary Table 2.

**Non-parametric cluster-based-permutation analysis.** We used non-parametric cluster-based permutation testing[88] (as implemented in Fieldtrip[64]) to analyze data in the time (Fig. 1d; 2a/b; 3e/f; 5c; Supplementary Fig. 6a–d, g, h), frequency (Fig. 4c, g) or time-frequency (Fig. 3c, d) domain (Monte Carlo method; 10,000 iterations; maxsum criterion; two-tailed). Clusters were formed by thresholding a dependent $t$-test at a critical alpha of 0.05. We generated a permutation distribution by randomly shuffling trial labels and recomputing the cluster statistic. The $p$-value was then obtained by contrasting the true cluster statistic against the permutation distribution. Clusters were considered to be significant at $p < 0.05$. We also computed interaction effects (context x ROI) using cluster-based permutation testing. We therefore contrasted the difference between two context conditions (75% and 0% likelihood of stop) obtained per ROI using dependent $t$ tests (only performed on a subset of participants that were implanted with electrodes in both regions). Clusters were considered significant at $p < 0.05$. Note that the cluster-level test statistic reported throughout the text refers to the sum of the F- or t-values in the cluster.

**Bootstrapping.** To control for trial differences across conditions, we used a bootstrap procedure. We randomly resampled as many trials from the two context conditions (0% and 25% likelihood) as there were trials in the 75% condition. This procedure was repeated 500 times, if not stated otherwise. The bootstrapped mean was then considered the final value for the conditions with a higher-trial count[89].

### Reporting summary
Further information on research design is available in the Nature Portfolio Reporting Summary linked to this article.

## Data availability
The conditions of the ethical approval of this study do not permit public archiving of raw data. The patients have not consented to making their data publicly available, and the ethical approval conditions from the Regional Committees for Medical and Health Research Ethics (REC) do not permit the public archiving of study data. Readers seeking access to the raw data should contact the co-author Tor Endestad (TE), Department of Psychology, University of Oslo at tor.-endestad@psykologi.uio.no. Requests must meet the following conditions to obtain the data: A short study plan of the proposed research, a data-sharing agreement, and a formal ethical approval. The study plan would be evaluated by the project PI at the University of Oslo (TE), the head of the Department of Neurosurgery at Oslo University Hospital, and the head of research at the Department of Psychology. Next, the PI (TE) would seek REC permission to share raw data with the researcher/institution. After approval, the head of research at the Department of Psychology, the Data Protection Officer at Oslo University Hospital, and the other interested party would sign data transfer agreements before data transfer would take place. Source data are provided with this paper.

## Code availability
Freely available software and algorithms used for analysis are listed where applicable. Analysis code is available at https://github.com/ JanWeber-neuro/Weber_iEEG_NatCommun[90] (https://doi.org/10.5281/ zenodo.10350101).

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

## Acknowledgements

This work was funded by the Baden Wuerttemberg Foundation (Postdoc Fellowship; R.F.H.), German Research Foundation, Emmy Noether Program (DFG HE8329/2-1; R.F.H.), Hertie Foundation, Network for Excellence in Clinical Neuroscience (R.F.H.), the Research Council of Norway (grant number 240389; A.K.S., T.E., P.G.L.), the Research Council of Norway (Centre of Excellence scheme, grant number 262762; RITMO,

RITPART International Partnerships for RITMO Centre of Excellence, grant number 274996; A.K.S., T.E., P.G.L.) and by a NIMH Conte Center Grant (1 PO MH109429, R.T.K.) and the NINDS (2 R01 NS021135, R.T.K.). Publication of this work was support by the Open Access Publication Fund of the University of Tübingen.

## Author contributions

Conceptualization: T.E., A.K.S., R.F.H.; Methodology: J.W., R.F.H.; Investigation: A.K.S., T.E., A.O.B., A.L., I.F., S.K., P.L., J.I., R.T.K.; Visualization: J.W., R.F.H.; Funding acquisition: A.K.S., R.T.K., T.E., R.F.H.; Project administration: T.E., A.K.S., R.F.H.; Supervision: R.F.H.; Writing – original draft: J.W., R.F.H.; Writing – review & editing: A.K.S., T.E., A.O.B., A.L., I.F., S.K., P.L., J.I., R.T.K.

## Funding

## Competing interests

The authors declare no competing interests.
