## [Peer Review File · Nature Communications]

Reviewers' Comments:

Reviewer #1:

Remarks to the Author:

Weber et al. investigate electrophysiological dynamics in prefrontal and motor cortices during a motor task. The task required subjects to release a button when a horizontal bar reached a certain point on the screen. The likelihood with which the bar reached that point was modulated, and subjects were informed of that likelihood on the start of each trial (termed behavioural context). The results show that a continuous ramping of high-frequency activity tracked context and action, whereas in the motor cortex high-frequency activity only tracked action. In addition, theta oscillations appeared to mediate the prefrontal to motor cortex connectivity to implement the motor response. The authors conclude that their data suggest the existence of two orthogonal but complimentary neural signatures, high-frequency activity and theta oscillations, which together guide context-dependent behaviour.

General assessment:

In general I found the manuscript very interesting to read and the data quite compelling. The analyses seem rather sound, although I want to point out that I am not an expert in state space analysis and hence cannot comment on the soundness of that aspect. With that in mind, I am generally supportive of this work as I think it represents a thorough investigation of the role of prefrontal population activity and theta oscillations during context dependent behaviour. I also think there are a couple of places where the manuscript can be improved, and have a few questions about the methods which I list below in chronological order as they occur in the manuscript.

Specific comments:

1. Page 7, line 30-31. Were only correct trials used for this analysis? I assume so, because otherwise the effects could be driven by a difference in the ratio between correct/incorrect trials. In general, I found it difficult to ascertain for which analysis only correct trials were used, and which one considered both correct and incorrect trials. I think this could be made more clear.
2. Page 8, line 19. This conflicts with what is said earlier, i.e. that 27% of motor cortex electrodes do encode context, which left me confused.
3. Page 12, lines 6-7. How can a 2x3 interaction be tested with a t-test? I understood this only after reading through the methods, i.e. that the authors tested this by contrasting the two extreme context manipulations, but that doesn't become clear here. This should be clarified.
4. Page 19, line 17. The t-value and p-value reported here don't add up. A t-value of 39.32 at 13 df should be highly significant unless I am missing something here.
5. Page 24, lines 12-14. I could not quite understand how the shift from oscillatory to continuous ramping is actually present in the data. Which data points specifically indicate this shift? Please clarify.
6. Page 26, line 4-5. Similar to the above, I could not really ascertain what the evidence is for a "hand-off" of the action plan from PFC to motor cortex. Is this solely based on the PSI data? If yes, then PSI would only serve to indicate directed connectivity, but not necessarily transfer of a specific type of information (i.e., action plan). In order to be able to claim that multi-variate connectivity measures would need to be used, unless I am missing something here.
7. Page 26, lines 18-19. The authors assume causality here, i.e., oscillations to control neural firing, but no such causal evidence is presented. As it stand, I think, the authors can only claim a correlation between oscillations and HFA (i.e., the proxy for neural firing).
8. Page 35, line 2. How was the 10% threshold decided? Was there a constraint that these 10% would have to cover adjacent (i.e. connected) time bins?
9. Page 35, lines 8-9: Is it legit to apply a PCA on the F values? I only know PCA being applied to the raw or filtered data using the co-variance as a basis for data reduction but haven't seen cases where PCA was applied on second-level statistical data.
10. Page 35, lines 11-12: I am not sure the randomization procedure here actually tests for the proportion of variance that can be explained by chance. If the authors want to do that, then I think a randomization procedure that shuffles trial labels would be more appropriate (as opposed to shuffling F value time series).
11. Page 27, lines 1-3. This would only be correct if $1/f$ doesn't change between pre and poststimulus interval. However, the $1/f$ is expected to change from pre to post-interval simply

because of high-frequency power increases and low-frequency power decreases induced by a stimulus/task (<https://pubmed.ncbi.nlm.nih.gov/25855698/>).

Reviewer #2:

Remarks to the Author:

In this paper, Weber et al. examined population coding in PFC and motor cortex under predictable and unpredictable response conditions. They recorded intracranial EEG from epilepsy patients and made multiple notable findings by analyzing high-frequency activity. First, as the probability of unpredictable cue increases (higher response uncertainty), PFC activity appears to switch from oscillatory to ramping responses. The author claims that this ramping activity reflects task context (response uncertainty, in this case) and this is absent in the motor cortex. Second, they showed that this context encoding was orthogonal to action coding, which was related to the dimension of theta oscillation. Finally, they showed this theta oscillation dimension was coupled between PFC and motor cortex. Together, the authors conclude that context information scaled with uncertainty is encoded in PFC orthogonal to action encoding, while action encoding is coupled with the motor cortex through theta oscillation that mediates the transfer of action plans.

The conclusions of the paper are very interesting; it demonstrates a potential link between context-dependent computations in PFC and inter-areal communications from PFC to the motor cortex through oscillation. The amount of analyses they performed is quite impressive, and the paper is written well, but it was still hard for me to process the results, probably because I did not agree with the authors' interpretations in a number of places in the manuscript.

Major

1. The authors claim that they tested context-dependent activity in PFC and motor cortex. This is true in a broad sense, but what's exactly happening in the task is a monotonic increase in uncertainty of an event across conditions, or more specifically, increase in task difficulty, which is reflected in behavior as lower accuracy and higher reaction time. Enhanced PFC signals for greater task difficulty (potentially reflecting higher task engagement/attention) are something very common, and I feel like, calling them context encoding is like masquerading as something that sounds more important and relevant to specific computations in the task. I wonder if the authors have strong justification of why these signals should be taken seriously as context encoding. For example, is there proof that the activity really reflects subjects' changes in strategy rather than just changes in overall performances?

2. The authors conclude that PFC activity shifts from an oscillatory to a ramping mode with increased uncertainty (Page 2 L4-6), but is it true? Figure 2 seems to be the support of this claim, but Figure 4 shows only a slight decrease in oscillatory power across conditions. And Figure 6c shows no encoding of context along theta oscillation. So, isn't it more like oscillation is always present, but just the ramping activity becomes stronger with increased uncertainty (hence task difficulty)?

3. The authors claim that the ramping activity reflects evidence accumulation, but unfortunately I did not understand this part. What evidence is accumulated in this task? Why should more evidence be accumulated in uncertain conditions?

4. The motor cortex appears to be showing ramping dynamics all the time (Figure 3b). How do the authors interpret this? The authors claimed that the dimension of theta oscillation is particularly relevant to action execution, but how about the dimension of ramping activity? I feel like this is more relevant to action execution (like the representation of hazard rate; Janssen & Shadlen 2005). Then, why specifically focus on theta oscillation as encoding of action plans?

5. Does the motor cortex encode context (uncertainty) or not? I think the paper is giving mixed results. In earlier figures (Figure 1d), the motor cortex is shown to have context information as strongly as PFC. But later, all the analyses deny the presence of context dependency in the motor cortex. How could this happen? How does this relate to the conclusions?

6. The authors claim significant encoding of action based on significant classification accuracy of action; but here, the classifier is trained to discriminate trials with different reaction times. First, I am not entirely clear if it could be said as encoding of action. The plots of classifier accuracy are aligned to the timing of button press, so the same action is happening in every trial regardless of reaction times. Then it is not classifying different actions. Second, reaction times are highly correlated with uncertainty (Figure. 1C), so I am afraid that the classification performance is easily confounded with the encoding of context (or task difficulty) or vice versa. How do the authors justify that this is not the case?

7. The authors claimed the encoding of context and action is orthogonal in state space (Page 19 L13-), but that's a strong claim and unfortunately, I couldn't fully understand how the authors reached this conclusion. Is it based on the finding that the PCs best decoding context and action were different? This could mean that the two axes are not aligned but does not necessarily mean they are orthogonal. To test this, the authors should compute the angle between the two decoding axes and show that it is indistinguishable from 90 degrees in state space (through a permutation test or some sort). If the authors just wanted to claim these two axes are not aligned (but not necessarily orthogonal), then the test would be whether the angle is significantly greater than 0 deg or not.

8. Page 22, L12-: the authors stated that "the dimension with the strongest theta power most likely matched PC1", but I was confused. Isn't PC1 supposed to carry context information (Figure 5c)? But here, the authors claim there is no context information along this axis (Figure 6c).

Minor

1. Page 5, L19-: It would be great if the authors could briefly mention the definition of RTs and accuracy around here. I was initially confused what is the definition of accuracy in this task (and had to check methods).

2. All figures: all legends and axis labels are a bit too small and hard to read.

3. Figure 1e: Significant and non-significant sites are drawn using large and small spheres, but it was also a bit difficult to distinguish them. Perhaps use different colors?

4. Figure 2d: it looks like motor cortex activity is also shifting from oscillatory to ramping state. But this is not what the authors wanted to say? Maybe this was not the best example?

5. Page 31, L8-: in uncertain conditions, when could the target stop? Uniform probability over all the time, or flat hazard rate?

Point-by-Point Reply: NCOMMS-22-05636

We would like to thank the reviewers for their thoughtful and constructive comments. Below, we provide point-by-point responses to all queries. All changes in the revised manuscript are highlighted with track changes.

Reply formatting guide:

Reviewer remarks: **bold, black.**

Author response: *blue.*

Changes in the manuscript: *italics, blue.*

Reviewer #1 (Remarks to the Authors)

Weber et al. investigate electrophysiological dynamics in prefrontal and motor cortices during a motor task. The task required subjects to release a button when a horizontal bar reached a certain point on the screen. The likelihood with which the bar reached that point was modulated, and subjects were informed of that likelihood on the start of each trial (termed behavioural context). The results show that a continuous ramping of high-frequency activity tracked context and action, whereas in the motor cortex high-frequency activity only tracked action. In addition, theta oscillations appeared to mediate the prefrontal to motor cortex connectivity to implement the motor response. The authors conclude that their data suggest the existence of two orthogonal but complimentary neural signatures, high-frequency activity and theta oscillations, which together guide context-dependent behaviour.

General assessment:

In general, I found the manuscript very interesting to read and the data quite compelling. The analyses seem rather sounds, although I want to point out that I am not an expert in state space analysis and hence cannot comment on the soundness of that aspect. With that in mind, I am generally supportive of this work as I think it represents a thorough investigation of the role of prefrontal population activity and theta oscillations during context dependent behaviour. I also think there are a couple of places where the manuscript can be improved, and have a few questions about the methods which I list below in chronological order as they occur in the manuscript.

We thank the reviewer for this encouraging review and the constructive feedback that substantially improved the manuscript.

Specific comments:

1. Page 7, line 30-31. Were only correct trials used for this analysis? I assume so, because otherwise the effects could be driven by a difference in the ratio between correct/incorrect trials. In general, I found it difficult to ascertain for

which analysis only correct trials were used, and which one considered both correct and incorrect trials. I think this could be made more clear.

We agree that the previous version of the manuscript lacked a proper description regarding which trials were considered. We indeed used both correct and incorrect trials in all our analyses of neural data. Specifically, we included both trials in which participants released the button within the lower and upper limit (correct trials) and trials in which they released the button only after the upper limit was passed (incorrect trials; **Response Figure 1** for the data distribution of RTs and the limits where trials were considered to be correct/incorrect).

We have decided to include both correct and incorrect trials in order to maximize the number of trials available (in particular for the 75% likelihood condition). This is a direct consequence of the clinical time constraints, which limits the total number of trials that can be recorded during any given experiment. In the revised version of the manuscript, we now explicitly state that we used both correct and incorrect trials and provide a clear definition.

Response Figure 1. Distribution of reaction times for correct and incorrect trials. We used both correct and incorrect trials for all neural analyses. Correct trials were defined as trials in which participants released the button within the lower and upper limit (correct trials). Incorrect trials were defined as trials in which participants only released the button after the upper limit passed.

We now describe these criteria in the main manuscript:

Page 38, Line 2-7

“We considered both correct and incorrect trials for all following neural analyses in order to maximize the number of trials. Specifically, we included both trials in which participants released the button within the lower and upper limit (correct trials; 94.7%) and trials in which participants released the button after the upper limit passed (incorrect trials; 4.8%). Thus, we only excluded trials in which participants released the button prior to the lower limit.”

However, to ensure that the effect of was not driven by a difference in the ratio between correct and incorrect trials, we re-computed the context-dependent

neural information while accounting for accuracy using the same analytical approach as in the previous version of the manuscript (sliding-window ANOVA and extracting the amount of percent explained variance as ω^2 ; two-way ANOVA with factors *context* and *accuracy*).

Importantly, to account for the collinearity between context and accuracy, we employed an unbalanced ANOVA that orthogonalizes the different factors^{1,2} to distill the unique explained variance by every factor. Hence, variance explained by context could not be explained by accuracy.

We observed no reduction in context-dependent neural information after accounting for accuracy (**Supplementary Figure 2**). Collectively, the observed effects cannot be explained by a difference in the ratio between correct/incorrect trials.

Supplementary Figure 2. Context-dependent neural information cannot be explained by accuracy. An unbalanced ANOVA, that implicitly orthogonalized the factors context and accuracy, was computed. Thus, explained variance by context cannot be explained by accuracy, or vice versa. Context-dependent neural information is neither significantly altered in PFC (left) nor motor cortex (right) after orthogonalization of task factors. Hence, context-dependent neural information cannot be explained by accuracy. Lines and shaded regions indicate the mean and SEM.

We added an additional paragraph of this control analysis for clarification.

Page 8, Line 11-15

*“To ensure that the difference in the ratio between correct and incorrect trials did not confound our analysis, we orthogonalized the factors context and accuracy using an unbalanced ANOVA (**Supplementary Fig. 2**). This confirmed the existence of context-dependent neural information, precluding spurious effects as driven by the ratio of correct/incorrect trials.”*

2. Page 8, line 19. This conflicts with what is said earlier, i.e. that 27% of motor cortex electrodes do encode context, which left me confused.

Thank you for highlighting this inconsistency. We agree that our previous statement needs to be nuanced as it is not fully supported by the data and caused confusion. The critical point that we intended to highlight was that we did not find statistical evidence for context-dependent differences in HFA peak amplitudes or peak latencies within motor cortex using single trial estimates.

Instead, only the group-level statistic over time (**Figure 2b**) revealed a significant main effect of context in motor cortex (although post-hoc pairwise comparisons were not significant as compared to PFC). However, this is certainly not evidence

for the absence of context-dependent computations within motor cortex. Instead, this observation suggests that context-dependent computations are more pronounced in PFC. We thus rephrased this paragraph accordingly and emphasized context-dependent computations in PFC.

Page 9, Line 20 – Page 10, Line 2

“Collectively, these findings indicate that context-dependent computations are pronounced in PFC. In contrast, contextual information only marginally modulates activity in motor cortex.”

We would further like to point out that clinical recordings are not well suited to study distinct motor sub-regions, such as the FEF or premotor cortex. Thus, we cannot rule out that context-dependent computations might be highly localized to specific sub-regions in motor cortex. We would like to emphasize that this is an inherent limitation of human intracranial recordings. We added an additional paragraph addressing this limitation.

Page 32, Line 16-20

“However, due to the inherent limited coverage in intracranial recordings, we cannot preclude that specialized sub-regions within the human motor cortex (e.g. anterior or posterior parts of the supplementary motor area, frontal eye fields, premotor cortex) might independently encode contextual information.”

3. Page 12, lines 6-7. How can a 2x3 interaction be tested with a t-test? I understood this only after reading through the methods, i.e. that the authors tested this by contrasting the two extreme context manipulations, but that doesn't become clear here. This should be clarified.

We thank the reviewer for this suggestion. We added an additional explanation within the results section on how a cluster-based interaction was quantified.

Page 13, Line 6-11

“To quantify a context \times ROI interaction effect, we contrasted the difference between the two extreme context conditions (75% and 0% likelihood of stop) obtained per ROI using cluster-corrected dependent t-tests (Methods). This analysis confirmed that low-frequency desynchronization during states of high uncertainty was specific to PFC (2-19 Hz; $\text{sum}(t_{10}) = -548.72$, $P = 0.006$).”

4. Page 19, line 17. The t-value and p-value reported here don't add up. A t-value of 39.32 at 13 df should be highly significant unless I am missing something here.

Thank you for raising this important point. The reported t-value of 39.32 (df = 13) is not significant because it corresponds to the sum of all t-values within a significant cluster. Therefore, the t-values are not comparable to conventional t-values as reported in a t-distribution. We apologize for not making this clear enough.

In order to avoid confusion, we now report cluster-based statistics as: **sum**(t_{xx}) or **sum**($F_{x,x}$) = XX, $P = [..]$, *Cohen's d* = [..].

We now also explicitly state the description of cluster statistics in the methods:

Page 52, Line 5-7

“[...] the cluster-level test statistic reported throughout the text refers to the sum of the F- or t-values in the cluster”.

5. Page 24, lines 12-14. I could not quite understand how the shift from oscillatory to continuous raming is actually present in the data. Which data points specifically indicate this shift? Please clarify.

We apologize for the lack of clarity in our descriptions and agree that calling it a “*shift from oscillatory and continuous ramping*” is misleading and not warranted by the data. What we intended to highlight is that there is a shift from a purely oscillatory towards a mixed “oscillatory-plus-ramping” regime as displayed in **Figure 3a & 4c**.

We interpret this as evidence that oscillations are ubiquitous and primarily synchronize the prefrontal-motor network for inter-areal communication (**Figure 4**). The presence of additional ramping dynamics is driven by a shift in uncertainty as outlined below (reviewer #2, comment #2). We carefully revised the manuscript and rephrased sentences accordingly in several instances, for example:

Page 2, Line 4-7

“We demonstrate that increasing behavioral uncertainty introduces a shift from a purely oscillatory to a mixed mode processing regime with an additional ramping component present in the human prefrontal cortex.”

Page 28, Line 11-14

“In line with the active sensing framework, we show that (II) behavioral uncertainty introduces a shift from an energy-efficient oscillatory to an energy-costly processing mode with mixed oscillatory and ramping dynamics (Fig. 3/4).”

6. Page 26, line 4-5. Similar to the above, I could not really ascertain what the evidence is for a “hand-off” of the action plan from PFC to motor cortex. Is this solely based on the PSI data? If yes, then PSI would only serve to indicate directed connectivity, but not necessarily transfer of a specific type of information (i.e., action plan). In order to be able to claim that multi-variate connectivity measures would need to be used, unless I am missing something here.

Thank you for raising this highly relevant point. We initially based our statement on two main findings.

1. We demonstrated a directed interaction from PFC to motor cortex using the phase-slope index (PSI) as a metric of directed connectivity. However, as correctly noted, this does not allow us to infer that a transfer of a specific type of information is unfolding from PFC to motor cortex.
2. We then showed that subspaces in PFC and motor cortex contained information about the timing of action (classifier trained to distinguish reaction times) and that the onset latency of significant action classification started earlier in PFC as compared to motor cortex (**Figure 5g and 5h**).

Yet, as correctly pointed out, this was insufficient evidence for a hand-off of specific action plans. Therefore, we added three additional analyses to complement these findings.

1. We performed a cross-regional pattern analysis to explicitly quantify the hand-off of a specific neural pattern from PFC to motor cortex that discriminates action encoding. We trained a linear classifier on the PFC action subspace and subsequently tested if the classifier generalized to the motor cortex. In other words, we tested if a discriminative (informative) pattern that was present in PFC was equally present in motor cortex. We time-resolved this decoding approach, since we predicted that the cross-regional generalization should occur at a lag if this truly constitutes a hand-off of a specific information. In this scenario, above-chance classification along the diagonal indicates that the information between PFC and motor cortex is temporally specific without delay, whereas off-diagonal classification indicates the presence of a temporal lag. As shown in **Figure 5j**, a linear decoder trained on the PFC action subspace generalized to the action subspace in motor cortex with a temporal delay, providing further evidence that action-specific information was transferred from PFC to motor cortex.

Figure 5j. Left: cross-temporal classification matrix. A linear classifier was trained to discriminate reaction time based on the action subspace in PFC and subsequently tested for generalization in the motor cortex action subspace. The black outline indicates the extent of significant cross-temporal classification. Note that significant cross-temporal classification is dominant along the off-diagonal, indicating substantial cross-temporal generalization. Prior to the button release, cross-temporal generalization was mainly observed along the upper triangular, indicating that trained decoder in PFC generalizes with a temporal delay in motor cortex. Right: cross-temporal classification within the upper triangular matrix was averaged across the x-axis (training time) and across the y-axis (testing time), yielding a time series vector with the proportion of significant samples with respect to the training time in PFC and the generalization time in motor cortex. This clearly shows a temporal lag of 105ms between PFC and motor cortex, indicating that action information at time t in PFC only generalizes at time $t+n$ in motor cortex, therefore supporting the claim of a hand-off from PFC to motor cortex.

2. We also quantified the temporal dynamics of context and action encoding. Therefore, we computed the peak decoding latency for context in PFC, action in PFC and action in motor cortex. This revealed a clear serial unfolding of events (**Figure 5i**) from initial context integration in PFC followed by action encoding in PFC and motor cortex.

Figure 5i. Onset of peak-decoding performance demonstrating that initial context information in PFC serially unfolds into action encoding within the prefrontal-motor network (PFC context = -0.09 ± 0.19 s.; PFC action = 0.05 ± 0.19 s.; motor cortex action = 0.11 ± 0.12 s; mean \pm SD).

3. We further assessed the temporal relationship between action encoding subspaces in PFC and motor cortex using cross-correlation analysis (**Response Figure 2**). We extracted the lag at which cross-correlation peaked (trial-by-trial basis) to infer the relative timing between the two subspaces. The cross-correlation revealed a significant time lag ($P = 0.019$; Wilcoxon rank sum test) between the action encoding subspaces, demonstrating that the PFC leads motor cortex.

Response Figure 2. Cross-correlation between action subspaces in PFC and motor cortex reveals a significant time-lag ($P = 0.019$; Wilcoxon rank sum test), further supporting the idea that the hand-off of action plans serially unfolds from PFC to motor cortex.

In sum, these three analyses substantially extend our previous findings and provide robust evidence for a hand-off of action plans from PFC to motor cortex.

Therefore, we decided to incorporate these additional analyses into the main manuscript (**revised Figure 5**).

Page 21, Line 6 – Page 22, Line 21

*“Critically, we performed additional control analyses demonstrating that these two subspaces indeed capture dissociable processes related to contextual computations and the planning of actions (**Supplementary Fig. 5**; Methods). Having established that context and action can be decoded from the prefrontal-motor network, we next quantified their temporal dynamics by computing the latency of maximal decoding accuracy. We observed a serial unfolding of events from context integration in PFC (**Fig. 5i**; $P = 0.024$; Kruskal-Wallis; -0.09 ± 0.19 s. with respect to BR; mean \pm SD) towards action-relevant processes in PFC (0.05 ± 0.15 s.) and motor cortex (0.11 ± 0.12 s.), suggesting that contextual computations in PFC are completed prior to action-relevant computations.*

This raises the question, whether and how the representation of the action plan in PFC is passed on to motor cortex for final action execution? To address this question, we first computed the direction of propagation of neural activity between action encoding subspaces in PFC and motor cortex using cross-correlation analysis (Methods). This revealed a significant time lag of activity between the action encoding subspaces ($P = 0.019$; -18.16 ± 16.34 ms; mean \pm SD; Wilcoxon rank sum test). Activity within the PFC action subspace significantly preceded activity within the action subspace in motor cortex, demonstrating that action-relevant processes in PFC temporally precede those in motor cortex.

*Finally, to infer whether specific information is handed-off from PFC to motor cortex prior to the final action, we conducted a cross-regional pattern analysis (Methods). We trained a linear classifier on the activity of the PFC action subspace to discriminate action-timing and subsequently tested if the classifier generalized to motor cortex. Hence, we tested if a discriminative pattern that was present in PFC was equally present in motor cortex. We employed a time-generalization decoding approach, since we predicted that a cross-regional generalization should occur at a lag if this truly constitutes a hand-off of a specific information. In this scenario, above-chance classification along the diagonal indicates that the information between PFC and motor cortex is temporally specific without delay, whereas off-diagonal classification indicates the presence of a temporal lag. A linear decoder trained on the PFC action subspace generalized to the action subspace in motor cortex with a temporal delay (**Fig. 5j**; $\text{sum}(t_{\theta}) = 70605$, $P = 0.002$, Cohen’s $d = 0.64$), providing further evidence that action-specific information was transferred from PFC to motor cortex.*

In sum, this set of findings demonstrates that the human PFC integrates contextual information at early stages using a low-dimensional subspace, devises them into an appropriate action plan and hands-off this action plan to motor cortex for final action-execution.”

Page 49, Line 4-16

Cross-regional pattern analysis. *To quantify whether a discriminative action-specific pattern present in PFC is equally present in motor cortex, we trained a linear classifier on every time point in the action subspace in PFC and subsequently applied it on every time point in the action subspace in motor cortex. No cross-validation was required since training and testing datasets were independent. Finally, classification values were tested against chance level and corrected for multiple comparison using cluster-based permutation statistics.*

Cross-correlation analysis. *We computed the cross-correlation of neural activity between the action subspaces in PFC and motor cortex in order to examine their*

temporal relation on a trial-by-trial basis. The time lags were then averaged across trial for each participant. Negative time lags indicate that neural activity within the motor cortex action subspace temporally lags neural activity within the PFC action subspace and vice versa for positive time lags.

7. Page 26, lines 18-19. The authors assume causality here, i.e., oscillations to control neural firing, but no such causal evidence is presented. As it stand, I think, the authors can only claim a correlation between oscillations and HFA (i.e., the proxy for neural firing).

We fully agree that we cannot draw any causal inference between neural oscillations and neural firing based on the correlative nature of our study. We did not intend to claim that we found neural oscillations to control neural firing. Instead, we intended to state that our result on rhythmic HFA modulation based on theta oscillations complements previous work in animals demonstrating that neural firing is modulated by network oscillations.

We re-organized the sentence to make it explicit that we do not claim causality between neural oscillations and HFA in our study, but refer to previous studies that could indeed show more causal evidence for an oscillatory-based control mechanism of population spiking activity (e.g. Fröhlich & McCormick, 2010³).

Page 31, Line 9-12

“Furthermore, theta oscillations rhythmically structure HFA through phase-amplitude cross-frequency coupling. Thus, our results complement previous findings in animal models demonstrating that neural firing is linked to network oscillations^{3,4}.”

8. Page 35, line 2. How was the 10% threshold decided? Was there a constrained that these 10% would have to cover adjacent (i.e. connected) time bins?

The 10% threshold was based on several prior studies using human intracranial recordings. It should be however noted, that the exact temporal thresholds vary between studies, e.g. 6% in Voytek et al. 2015⁵ or 13% in Kam et al. 2021⁶. We decided to choose a 10% threshold as in Helfrich et al. 2018⁷. Furthermore, as correctly noted, these 10% were constrained to cover adjacent time bins in order to avoid a high type II error rate and follows examples in the literature⁵⁻⁸. We clarified these points in the revised manuscript.

Page 40, Line 6-9

“Electrodes that exhibited a significant main effect of predictive context for at least 10% of consecutive samples across the trial segment were defined as context-encoding electrodes⁵⁻⁸.”

Furthermore, we would like to explicitly highlight that this preselection criterion only applied to the univariate analyses, while no preselection was performed for the multivariate analyses.

Page 41, Line 5-7

“We used context-encoding electrodes for univariate analyses (Fig. 1 – Fig. 4g). Instead, we used all available electrodes (context-encoding and non-encoding electrodes for multivariate analyses (Fig. 4g – Fig. 6).”

9. Page 35, lines 8-9: Is it legit to apply a PCA on the F values? I only know PCA being applied to the raw or filtered data using the co-variance as a basis for data reduction but haven't seen cases where PCA was applied on second-level statistical data.

Thank you for raising this relevant point. Indeed, it is valid to apply PCA on second-order statistical data, such as F values as demonstrated in several previous studies⁹⁻¹¹. The rationale behind this approach is comparable as to when PCA is applied to e.g. filtered data: The overall aim is to identify neural modes that capture a significant proportion of population covariance. However, instead of identifying neural modes as dominant patterns in the HFA covariance matrix, PCA applied to F values aims to identify a low-dimensional pattern that maximally accounts for context-dependent processing. This step enabled us to find a temporally consistent pattern of F value time series that was further used to minimize inter-individual variance for group-level analyses. We now explain this approach in more detail and moreover, refer to two intracranial studies that have employed a similar approach with a comparable rationale.

Page 40, Line 10-15

“Finally, to minimize inter-individual variance and maximize the sensitivity to identify a temporally consistent pattern that accounts for most of the variance explained by predictive context within the context-encoding electrodes across participants, we employed principal component analysis (PCA)^{6,9}. PCA was applied to the F value time series concatenated across participants (channel x time matrix)^{9,10}.”

10. Page 35, lines 11-12: I am not sure the randomization procedure here actually tests for the proportion of variance that can be explained by chance. If the authors want to do that, then I think a randomization procedure that shuffles trial labels would be more appropriate (as opposed to shuffling F value time series).

Thank you for bringing up this important point. As stated in the previous point (remark #9), we applied PCA on the F value time series in order to identify a temporally consistent pattern of context-dependent activity.

A common problem with PCA is distinguishing “significant” from “non-significant” components – more broadly speaking, separating signal from noise. Many previous studies in the field have applied a simple threshold criterion, e.g. retain as many PCs as needed to explain at least e.g. 95% of the variance in the data¹²⁻¹⁴.

Here, we employed a more empirical approach to find a set of relevant PCs. We followed recommendations by M. X. Cohen as provided in *Analyzing Neural Time Series Data*¹⁵, suggesting that “[...] another way to determine a threshold is to use permutation testing, whereby the data are randomly shuffled, a PCA is computed on the shuffled data, and the amount of variance explained in the shuffled data, [...], is taken as the threshold” (page 298; printed version).

As displayed in **Supplementary Figure 7a**, this approach revealed that the average percent variance explained of the permutation distribution was ~1.04%. We considered principal components (PCs) to be significant if they explained more variance as expected by chance. This yielded 9 significant PCs that cumulatively explained 99.76% of the variance (**Supplementary Figure 7b**).

Supplementary Figure 7. Distinguishing significant from non-significant components. **a**, Permutation distribution of explained variance by randomly shuffling the data. The red dashed line indicates the mean variance explained by chance. Principal components (PC) that explained more variance as expected by chance were considered to be significant. PC9 was the final component to be considered significant as it still explained more variance as to be expected by chance. **b**, PC 1-9 cumulatively explained 99.76% of the variance.

Thus, our approach yields qualitatively similar results as compared to prior studies. Yet, it builds upon a more empirical approach to assess the number of significant PCs present in the data (instead of defining an arbitrary threshold).

We also implemented a conceptually similar analysis as suggested by the reviewer. However, instead of performing a randomization procedure by shuffling the trial labels and recomputing the explained variance (a computationally very time-consuming approach), we applied PCA onto the *F*-value time series of the context non-encoding electrodes. This approach is conceptually very similar to shuffling trial labels as non-encoding electrodes contain no systematic context-dependent neural information (cf. **Figure 1d**).

The results of this analysis are depicted below. It is evident that such an approach leads to a highly conservative threshold that would only suggest to retain only the first two PCs that cumulatively explain only 59.8% of the variance (**Response Figure 3b**). In turn, this would imply that potentially meaningful components (e.g. PC3; **Response Figure 3a**) are removed and relevant information is lost.

Response Figure 3. Distinguishing significant from non-significant components. **a**, Time courses of the first three PCs. **b**, Results of a more conservative approach in which we used the explained variance as revealed by context non-encoding electrodes. This approach yielded only two significant PCs that cumulatively explained only 59.8% of the variance. Note that PC3 which also shows a clear time series of context-dependent activity as shown in **(a)** would be neglected by using a more conservative approach.

Therefore, while recognizing that different approaches have their specific limitations, we are convinced that our approach yields an objective, valid quantification to distinguish significant from non-significant components. We have added an additional figure to the supplements (**Supplementary Fig. 7**) to better illustrate our analytical approach taken.

Page 40, Line 15-18

*“In order to define PCs that explain a significant proportion of variance in the data, we used non-parametric permutation testing to determine the proportion of variance that can be explained by chance (**Supplementary Fig. 7**).”*

11. Page 27, lines 1-3. This would only be correct if 1/f doesn't change between pre and poststimulus interval. However, the 1/f is expected to change from pre to post-interval simply because of high-frequency power increases and low-frequency power decreases induced by a stimulus/task (<https://pubmed.ncbi.nlm.nih.gov/25855698/>).

Thank you for this comment and the reference. We did not find this information on page 27 and therefore assumed that the comment referred to *page 37, line 1-3*. We feel that our phrasing of the sentence was misleading. We did not intend to claim that z-normalizing the data mitigates effects that are driven by changes in the spectral slope from pre- to post-stimulus interval, but simply mitigates the drop off of power with increasing frequencies in the high-frequency band signal. However, the statement did not contribute any relevant information to our approach and was therefore removed from the manuscript.

Reviewer #2 (Remarks to the Authors)

In this paper, Weber et al. examined population coding in PFC and motor cortex under predictable and unpredictable response conditions. They recorded intracranial EEG from epilepsy patients and made multiple notable findings by analyzing high-frequency activity. First, as the probability of unpredictable cue increases (higher response uncertainty), PFC activity appears to switch from oscillatory to ramping responses. The author claims that this ramping activity reflects task context (response uncertainty, in this case) and this is absent in the motor cortex. Second, they showed that this context encoding was orthogonal to action coding, which was related to the dimension of theta oscillation. Finally, they showed this theta oscillation dimension was coupled between PFC and motor cortex. Together, the authors conclude that context information scaled with uncertainty is encoded in PFC orthogonal to action encoding, while action encoding is coupled with the motor cortex through theta oscillation that mediates the transfer of action plans.

The conclusions of the paper are very interesting; it demonstrates a potential link between context-dependent computations in PFC and inter-areal communications from PFC to the motor cortex through oscillation. The amount of analyses they performed is quite impressive, and the paper is written well, but it was still hard for me to process the results, probably because I did not agree with the authors' interpretations in a number of places in the manuscript.

We would like to thank the reviewer for taking time to review the manuscript, and for the positive and constructive appraisal.

Major

1. The authors claim that they tested context-dependent activity in PFC and motor cortex. This is true in a broad sense, but what's exactly happening in the task is a monotonic increase in uncertainty of an event across conditions, or more specifically, increase in task difficulty, which is reflected in behavior as lower accuracy and higher reaction time. Enhanced PFC signals for greater task difficulty (potentially reflecting higher task engagement/attention) are something very common, and I feel like, calling them context encoding is like masquerading as something that sounds more important and relevant to specific computations in the task. I wonder if the authors have strong justification of why these signals should be taken seriously as context encoding. For example, is there proof that the activity really reflects subjects' changes in strategy rather than just changes in overall performances?

We would like to thank the reviewer for raising this important point. We fully concur that an increase in uncertainty scales with task difficulty. However, based on our additional analyses, it is unlikely that task difficulty can fully explain our findings for two main reasons:

1. We employed signal detection theoretic measures d' and criterion. While d' quantifies the distance between the signal (e.g. go-trials) and noise distribution (e.g. stop-trials), criterion reflects a participant's propensity to choose yes or no. We observed that d' decreased (**Supplementary Figure 1a**; $P = 0.002$) and criterion increased (**Supplementary Figure 1b**; $P = 0.008$) with uncertainty. The shift in criterion indicates a shift along the internal response axis and can be interpreted as a change in participant's decision strategy (e.g. preparation of movement in the 0% likelihood condition vs. inhibition of movement in the 75% condition).

Supplementary Figure 1. Prior evidence modulates the decision strategy. **a**, The predictive cue significantly shifted d-prime, reflecting an increased separation between the internal signal and noise distribution. **b**, Participant's response bias (criterion) significantly increased during trials with high uncertainty, suggesting a more conservative response strategy.

2. To further substantiate our interpretation that predictive context altered participant's decision strategy, we applied a linear ballistic accumulator model¹⁶ in order to quantify whether predictive context led to a shift in starting point or the drift rate of the decision process. This revealed a significant change in the starting point (**Supplementary Figure 1c**; $P = 7.5 \times 10^{-7}$; Friedman test), but not the drift rate (**Supplementary Figure 1d**; $P = 0.229$; Friedman test). Specifically, the starting point was increased during trials with no uncertainty as compared to trials with medium or high uncertainty (**Supplementary Figure 1c**). This observation indicates that less information was needed to commit to a decision (execute vs. withhold response).

Supplementary Figure 1. Prior evidence modulates the decision strategy. **c**, The starting point was significantly reduced when participants received less prior evidence, suggesting that more evidence had to

be acquired to make a response. **d**, Schematized outcome of the linear ballistic accumulator model. Prior evidence shifted the starting point, but not the drift rate of the evidence accumulation process.

Collectively, this set of results supports our claim of context-dependent computations in the task that cannot be fully explained by task difficulty. We added the analyses and results to the Supplementary Material (**Supplementary Figure 1**) and refer to them in the results and methods section:

Page 6, Line 7-13

*“We observed that sensitivity d' decreased (**Supplementary Fig. 1**; $P = 0.002$, Cohen’s $d = 1.03$; Wilcoxon rank sum test) and criterion c increased (**Supplementary Fig. 1**; $P = 0.008$, Cohen’s $d = -0.7$) with uncertainty, indicating a more conservative response strategy as stop trials became more likely. Furthermore, linear ballistic accumulator modeling (Methods) revealed that prior evidence caused a shift in the starting point (**Supplementary Fig. 1**; $P = 7.5 \times 10^{-7}$; Friedman test), but not the drift rate ($P = 0.229$) of the decision process.”*

Page 6, Line 18-22

“In sum, these results demonstrate that states of high behavioral uncertainty are detrimental for the speed, accuracy, and sensitivity of action-linked decisions. Furthermore, they demonstrate that participants altered their response strategy, thereby providing evidence that they used the predictive cue to guide their decisions.”

Page 37, Line 1-6

“We also considered the signal detection theoretic measures d' (d -prime) and c (criterion)¹⁷. While d' quantifies the distance between the signal (e.g. go trials) and noise distribution (e.g. stop trials), c reflects a participant’s propensity to choose yes or no (decision criterion). Due to the nature of the task (absence of noise distribution in the 0% condition), we were only able to quantify d' and c for conditions with a 25% or 75% likelihood of stopping.”

Page 37, Line 7-12

“Linear Ballistic Accumulator Model. For each subject, we fitted a linear ballistic accumulator model¹⁶ for the reaction time using a convolutional method to estimate the non-decision time¹⁸. We fixed the variance of the drift and the non-decision time at 0.5. We estimated the drift and offset, allowing them to differ between conditions. Model fit was performed in R v. 4.1.3 (R Core Team, 2016), using the DstarM package¹⁹.”

2. The authors conclude that PFC activity shifts from an oscillatory to a ramping mode with increased uncertainty (Page 2 L4-6), but is it true? Figure 2 seems to be the support of this claim, but Figure 4 shows only a slight decrease in oscillatory power across conditions. And Figure 6c shows no encoding of context along theta oscillation. So, isn’t it more like oscillation is always present, but just the ramping activity becomes stronger with increased uncertainty (hence task difficulty)?

Thank you for raising this important point that was emphasized by Reviewer #1, *comment #5* as well. We think that our original phrasing may have caused confusion and would like to apologize for the lack of clarity. We fully agree that the conclusion of a “*shift from oscillatory and continuous ramping*” is misleading and not substantiated by our data.

Instead, we aimed to suggest that there is a shift from a purely oscillatory towards a mixed “oscillatory-plus-ramping” processing regime as shown in **Figure 3a & 4c**. We interpret this as evidence that oscillations are ubiquitous and primarily synchronize the prefrontal-motor network for communication whereas the switch towards more ramping is driven by a shift in uncertainty. We carefully revised the manuscript accordingly in multiple instances.

Page 2, Line 4-7

“We demonstrate that increasing behavioral uncertainty introduces a shift from a purely oscillatory to a mixed mode processing regime with an additional ramping component present in the human prefrontal cortex.”

Page 28, Line 11-14

“In line with the active sensing framework, we show that (II) behavioral uncertainty introduces a shift from an energy-efficient oscillatory to an energy-costly processing mode with mixed oscillatory and ramping dynamics (Fig. 3/4).”

3. The authors claim that the ramping activity reflects evidence accumulation, but unfortunately I did not understand this part. What evidence is accumulated in this task? Why should more evidence be accumulated in uncertain conditions?

Thank you for this comment. We fully concur that a clear definition of what we refer to as evidence accumulation was missing. We used the term evidence accumulation as the task required participants to discriminate between two possible and competing actions (executing vs. withholding the response), a process that can be understood in terms of competitive accumulation evidence in favor of each choice alternative^{20,21}.

However, we agree that the claim of ramping activity reflecting evidence accumulation is not fully substantiated by our data. We have therefore removed it from the manuscript and rephrased the respective sentences accordingly, for example:

Page 5, Line 14

“Neural and behavioral signatures of context-dependent computations.”

Page 18, Line 16-17

“Having established that ramping dynamics reflect a mechanism of context-dependent computations, [...]”

Page 19, Line 1-3

“We extracted the most dominant, population-wide activity pattern using principal component analysis (PCA) to examine the latent dynamics underlying context-dependent computations [...]”

4. The motor cortex appears to be showing ramping dynamics all the time (Figure 3b). How do the authors interpret this? The authors claimed that the dimension of theta oscillation is particularly relevant to action execution, but how about the dimension of ramping activity? I feel like this is more relevant to action execution (like the representation of hazard rate; Janssen & Shadlen

2005). Then, why specifically focus on theta oscillation as encoding of action plans?

- a) As correctly remarked, we demonstrated that motor cortex shows clear ramping dynamics that were not modulated by predictive context (**Figure 3b/5d**). We argue that this likely reflects a non-specific and context-independent growing urgency signal^{22,23} that is likely driven by the necessity of rapid motor decisions in our task. Indeed, multiple single-unit studies across rodents and non-human primates have extensively documented a gradual ramp in neural firing across motor areas that peaks just prior to the volitional commitment to an action choice²³⁻²⁶.

We now discuss this finding in the discussion:

Page 30, Line 15-20

“Critically, we did not observe a context-dependent modulation of ramping activity in motor cortex. Instead, ramping dynamics in motor cortex were largely preserved across all trials, likely reflecting a non-specific and context-independent growing urgency signal^{22,23} driven by the necessity of rapid motor decisions in this task. This leads to suggest that ramping dynamics subserve anatomically specialized functions.”

- b) We focused on theta oscillations as a candidate mechanism for the encoding and transfer of action plans between prefrontal and motor cortex for two specific reasons:

1. In our univariate analyses, we have shown that theta oscillations temporally structure the HFA in both PFC (**Figure 4a-f**) and motor cortex (**Supplementary Figure 4**). Critically, we observed that theta oscillations were not modulated by predictive context (**Figure 4c, Supplementary Figure 4**). Therefore, we reasoned that theta oscillations might be a candidate mechanism for the inter-areal interactions. This was further inspired by the communication-through-coherence framework²⁷ and a seminal study on theta-oscillatory interactions between the prefrontal-motor network during cognitive control⁵. Finally, using multivariate analyses, we demonstrated that neural activity in both PFC and motor cortex carried information about the timing of actions (**Figure 5g/h**). Consequently, we aimed to test whether theta oscillations further temporally synchronize prefrontal and motor cortex for the transfer of action plans (**Figure 6d**). As a prerequisite, we first needed to establish that it is possible to decode the timing of action from the subspace with the strongest theta oscillations in both PFC and motor cortex (**Figure 6c**). We added an additional paragraph to further explain the rationale as to why we focused on theta oscillations:

Page 25, Line 26 – Page 26, Line 4

“In a final analysis step, we characterized how population dynamics interact with neural oscillations to support goal-directed behavior. We specifically focused on theta oscillations as a potential candidate mechanism for the temporal synchronization and transfer of action plans. This step was chosen in a data-and theory-driven way. Since we observed that theta oscillations were not modulated by context in neither PFC nor motor cortex, but rhythmically structure neural

activity in both regions (cf. **Fig. 4c**; **Supplementary Fig. 4**), we reasoned that they might synchronize the prefrontal-motor network for inter-areal communication (cf. **Fig. 4g**). These data-driven findings were further supported by the communication-through-coherence framework²⁷ and a seminal study on theta-oscillatory, prefrontal-motor interactions during cognitive control⁵. Consequently, we aimed to test whether theta oscillations temporally synchronize low-dimensional subspaces between PFC and motor cortex for the transfer of action plans from PFC to motor cortex.”

2. We agree that ramping dynamics also reflect a viable mechanism for the preparation of action dynamics (e.g. reflecting an internal representation of the distribution of go-times in LIP as shown by Janssen & Shadlen, 2005). Moreover, we do not intend to claim that theta oscillations are the only mechanism related to the encoding and transfer of action plans. However, in our present study, theta oscillations were the only feature that was ubiquitous across all task conditions (e.g. as compared to ramping dynamics, cf. **Figure 3a**), rendering theta synchrony a key candidate mechanism for inter-areal subspace synchronization. We added a paragraph to discussion section where we emphasize that there might be multiple mechanisms coding for action dynamics:

Page 30, Line 5-15

“[...] While we focused on theta oscillations as a candidate mechanism for the transfer of motor plans via inter-areal synchronization of action subspaces, other mechanisms related to the temporal control of action have been demonstrated, such as single neuron ramping activity in the lateral intraparietal area²⁸, medial PFC²⁹, or in frontal-striatal circuits³⁰. In these studies, ramping dynamics mirrored the temporal integration of time (e.g. by representing the hazard rate of reward probabilities²⁸). Yet, in our study, ramping dynamics in PFC strongly dissociated between distinct states of uncertainty while the temporal dynamics in the task were kept constant across trials. This suggests that ramping dynamics in the human PFC encode latent variables in addition to timing³¹. Critically, we did not observe a context-dependent modulation of ramping activity in motor cortex. Instead, ramping dynamics in motor cortex were largely preserved across all trials, likely reflecting a non-specific and context-independent growing urgency signal^{22,23} driven by the necessity of rapid motor decisions in this task. This leads to suggest that ramping dynamics subserve anatomically specialized functions.”

3. We also feel that some confusion on why we focused on theta oscillations as the encoding of action plans may have arisen because we did not make sufficiently clear what we refer to as *action-encoding*. Thus, we carefully revised the manuscript and used more specific wording to provide a clearer definition of *action-encoding*.

Page 20, Line 3-16

*“Next, we specifically assessed which latent dimension reflects contextual encoding in human PFC, where population dynamics are high-dimensional. To determine the relevant coding dimensions, we employed a Linear Discriminant Analysis (LDA) classifier in PC space, separately for both regions. This approach defined the coding dimensions that maximally discriminated context and reaction times (split into terciles; referred to as *action-encoding*; Methods). This approach dissociates dynamics that are relevant for contextual processing and subsequent*

action planning. We time-locked the population response to the timing of movement execution. Thus, any neural activity that dissociates reaction times prior to the actual movement execution (and that is not primarily driven by early context-integration signals) reflects neural dynamics related to the planning of movement. To separate these two processes, we constrained the coding dimensions of context and action to be orthogonal in neural state space (Methods).”

5. Does the motor cortex encode context (uncertainty) or not? I think the paper is giving mixed results. In earlier figures (Figure 1d), the motor cortex is shown to have context information as strongly as PFC. But later, all the analyses deny the presence of context dependency in the motor cortex. How could this happen? How does this relate to the conclusions?

Thank you for addressing this critical point. We fully acknowledge that this might have been confusing and that the previous version of the manuscript was lacking a proper explanation.

1. As correctly remarked, **Figure 1d** suggests that motor cortex carries context-dependent information as strongly as PFC. This conclusion was derived from univariate analyses based on sliding-window ANOVAs used to extract the percent variance explained in the HFA signal by the factor *context*. However, per definition ω^2 is an unsigned estimate of neural information and does not indicate the direction of the association (positive or negative).

In order to illustrate this issue, we have visualized three example electrodes in motor cortex (**Supplementary Figure 3**) that all exhibit significant context-dependent neural information (top row). It is apparent that the direction varies between electrodes (bottom row). This is not a caveat of this specific analysis, but an important feature that enables extracting encoding electrodes without imposing any bias with respect to the direction of the effect.

We would like to emphasize that we employed this analysis in order to identify context-encoding electrodes to maximize the sensitivity for all *univariate* analyses (without imposing any bias). This is a common procedure in the field of intracranial as well as single-unit recordings^{2,5,9,32}.

We fully acknowledge that this is highly confusing for the reader and misleading in terms of interpretation. We therefore decided to add a paragraph in the results to clarify the complementary nature of univariate and multivariate analysis approaches.

Supplementary Figure 3. Single electrode examples of percent explained variance and underlying high-frequency activity traces. **a-c**, Single electrode examples from motor cortex showing significant context-dependent information (top row). Note however, that the direction of the effect differs across electrodes (bottom row), e.g. while the explained variance in panel **a** is driven by a gradual change in HFA from low to high states of uncertainty (green = low uncertainty; orange = medium uncertainty; red = high uncertainty), this is not visible in single electrode in panel **b** and **c**. Yet, these directional differences on the effect cannot be dissociated solely based on the percent explained variance (PEV) as this metric is blind with respect to the direction of the effect. **d**, *Top*: PEV averaged across examples shown in (**a-c**). Note that the effects from **b** and **c** are largely cancelled out on the grand average because the effect occurred at distinct timepoints. Yet, on average, variability in neural activity still explains up to 4.48% variance of the task-feature, although individual electrodes strongly differed in terms their selectivity. *Bottom*: shows the averaged neural response across all three example electrodes per condition. Because of the variability between the electrodes, averaging them largely cancels out the effects.

Page 8, Line 15-23

“Note that explained variance is an unsigned estimate of neural information and does not indicate the direction of the association (positive or negative). Hence, no inference on the sign of context-dependent effects can be drawn based on this analysis. While this approach allows for the extraction of context-encoding electrodes, it does neither impose any bias nor provide any information with respect to the direction of the effect. Thus, any randomly distributed effect across time and/or conditions would result in an inconsistent context-dependent modulation in the grand average HFA traces (Supplementary Fig. 3; Methods).”

2. Another explanation why the univariate analyses (cf. **Figure 1-4**) do not necessarily fully converge with the multivariate analyses (cf. **Figure 5-6**) is the use of a different set of electrodes. While we used only context-encoding electrodes for all univariate analyses in order to maximize sensitivity, we used all available electrodes (*entirely sampled population; context-encoding + non-encoding electrodes*) for all multivariate analyses in order to identify the population correlates of continuous and oscillatory dynamics as well as the low-dimensional representations of context and action.

This distinction is very important as the population response is more than the sum of individually tuned neurons^{33,34}. Neurons that are highly tuned to the latent variable of interest (e.g. context or action) contribute to the population response and define the coding dimensions, but so do untuned neurons as well³³. Thus, the functional interaction between differently tuned neural populations is equally important for information coding³⁴.

However, context-dependent effects of tuned electrodes in motor cortex were already small as compared to PFC (cf. **Figure 2a/b**). Therefore, it is likely that the majority of neural populations in motor cortex did not perform context-dependent computations, suggesting that context coding cannot be inferred from the overall population response in motor cortex.

6. The authors claim significant encoding of action based on significant classification accuracy of action; but here, the classifier is trained to discriminate trials with different reaction times. First, I am not entirely clear if it could be said as encoding of action. The plots of classifier accuracy are aligned to the timing of button press, so the same action is happening in every trial regardless of reaction times. Then it is not classifying different actions. Second, reaction times are highly correlated with uncertainty (Figure. 1C), so I am afraid that the classification performance is easily confounded with the encoding of

context (or task difficulty) or vice versa. How do the authors justify that this is not the case?

Thank you for the opportunity to clarify. We agree that a significant decoding of response time does not provide information about the type of action being performed. As outlined above (*Reviewer #2, comment #4*), we did not sufficiently define *action-encoding*. Here, we first clarify our terminology to better contextualize these assertions. Subsequently, we performed several control analyses that demonstrate the context and action can successfully be dissociated.

- a) We intentionally time-locked the neural data to the button release in order to classify preparatory activity prior to the subsequent action, not different types of actions. This procedure allowed us to dissociate the initial integration of context from subsequent action planning (cf. **Figure 5g/h**). We now clarify the use of our terminology more explicitly in the results section:

Page 20, Line 3-16

“Next, we specifically assessed which latent dimension reflects contextual encoding in human PFC, where population dynamics are high-dimensional. To determine the relevant coding dimensions, we employed a Linear Discriminant Analysis (LDA) classifier in PC space, separately for both regions. This approach defined the coding dimensions that maximally discriminated context and reaction times (split into terciles; referred to as action-encoding; Methods). This approach dissociates dynamics that are relevant for contextual processing and subsequent action planning. We time-locked the population response to the timing of movement execution. Thus, any neural activity that dissociates reaction times prior to the actual movement execution (and that is not primarily driven by early context-integration signals) reflects neural dynamics related to the planning of movement. To separate these two processes, we constrained the coding dimensions of context and action to be orthogonal in neural state space (Methods).”

- b) While it is absolutely correct that reaction times and predictive context are highly correlated (cf. **Figure 1c**), it is unlikely that the decoding traces of context and action capture the same processes for three main reasons:
1. As described in methods section (*“Identification of coding dimensions”*), we constrained the PC dimensions coding for context and action to be orthogonal (mapping onto different PCs; orthonormal basis) in order to minimize the confound that we assess the same neural mechanism. Orthogonal encoding was present in 11/14 participants without adding any additional constraints; thus, suggesting that context integration and action planning are dissociable processes.
 2. The temporal evolution of context and action coding follows distinct patterns (cf. **Figure 5g**). While we observed an early decoding peak for context, action decoding gradually built-up over time, likely reflecting action preparatory activity. Thus, it is unlikely that these traces map the same neural processes.
 3. Finally, if they would capture the same processes on a neural level, context as well as action should be decodable from both PFC and motor cortex. However, since we were only able to decode action, but not context from

motor cortex, we would argue that it is rather unlikely that the decoding traces reflect similar underlying neural processes.

In sum, this set of consideration implies that we capture different neural processes – context integration and action planning. Yet, we fully acknowledge that these processes are not completely independent on a behavioral level, but likely exhibit distinct neural underpinnings. To disentangle their relative contributions in our task, we also performed several additional analyses:

1. We tested whether we can successfully decode context based on the action-dimension (**Supplementary Figure 5a**). We did not find evidence for context-coding within the action subspace (no cluster at $p < 0.05$), strongly suggesting that these processes are dissociable on a neural level.

Supplementary Figure 5. Subspaces maximally discriminating context and action capture dissociable processes. **a**, Context could not be successfully decoded in neither PFC (left) nor motor cortex (right) based on the dimension maximally discriminating reaction times (no cluster at $p < 0.05$). This suggests that the dimensions maximally dissociating context and action reflect dissociable processes.

2. We also tested whether we can successfully decode action based on the dimension maximally discriminating the type of context. This analysis revealed that the context-dimension in PFC contains some information about reaction times (**Supplementary Figure 5b**; $\text{sum}(t_{13}) = 201.28$, $P = 0.028$, Cohen's $d = 0.77$), but this information was only accessible after the response (0.15s. to 0.29s. after button release) and therefore, likely reflected response monitoring³⁵⁻³⁸.

Supplementary Figure 5. Subspaces maximally discriminating context and action capture dissociable processes. **b**, Action-decoding based on the context-dimension was still possible in PFC ($\text{sum}(t) = 201.28$, $P = 0.028$, Cohen's $d = 0.77$), but not in motor cortex ($\text{sum}(t) = 121.5$, $P = 0.055$). Note, however, that significant action-decoding was only possible after the action has already been executed, possibly reflecting action-outcome monitoring--.

Collectively, this set of findings suggests that context integration and action planning rely on distinct neural mechanisms.

We modified the manuscript and supporting information in multiple instances to highlight these considerations:

Page 21, Line 6-8

*“Critically, we performed additional control analyses demonstrating that these two subspaces indeed capture dissociable processes related to contextual computations and the planning of actions (**Supplementary Fig. 5**; Methods).”*

Page 49, Line 17-22

*“Decoding control analyses. We performed additional control analyses to ensure that our two identified subspaces (context and action) capture dissociable processes (**Supplementary Fig. 5**). We tested whether decoding performance was still above chance level when the action dimension was used to predict context and vice versa. Non-significant classification performance would imply that these two subspaces capture distinct processes.”*

7. The authors claimed the encoding of context and action is orthogonal in state space (Page 19 L13-), but that’s a strong claim and unfortunately, I couldn’t fully understand how the authors reached this conclusion. Is it based on the finding that the PCs best decoding context and action were different? This could mean that the two axes are not aligned but does not necessarily mean they are orthogonal. To test this, the authors should compute the angle between the two decoding axes and show that it is indistinguishable from 90 degrees in state space (through a permutation test or some sort). If the authors just wanted to claim these two axes are not aligned (but not necessarily orthogonal), then the test would be whether the angle is significantly greater than 0 deg or not.

Thank you for your suggestions. We based our claim (encoding axes of context and action are orthogonal) on the fact that the principal components maximally

discriminating context and action were different in 11/14 participants (without adding additional constraints; PCA assumes orthogonality).

As suggested, we also tested whether the PCs are indistinguishable from 90 degrees. We therefore computed the subspace angle between the principal eigenvectors encoding context and action and show that the angle between these two decoding axes is indistinguishable from 90 degrees (**Response Figure 4**). As outlined above, this was expected as the eigenvectors of the covariance matrix form an orthonormal basis set.

We realized that it is more adequate to highlight that the coding axes for action and context map onto distinct PCs and *omit the claim on orthogonality*. The claim of orthogonality is not critical to convey the main message in the present manuscript and we therefore rephrased this claim where applicable, for example:

Page 2, Line 9-10

*“Prefrontal population activity encodes predictive context and action plans in **distinct** and serially unfolding subspaces [...].”*

Page 20, Line 21 – Page 21, Line 1

“Importantly, we found that the coding dimensions maximally discriminating context and action planning mapped onto distinct PCs in 11/14 participants ($P = 0.057$; Binomial test), suggesting that these processes are dissociable in the neural state space.”

Page 23, Line 3-4

*“Fig. 5. Low-dimensional representation of context and action in **distinct** coding dimensions”*

Page 28, Line 17-19

*“Specifically, we show that (IV) prefrontal population activity encodes predictive context and action plans in **distinct** and serially unfolding subspaces [...].”*

Response Figure 4. Subspace angle between context and action PCs are indistinguishable from 90 degrees. We computed the subspace angle between the PC eigenvectors encoding context and action. Since

the eigenvectors form an orthonormal basis set, they are by default orthogonal (angle equal to 90 degrees) to each other. The minimal difference in the right panel is caused by computer rounding errors.

8. Page 22, L12-: the authors stated that “the dimension with the strongest theta power most likely matched PC1”, but I was confused. Isn’t PC1 supposed to carry context information (Figure 5c)? But here, the authors claim there is no context information along this axis (Figure 6c).

Thank you for bringing this issue into our attention. This is indeed a confusing statement, which did not convey relevant information. The proportion of participants in which the theta dimension matched PC1 was actually chance ($P = 0.332$; binomial test). Thus, this sentence was indeed misleading and was therefore removed from the manuscript.

Minor

1. Page 5, L19-: It would be great if the authors could briefly mention the definition of RTs and accuracy around here. I was initially confused what is the definition of accuracy in this task (and had to check methods).

Thank you for this excellent suggestion. We now added a brief definition of reaction times and accuracies.

Page 5, Line 14 – Page 6, Line 1

“We confirmed that participants used the predictive cue to guide behavior using reaction time, accuracy and signal detection theory. Here, reaction time was quantified as the time interval between the moving target reaching the predefined lower limit and the participants’ response. Accuracy was defined as the percentage of correct responses relative to the number of trials. Trials in which participants released the button within the time interval between the lower and upper limit (Fig. 1a) were considered as correct trials whereas trials in which they released the button either before the lower limit or after the upper limit were considered as incorrect.”

2. All figures: all legends and axis labels are a bit too small and hard to read.

We fully agree that the axis labels in the figures are hard to read. We therefore modified all figures to improve readability.

3. Figure 1e: Significant and non-significant sites are drawn using large and small spheres, but it was also a bit difficult to distinguish them. Perhaps use different colors?

Thanks for the suggestion. We fully concur and changed the figure accordingly. We now use white spheres for non-encoding electrodes in both PFC and motor cortex. We hope that it is now easier to distinguish encoding from non-encoding electrodes.

e Context-encoding and non-encoding electrodes

4. Figure 2d: it looks like motor cortex activity is also shifting from oscillatory to ramping state. But this is not what the authors wanted to say? Maybe this was not the best example?

Thank you for this suggestion. These examples are indeed misleading and do not support the group level analyses. We changed the single-trial examples in motor cortex accordingly to be in line with the group level analysis (cf. Figure 3b).

5. Page 31, L8-: in uncertain conditions, when could the target stop? Uniform probability over all the time, or flat hazard rate?

In conditions with either a 25% or 75% likelihood of stop, the stop timing of the target was normally distributed before the predefined lower limit (upon which participants were allowed to release the button to achieve a correct trial; **Supplementary Figure 6**). The target could only stop up to 150ms before reaching the lower limit which reinforced participants to integrate the prior evidence to guide their decision as they were required to make fast decisions. We have added this information into the methods section and added the figure into the supplementary materials.

Page 36, Line 1-2

“The timing of a premature stop was normally distributed prior to the HLL (Supplementary Fig. 6).”

Supplementary Figure 6. Timing of stop trials in uncertain conditions. The probability of stop trials over time is depicted relative to the lower limit (HLL). The short time window between the probability of a target stopping and the HLL reinforced participants to integrate the predictive cue to guide their motor decisions.

References

- 1 Buschman, T. J., Siegel, M., Roy, J. E. & Miller, E. K. Neural substrates of cognitive capacity limitations. *Proc Natl Acad Sci U S A* **108**, 11252-11255, doi:10.1073/pnas.1104666108 (2011).
- 2 Siegel, M., Buschman, T. J. & Miller, E. K. Cortical information flow during flexible sensorimotor decisions. *Science* **348**, 1352-1355, doi:10.1126/science.aab0551 (2015).
- 3 Frohlich, F. & McCormick, D. A. Endogenous electric fields may guide neocortical network activity. *Neuron* **67**, 129-143, doi:10.1016/j.neuron.2010.06.005 (2010).
- 4 Canolty, R. T. *et al.* Oscillatory phase coupling coordinates anatomically dispersed functional cell assemblies. *Proc Natl Acad Sci U S A* **107**, 17356-17361, doi:10.1073/pnas.1008306107 (2010).
- 5 Voytek, B. *et al.* Oscillatory dynamics coordinating human frontal networks in support of goal maintenance. *Nat Neurosci* **18**, 1318-1324, doi:10.1038/nn.4071 (2015).
- 6 Kam, J. W. Y. *et al.* Top-Down Attentional Modulation in Human Frontal Cortex: Differential Engagement during External and Internal Attention. *Cereb Cortex* **31**, 873-883, doi:10.1093/cercor/bhaa262 (2021).
- 7 Helfrich, R. F. *et al.* Neural Mechanisms of Sustained Attention Are Rhythmic. *Neuron* **99**, 854-865 e855, doi:10.1016/j.neuron.2018.07.032 (2018).
- 8 Haller, M. *et al.* Persistent neuronal activity in human prefrontal cortex links perception and action. *Nat Hum Behav* **2**, 80-91, doi:10.1038/s41562-017-0267-2 (2018).
- 9 Durschmid, S. *et al.* Hierarchy of prediction errors for auditory events in human temporal and frontal cortex. *Proc Natl Acad Sci U S A* **113**, 6755-6760, doi:10.1073/pnas.1525030113 (2016).
- 10 Durschmid, S. *et al.* Direct Evidence for Prediction Signals in Frontal Cortex Independent of Prediction Error. *Cereb Cortex* **29**, 4530-4538, doi:10.1093/cercor/bhy331 (2019).
- 11 Yamada, H., Imaizumi, Y. & Matsumoto, M. Neural Population Dynamics Underlying Expected Value Computation. *J Neurosci* **41**, 1684-1698, doi:10.1523/JNEUROSCI.1987-20.2020 (2021).
- 12 Mante, V., Sussillo, D., Shenoy, K. V. & Newsome, W. T. Context-dependent computation by recurrent dynamics in prefrontal cortex. *Nature* **503**, 78-84, doi:10.1038/nature12742 (2013).
- 13 Zheng, J. *et al.* Neurons detect cognitive boundaries to structure episodic memories in humans. *Nat Neurosci* **25**, 358-368, doi:10.1038/s41593-022-01020-w (2022).
- 14 Gallego, J. A. *et al.* Cortical population activity within a preserved neural manifold underlies multiple motor behaviors. *Nat Commun* **9**, 4233, doi:10.1038/s41467-018-06560-z (2018).
- 15 Cohen, M. X. Analyzing Neural Time Series Data: Theory and Practice. *Iss Clin Cogn Neurop*, 1-578 (2014).
- 16 Brown, S. D. & Heathcote, A. The simplest complete model of choice response time: linear ballistic accumulation. *Cogn Psychol* **57**, 153-178, doi:10.1016/j.cogpsych.2007.12.002 (2008).
- 17 Stanislaw, H. & Todorov, N. Calculation of signal detection theory measures. *Behav Res Methods Instrum Comput* **31**, 137-149, doi:10.3758/bf03207704 (1999).

- 18 Verdonck, S. & Tuerlinckx, F. Factoring out nondecision time in choice reaction time data: Theory and implications. *Psychol Rev* **123**, 208-218, doi:10.1037/rev0000019 (2016).
- 19 van den Bergh, D., Tuerlinckx, F. & Verdonck, S. DstarM: an R package for analyzing two-choice reaction time data with the D *M method. *Behav Res Methods* **52**, 521-543, doi:10.3758/s13428-019-01249-7 (2020).
- 20 Gold, J. I. & Shadlen, M. N. The neural basis of decision making. *Annu Rev Neurosci* **30**, 535-574, doi:10.1146/annurev.neuro.29.051605.113038 (2007).
- 21 Murphy, P. R., Robertson, I. H., Harty, S. & O'Connell, R. G. Neural evidence accumulation persists after choice to inform metacognitive judgments. *Elife* **4**, doi:10.7554/eLife.11946 (2015).
- 22 Cisek, P., Puskas, G. A. & El-Murr, S. Decisions in changing conditions: the urgency-gating model. *J Neurosci* **29**, 11560-11571, doi:10.1523/JNEUROSCI.1844-09.2009 (2009).
- 23 Thura, D. & Cisek, P. Deliberation and commitment in the premotor and primary motor cortex during dynamic decision making. *Neuron* **81**, 1401-1416, doi:10.1016/j.neuron.2014.01.031 (2014).
- 24 Hanes, D. P. & Schall, J. D. Neural control of voluntary movement initiation. *Science* **274**, 427-430, doi:10.1126/science.274.5286.427 (1996).
- 25 Murakami, M., Vicente, M. I., Costa, G. M. & Mainen, Z. F. Neural antecedents of self-initiated actions in secondary motor cortex. *Nat Neurosci* **17**, 1574-1582, doi:10.1038/nn.3826 (2014).
- 26 Thura, D. & Cisek, P. Modulation of Premotor and Primary Motor Cortical Activity during Volitional Adjustments of Speed-Accuracy Trade-Offs. *J Neurosci* **36**, 938-956, doi:10.1523/JNEUROSCI.2230-15.2016 (2016).
- 27 Fries, P. Rhythms for Cognition: Communication through Coherence. *Neuron* **88**, 220-235, doi:10.1016/j.neuron.2015.09.034 (2015).
- 28 Janssen, P. & Shadlen, M. N. A representation of the hazard rate of elapsed time in macaque area LIP. *Nat Neurosci* **8**, 234-241, doi:10.1038/nn1386 (2005).
- 29 Kim, J., Ghim, J. W., Lee, J. H. & Jung, M. W. Neural correlates of interval timing in rodent prefrontal cortex. *J Neurosci* **33**, 13834-13847, doi:10.1523/JNEUROSCI.1443-13.2013 (2013).
- 30 Emmons, E. B. *et al.* Rodent Medial Frontal Control of Temporal Processing in the Dorsomedial Striatum. *J Neurosci* **37**, 8718-8733, doi:10.1523/JNEUROSCI.1376-17.2017 (2017).
- 31 Narayanan, N. S. Ramping activity is a cortical mechanism of temporal control of action. *Curr Opin Behav Sci* **8**, 226-230, doi:10.1016/j.cobeha.2016.02.017 (2016).
- 32 Saez, I. *et al.* Encoding of Multiple Reward-Related Computations in Transient and Sustained High-Frequency Activity in Human OFC. *Curr Biol* **28**, 2889-2899 e2883, doi:10.1016/j.cub.2018.07.045 (2018).
- 33 Ebitz, R. B. & Hayden, B. Y. The population doctrine in cognitive neuroscience. *Neuron*, doi:10.1016/j.neuron.2021.07.011 (2021).
- 34 Panzeri, S., Moroni, M., Safaai, H. & Harvey, C. D. The structures and functions of correlations in neural population codes. *Nat Rev Neurosci*, doi:10.1038/s41583-022-00606-4 (2022).
- 35 Luk, C. H. & Wallis, J. D. Dynamic encoding of responses and outcomes by neurons in medial prefrontal cortex. *J Neurosci* **29**, 7526-7539, doi:10.1523/JNEUROSCI.0386-09.2009 (2009).

- 36 Alexander, W. H. & Brown, J. W. Medial prefrontal cortex as an action-outcome predictor. *Nat Neurosci* **14**, 1338-1344, doi:10.1038/nn.2921 (2011).
- 37 Spellman, T., Svej, M., Kaminsky, J., Manzano-Nieves, G. & Liston, C. Prefrontal deep projection neurons enable cognitive flexibility via persistent feedback monitoring. *Cell* **184**, 2750-2766 e2717, doi:10.1016/j.cell.2021.03.047 (2021).
- 38 Gehring, W. J. & Knight, R. T. Prefrontal-cingulate interactions in action monitoring. *Nat Neurosci* **3**, 516-520, doi:10.1038/74899 (2000).

Reviewers' Comments:

Reviewer #1:

Remarks to the Author:

The authors have done an excellent job in addressing my concerns. I have no further comments and congratulate the authors on a splendid paper.

Reviewer #2:

Remarks to the Author:

I would like to thank the authors for addressing all of my comments. I agree with their responses and have no further comments on the revised manuscript. This is an impressive work that thoroughly investigated the activity and communication of PFC and motor cortex in their task with iEEG data.

Regarding comment 7 (orthogonality of context and action encoding), what I suggested is to find axes in neural space that maximally discriminate context and action independently (without relating them to PCs, which are granted to be orthogonal) and then compute their angle in neural space. But I agree that orthogonality is not a necessary claim, and I am satisfied with the authors' revision.

Reviewer #3:

Remarks to the Author:

I think that the results from Figs 1-4 are solid, and the authors did an excellent job at addressing the reviewers' comments on these sections. However, Figs 5-6 have some serious methodological flaws that render them either hard to interpret, or just not valid.

Major Points

1- The LL appears to trigger a large activity change. Uncertain trials have longer RTs. So the trajectory analyses could be affected by this difference (as pointed out by Reviewer 2 and acknowledged by the authors). The issue with the action decoder is that it captures different periods of the task for different categories (pre-LL for fast RTs, post-LL for slow RTs and a mixture for the mid RTs). To carry out the intended analysis the authors would need to match the RTs for the different contexts before comparing the activities. Unfortunately, this observation renders the rest of the analyses using this action space uninterpretable.

2- Line 366: "This indicates that PFC, but not motor cortex, is gradually recruited as a function of behavioral uncertainty (Fig. 5e)." I don't think your results should be interpreted this way. When aligned to button release (as in Fig5), the differences begin emerging around 0.2s before movement, which happens to coincide with the LL (lower limit) onset for the slowest RT trials. If PFC was gradually recruited, these differences should start emerging from the moment that the contextual cue is given (560-750ms before the LL). But that is not the case. Go-cues (like the LL) trigger a condition-invariant signal, or CIS (Kaufman et al. 2016, Guo et al. 2014, Inagaki et al. 2022). In light of the CIS, averaging trials, aligned to movement onset, could lead to the exact figure shown. So this analysis should not be interpreted as a gradual increase.

3- LDA analysis: This is a very unconventional way to carry out this analysis. The decoding latent space could be a high-dimensional space that spans multiple PCs. So normally these analyses are done on all dimensions at once (or all dimensions that explain more than 95% of variance), and not one PC dimension at a time, as was done here. I think that this analysis, as carried out, is not very meaningful. If the goal was to enforce orthogonality, you can impose this constraint in the higher-dimensional LDA spaces (although I'm not sure why you'd want to impose orthogonality).

4- Fig 5i shows that most action peaks occur post-movement (BR), so these cannot be causal in

the movement generation nor planning. As such this "serial unfolding" claim does not really rest in solid grounds. This also motivates pg 21: "This raises the question, whether and how the representation of the action plan in PFC is passed on to motor cortex for final action execution?", which loses force when we realize the post-movement nature of these sequential signals.

5- The methods description of the cross-regional pattern analysis is not clear from the description of the methods and results section. I'm not sure how you can test classification using different sets of electrodes that were not used in the training of the LDA. As such, I cannot assess fig 5j and the "information handoff" claim.

Minor Points

1- Fig g and h right are not very useful without stats... the axis spans seem to be chosen to make a particular point rather than to reflect the data's structure.

2- Pg 32: "Here, we replicate this finding in humans, but in contrast to NHPs, we found no evidence that the human motor cortex encodes predictive context." This is not quite true, since the NHP work referenced provides evidence of pre-movement prediction regarding the specific movement (not context). This study only had one movement (button release) so this could not be assessed.

3- The correlation analysis in Figure 5e can also be confounded by the fact that high-uncertainty trials had longer RTs... the differences in correlation values could be a simple reflection of a larger number of points being included in the correlations of longer-RT trials.

4- PCA methods: There is not enough information in the methods to understand what was done with the PCA analysis. Were the time bins aligned to movement, or to task events? Were all trials (across all contexts) used to build the space, or were spaces built on context-specific activity? Which time bins were included in the PCA analysis?

5- Answers to reviewers mostly address their issues, except for Reviewer 2, point 1. All those analyses are also consistent with difficulty being the main factor. At the very least this should be discussed as a confound in the paper, rather than brushing it off.

Reviewers' Comments:

Reviewer #3:

Remarks to the Author:

Point-by-Point Reply: NCOMMS-22-05636

We would like to thank the reviewers for their thoughtful and constructive comments. Below, we provide point-by-point responses to all queries. All changes in the revised manuscript are highlighted with tracked changes.

Reply formatting guide:

Reviewer remarks: **bold, black.**

Author response: *blue.*

Changes in the manuscript: *italics, blue.*

Reviewer #1 (Remarks to the Authors)

The authors have done an excellent job in addressing my concerns. I have no further comments and congratulate the authors on a splendid paper.

We would like to thank the reviewer for the constructive review that helped to improve the manuscript.

Reviewer #2 (Remarks to the Authors)

I would like to thank the authors for addressing all of my comments. I agree with their responses and have no further comments on the revised manuscript. This is an impressive work that thoroughly investigated the activity and communication of PFC and motor cortex in their task with iEEG data.

Regarding comment 7 (orthogonality of context and action encoding), what I suggested is to find axes in neural space that maximally discriminate context and action independently (without relating them to PCs, which are granted to be orthogonal) and then compute their angle in neural space. But I agree that orthogonality is not a necessary claim, and I am satisfied with the authors' revision.

We would like to thank the reviewer for the valuable and insightful feedback that substantially improved the manuscript.

Reviewer #3 (Remarks to the Authors)

I think that the results from Figs 1-4 are solid, and the authors did an excellent job at addressing the reviewers' comments on these sections. However, Figs 5-6 have some serious methodological flaws that render them either hard to interpret, or just not valid.

We would like to thank the reviewer for the insightful and constructive comments on our manuscript and the detailed queries regarding figures 5 & 6. The key criticisms are a direct result of either imprecise wording or an insufficient explanation of the rationale and methods. We now provide a more detailed explanation of the analytical approach and the employed methods. In addition, we conducted several additional control analyses, which further substantiated our findings.

Major Points

1- The LL appears to trigger a large activity change. Uncertain trials have longer RTs. So the trajectory analyses could be affected by this difference (as pointed out by Reviewer 2 and acknowledged by the authors). The issue with the action decoder is that it captures different periods of the task for different categories (pre-LL for fast RTs, post-LL for slow RTs and a mixture for the mid RTs). To carry out the intended analysis the authors would need to match the RTs for the different contexts before comparing the activities. Unfortunately, this observation renders the rest of the analyses using this action space uninterpretable.

We would like to thank the reviewer for bringing up this point. We would like to divide our responses to query 1 into two sections to separately address the query on (1) large activity changes upon the lower limit (LL or HLL) and (2) the concern regarding the action decoder.

"The LL appears to trigger a large activity change. Uncertain trials have longer RTs. So the trajectory analyses could be affected by this difference (as pointed out by Reviewer 2 and acknowledged by the authors)."

We agree that a go-signal can often lead to large neural activity changes in motor tasks (see also point #2 below, i.e. Kaufman et al.¹). As outlined below, we first clarified the experimental design and previous analyses and then conducted additional analyses, which collectively demonstrate that the large activity change is not the result of reaching the LL.

1. Experimental design: The lower limit is present throughout every trial. The limit never changes its position, nor does it suddenly appear within each trial. Thus, participants could see the horizontal bar already at trial onset and therefore, prepare their motor response. The early motor preparation was particularly evident in trials with no uncertainty (0% likelihood of a stop). In this condition, we did not observe a strong change in neural activity in human PFC (**Figure 2a**; **Figure 3a**; **Figure 5c**). In order to better understand whether there might be a strong phase-locked response, we computed event-related potentials for the lower limit. As displayed in **Response Figure 1**, we did not observe a strong change in neural activity upon reaching the LL. This finding demonstrates that the lower limit does not trigger a large activity change.

Response Figure 1. Grand-average event-related potentials relative to the HLL. We computed the average ERP (mean \pm SEM) across participants time-locked to the lower limit. Note that there is no phase-locked change in activity when the target reaches the HLL, precluding that the HLL triggers a consistent and large change in neural activity.

2. Single-trial dynamics in uncertain trials (25% or 75% likelihood) exhibit a ramping pattern prior to the lower limit (**Figure 2c/d**; see below). This observation demonstrates that the observed activity patterns were not triggered by the LL, but reflected intrinsic, context-dependent dynamics that were already present prior to the LL.

3. We also time-locked the population response (PC₁) in PFC and motor cortex to the LL to supplement our findings relative to the button release (**Supplementary Figure 5**). As visualized in **Supplementary Figure 5**, the population activity in PFC already ramps-up and dissociates the different conditions prior to the LL. This observation is also in line with our findings on univariate ramping dynamics (**Figure 2a/b**): No significant ramping was observed in PFC during trials with no uncertainty (0% likelihood), while significant ramping activity was present for trials with a 25% or 75% likelihood of stopping.

Supplementary Figure 5. Context-dependent population dynamics in PFC emerge prior to the HLL. **a**, Population activity in PFC as indexed by the first principal component reveals context-specific activation patterns prior to the onset of the HLL ($\text{sum}(F_{2,32}) = 637.4$, $P = 0.023$; cluster test). The single-colored horizontal lines show the temporal extent of a significant context-dependent dissociation. Two-colored horizontal lines indicate the temporal extent of significant clusters obtained from pairwise comparisons. **b**, Same as **(a)**, but for motor cortex. Note the similar, context-independent, temporal activation profile across all conditions in motor cortex ($\text{sum}(F_{2,26}) = 251.7$, $P = 0.073$).

In sum, this set of findings demonstrates that the LL did not trigger a large activity change. On the contrary, these additional analyses demonstrate that the activity change reflects the context integration process that started prior to reaching the LL.

We have added **Supplementary Figure 5** to the supplements and refer to it in the main text:

Page 19, Line 15-18

"We additionally time-locked the population activity to the HLL to ensure that context-dependent activity in PFC ramped up and was dissociable prior to the HLL and not a consequence of an activation change triggered by the HLL (Supplementary Fig. 5)."

Next, we addressed the second concern that was voiced in the first query.

"The issue with the action decoder is that it captures different periods of the task for different categories (pre-LL for fast RTs, post-LL for slow RTs and a mixture for the mid RTs). To carry out the intended analysis the authors would need to match the RTs for the different contexts before comparing the activities. Unfortunately, this observation renders the rest of the analyses using this action space uninterpretable."

We performed the analysis as suggested by the reviewer as well as another additional control analysis to preclude that the action decoder is confounded by capturing different periods of the task. We will separately outline the two analyses:

1. We agree that the action decoder captures different periods of the task for different categories. However, the lower limit did not trigger large activity changes as outlined above. Therefore, it is unlikely that the action decoder is confounded by capturing different periods of the task. However, in order to exclude this possibility, we time-locked the data to the LL (which serves as the 'go' cue) and trained a linear decoder to discriminate reaction times (action). Above-chance decoding prior to the LL indicates that the decoder captures processes

involved in the planning of action. As shown in **Supplementary Figure 6**, a linear decoder can successfully discriminate the reaction times prior to the lower limit. This finding indicates that action plans were initiated prior to reaching the lower limit.

Supplementary Figure 6. Action-specific information emerges prior to the HLL. **a**, Grand average decoding accuracy (mean \pm SEM) for action within PFC when locked to the HLL. Horizontal lines indicate the extent of significant temporal clusters. **b**, same as **(a)** but for motor cortex. Same conventions as in **(a)**.

We have added **Supplementary Figure 6** to the supplements. We have further added an additional paragraph of this control analysis for clarification:

Page 21, Line 9-19

*"We performed an additional control analysis to preclude that the action-decoding was confounded by capturing different periods of the task (i.e., pre-HLL for fast RTs, post-HLL for slow RTs) when locked to the BR. Therefore, we additionally time-locked the population response to the HLL and repeated the classification analysis. In this case, the linear decoder captures the same periods of the task regardless of the respective RT. This analysis revealed a build-up of action-specific information prior to the HLL (**Supplementary Fig. 6**; PFC: $\text{sum}(t_{13}) = 1074.87$, $P < 0.001$, Cohen's $d = 1.69$; Motor cortex: $\text{sum}(t_9) = 322.79$, $P < 0.001$, Cohen's $d = 1.67$; cluster test), therefore, supporting the notion that the decoder captured action-planning processes."*

2. Finally, we implemented the proposed analysis by the reviewer and trained and tested the decoder on a subset of trials with matched RTs (time-locked to the HLL). Note this analysis was only feasible for the 25% context condition where a sufficient number of trials was available. As shown in **Response Figure 2**, a linear decoder can reliably discriminate the onset of movement with matched RTs (PFC: first cluster, $\text{sum}(t_{13}) = 483.98$, $P = 0.008$, Cohen's $d = 1$; second cluster, $\text{sum}(t_{13}) = 241.98$, $P = 0.037$, Cohen's $d = 1.52$; Motor cortex: $\text{sum}(t_9) = 246.039$, $P = 0.015$, Cohen's $d = 0.94$; cluster test), indicating that the decoder is not confounded by possibly capturing distinct periods of the task.

Response Figure 2. Context-dependent decoding of action. **a**, Grand average decoding accuracy for action in PFC (mean \pm SEM) using only trials with a 25% likelihood. Horizontal lines indicate the extent of significant temporal clusters. **b**, same as (a) but for motor cortex.

Collectively, this set of findings demonstrates that action decoding is not driven by the inclusion of slightly different task periods. On the contrary, these analyses reveal that action information was present prior to the HLL, exhibited a gradual build-up and hence, most likely capturing action-planning processes.

These considerations are now also reflected in the discussion:

Page 31, Line 7-11

"However, we observed a build-up of action-specific information (cf. Figure 5g) that emerged prior to the HLL (Supplementary Fig. 6). This suggests that the observed dynamics track the internal transition from planning to the final movement execution, in line with a recent human iEEG-study²."

2- Line 366: "This indicates that PFC, but not motor cortex, is gradually recruited as a function of behavioral uncertainty (Fig. 5e)." I don't think your results should be interpreted this way. When aligned to button release (as in Fig5), the differences begin emerging around 0.2s before movement, which happens to coincide with the LL (lower limit) onset for the slowest RT trials. If PFC was gradually recruited, these differences should start emerging from the moment that the contextual cue is given (560-750ms before the LL). But that is not the case. Go-cues (like the LL) trigger a condition-invariant signal, or CIS (Kaufman et al. 2016, Guo et al. 2014, Inagaki et al. 2022). In light of the CIS, averaging trials, aligned to movement onset, could lead to the exact figure shown. So this analysis should not be interpreted as a gradual increase.

We thank the reviewer for this relevant query and bringing these papers to our attention. We believe that our description of the task structure caused this misunderstanding. We therefore briefly clarify the task structure again before we discuss our data in light of the CIS.

1. Task structure: Each trial started with a brief baseline period (500ms) followed by a contextual cue presented at the center of the screen (green, orange or red indicating the likelihood of a premature stop). Importantly, after receiving the cue, participants self-paced the trial start (average time to start the trial: $1.76s \pm 0.55s$; mean \pm SD). By pressing the space bar, the target would start moving upwards and reach the LL after 560 – 580ms. As a consequence, the time between the

contextual cue and the moment the target reaches the LL could vary from trial to trial. Once the target reached the LL, participants were instructed to release the button. We have added this information into the methods section:

Page 38, Line 1-4

"Upon receiving the predictive cue, participants were able to start the trial in a self-paced manner by pressing the space bar on the keyboard (average time to start the trial: 1.76s ± 0.55s; mean ± SD)."

2. As correctly remarked, the context-dependent dissociation – when locked to the button release – emerges ~0.2s prior to the actual movement onset. This approximately coincides with the lower limit. However, as can be seen in **Figure 1c**, reaction times strongly vary from trial to trial. Therefore, as depicted in **Response Figure 3**, the time between the movement onset and the moment the target reaches the LL is variable and jittered across trials. This temporal jitter renders it unlikely that our results can be fully explained by a change in neural activity that is time-locked to the LL.

Response Figure 3. Movement onset distribution with respect to HLL. The distribution depicts the movement onset timing relative to the HLL. The vertical dashed line reflects the average movement onset time. Note that the timing is jittered across trials.

3. We also time-locked the population activity to the LL (see comment #1; **Supplementary Figure 5**). This analysis revealed that context-dependent activity already emerged ~0.2s prior to the LL in PFC. This finding supports the univariate analyses where context-dependent activity is evident prior to the LL (**Figure 2a**; **Figure 3a**). In sum, this set of findings demonstrates that the LL did not trigger a large activity change. Instead, these findings highlight that the activity change reflects contextual processing that started prior to reaching the LL.

Collectively, these results are in line with a context-dependent activation in PFC, that is not evident in motor cortex. We carefully considered the suggested papers, in particular Kaufman et al.¹. CIS reflects a large and rapid change prior to the movement onset in

dorsal premotor regions as well as in M1 that reflects more variance than condition-specific (“tuned”) responses. The notion of CIS reflecting the internal transition from motor preparation to movement onset in motor cortex is fully in line with our findings. In the present dataset in humans, the principal motor component does not exhibit context-dependent activity. Therefore, we believe that our findings nicely complement and extend similar findings previously obtained in non-human primates. We now discuss our findings in light of condition-invariant signals in motor cortex:

Page 32, Line 3-14

“Neural activity in motor cortex revealed large, context-independent activity changes prior to the movement onset. This context-independent activity, previously also referred to as a condition-invariant signal (CIS)², typically reflects the largest response component in motor cortex². The present findings are compatible with a CIS in human motor cortex. The first principal component of the HFA-signal in motor cortex is (1) context-independent and (2) explaining the largest variance. An unresolved question is whether other areas might drive this sudden change in motor cortex activity. Previous studies have identified a large-scale network that might provide input to motor cortex, including subcortical^{3,4} and cortical structures^{5,6}. The present findings demonstrate that human PFC also modulates neural activity in motor cortex.”

In contrast to motor cortex activity, PFC population activity was strongly context-dependent (**Figure 5c; Supplementary Figure 5**). Therefore, the population activity in PFC cannot be explained by a CIS. We nuanced our phrasing of a gradual recruitment in PFC:

Page 19, Line 12-15

“This set of findings indicates that population activity in PFC is modulated by predictive context. In contrast, motor cortex population activity displayed context-invariant dynamics supporting a condition-invariant signal² (see Discussion).”

3- LDA analysis: This is a very unconventional way to carry out this analysis. The decoding latent space could be a high-dimensional space that spans multiple PCs. So normally these analyses are done on all dimensions at once (or all dimensions that explain more than 95% of variance), and not one PC dimension at a time, as was done here. I think that this analysis, as carried out, is not very meaningful. If the goal was to enforce orthogonality, you can impose this constraint in the higher-dimensional LDA spaces (although I’m not sure why you’d want to impose orthogonality).

We thank the reviewer for the comment and the excellent suggestion to run the analysis by using the full, high-dimensional space. We separate the response in two parts. First, we would like to clarify the rationale behind our analysis approach. Subsequently, we outline the results of the analysis when performed in high-dimensional space as suggested by the reviewer.

Rationale behind the analysis:

1. Using univariate analyses (**Figure 1-4**), we established that ramping, but not oscillatory dynamics are modulated by predictive context. Instead, oscillatory dynamics primarily supported the interplay between prefrontal and motor cortex. These findings then guided our multivariate analyses where we (1) examined the functional roles of oscillatory and ramping dynamics and (2) identified the

population-level interaction between the prefrontal and motor cortices. Therefore, we first established low-dimensional subspaces that could dissociate contextual from action-specific processing. This approach subsequently allowed us to then test whether the principal dimension with the strongest oscillatory power would (1) encode aspects of contextual processing or action-planning and (2) would temporally synchronize the prefrontal-motor network. In order to mitigate the different orientation of the multidimensional space of prefrontal and motor cortex, we constrained our analysis to the relevant one-dimensional subspace. As indicated by the reviewer, this might seem unconventional, however, it provides the basis to study information transfer between the relevant coding dimensions and to understand the role of oscillatory dynamics in the transfer of information on the population-level. We conducted several control analyses (i.e., decoding in high-dimensional space as suggested by the reviewer) to demonstrate that the relevant coding dimension is well approximated by a single dimension and that adding more dimensions does not necessarily improve the decoding performance.

2. Another reason that led us to constrain our decoding analysis to a low-dimensional subspace was the heterogenous number of electrodes implanted in different patients. In turn, this would have resulted in a largely heterogenous number of PCs that define a high-dimensional space. This is unfortunately an inherent limitation of intracranial recordings in humans.
3. However, we fully agree with the reviewer that the decoding latent space itself could be high-dimensional and thus span multiple PCs. Therefore, in the revised version of the manuscript, we now explicitly emphasize that our decoding analysis does not allow any inference regarding the dimensionality of the decoding latent space.

Page 34, Line 10-13

"Importantly, prefrontal population activity is high-dimensional in nature, where different operations are encoded in distinct subspaces. Yet, our analytical approach does not allow to draw any inference on the dimensionality of the decoding latent space, only on the overall dimensionality of the neural data."

Additional analyses:

Finally, we followed the reviewer's suggestion and performed the analysis using all principal components cumulatively explaining more than 95% of the variance. This revealed comparable effects (**Supplementary Figure 8**) supporting our observation that contextual processing is PFC-dependent whereas action-related processing occurs in both PFC and motor cortex.

Supplementary Figure 8. Decoding analysis of context and action in high-dimensional space. **a**, Grand average decoding accuracy (mean \pm SEM) for context and action in PFC. Classification was performed using all principal components cumulatively explaining more than 95% of the variance. Horizontal lines indicate the extent of significant temporal clusters (color coded for the respective feature). **b**, Grand average decoding accuracy (mean \pm SEM) for context and action in motor cortex. Only action, but not context, could be decoded in motor cortex. Same conventions as in (a).

We have implemented these results into the supplements and directly refer to and discuss them in the main text.

Page 21, Line 19 – Page 22, Line 8

"We performed additional control analyses demonstrating that the context-encoding and action-encoding subspaces capture dissociable processes related to contextual computations and the planning of actions (Supplementary Fig. 7; Methods). We also performed the decoding analyses using a high-dimensional neural space on all principal components cumulatively explaining more than 95% of the variance. This replicated our main findings that the human PFC encodes both context and action (Supplementary Fig. 8; context: first cluster, $\text{sum}(t_{16}) = 408.68$, $P = 0.002$, Cohen's $d = 0.74$; second cluster, $\text{sum}(t_{16}) = 151.62$, $P = 0.039$, Cohen's $d = 0.69$; action: $\text{sum}(t_{16}) = 1040.76$, $P < 0.001$, Cohen's $d = 1.08$; cluster test) whereas motor cortex only encodes action (Supplementary Fig. 8; context: first cluster, $\text{sum}(t_{13}) = 39.46$, $P = 0.427$, Cohen's $d = 0.66$; action: $\text{sum}(t_{13}) = 828.57$, $P < 0.001$, Cohen's $d = 1.11$; cluster test)."

To directly address the similarity between the decoding traces obtained from a high-dimensional vs. one-dimensional set, we adopted an analysis by Chiang et al.⁷ to find the optimal ensemble size for a decoder. We sorted the individual PCs according to their decoding accuracy and then iteratively added the sorted PCs to train the decoder. This analysis revealed that a high-dimensional space does not improve classification performance (Response Figure 4). Overall, the correlation between the decoding time series using multiple PCs and the decoding time series using only the most informative PC showed that the time courses are largely correlated (Response Figure 5).

Response Figure 4. Comparison of decoding performance in low- vs. high-dimensional space. We tested the optimal dimensionality for the linear classifier to decode context as well as action. We sorted individual PCs according to their decoding performance and then iteratively added the sorted PCs to the decoder. This revealed that decoding performance did not benefit from adding more (uninformative) components. Single-subject example, *Upper Left*: Depicts the magnitude of the decoding cluster as a function of the number of sorted PCs to train the decoder on context. *Lower Left*: Time-resolved context-decoding accuracies computed on distinct numbers of PCs. Note the strong temporal similarity of the decoding traces. *Upper Right*: Depicts the magnitude of the decoding cluster as a function of the number of sorted PCs to train the decoder on action. *Lower Left*: Time-resolved action-decoding accuracies computed on distinct numbers of PCs.

Response Figure 5. Group-level correlation between decoding traces resulting from quantification in low vs. high-dimensional spaces. We quantified the correlation between individual decoding traces (*left and middle panel*) to test their similarity. We observed a strong temporal correlation on the group-level (*right panel*) between the decoding traces independent of the number of PCs that were used to train the linear decoder.

Collectively, this set of findings demonstrates that our decoding analyses yields valid and interpretable results that replicated using a high-dimensional set of components.

4- Fig 5i shows that most action peaks occur post-movement (BR), so these cannot be causal in the movement generation nor planning. As such this “serial unfolding” claim does not really rest in solid grounds. This also motivates pg 21: “This raises the question, whether and how the representation of the action plan in PFC is passed on to motor cortex for final action execution?”, which loses force when we realize the post-movement nature of these sequential signals.

We thank the reviewer for the comment. We would like to clarify that we did not intend to draw any conclusions or make any claims regarding causality – neither with respect to movement generation nor movement planning. We apologize if our wording was not

clear-cut. To avoid further confusion on data interpretation, we now explicitly state that we cannot draw any causal inference on the directional flow of information from movement planning to movement generation.

Page 31, Line 2-5

"However, due to the correlative nature of our study we cannot draw any causal inference or directionality nor exclude the possibility that the observed effects could be driven by a third structure that we did not record from."

We would like to clarify that the serial unfolding claim only referred to the temporal segregation between context-integration in PFC and action-planning in both PFC and motor cortex. It did not refer to the temporal dynamics of action-related processes between PFC and motor cortex. In the revised version of the manuscript, we now make this explicit.

Page 22, Line 8-18

*"Having established that context and action can be decoded from the prefrontal-motor network, we next quantified their temporal dynamics by computing the peak-decoding latency. Peak-decoding accuracy revealed a distinct temporal pattern with initial information about contextual information in PFC (**Supplementary Fig. 9**; $P = 0.024$; Kruskal-Wallis; $-0.09 \pm 0.19s$. with respect to BR; mean \pm SD), followed by action-related processes in PFC ($0.05 \pm 0.15s$.) and motor cortex ($0.11 \pm 0.12s$.), suggesting that contextual computations in PFC are completed prior to action-relevant computations. Although decoding accuracy for action reached its maximum right after the movement onset, we observed a clear build-up of action-related information over time, suggesting that this the decoder maps information with respect to action-planning."*

However, we fully concur that it might be counterintuitive that the maximum decoding performance for action only culminates just after the movement onset. Yet, considering the human iEEG literature, this is not unexpected. For example, Haller et al.⁸ observed that the high-frequency activity in motor cortex peaks after the movement onset. Our results (**Figure 2b/5d**) substantiate this observation. Furthermore, Ter Wal et al.² used linear classifier on human iEEG data to decode left vs. right movement. The authors showed that decoding performance about the movement culminates after the movement onset.

There are at least two potential reasons for this observation:

- 1) As the computation of the HFA involves a filter in the temporal domain (i.e. from 70-150 Hz in non-overlapping 10 Hz wide bins), the temporal precision of the data is inevitably reduced as compared to single unit data, a necessary limitation of human intracranial recordings. As a direct consequence, the temporal uncertainty with respect to minima and maxima is increased.
- 2) Figure 2b/c in Ter Wal et al.² shows the decoding performance with respect to left vs. right movement. The decoding trace clearly builds up over time and yields sudden changes $\sim 0.2s$ prior to the movement onset. This fully matches our findings (**Figure 5g/h, left panels**) that action can be successfully decoded $\sim 0.2s$ prior to the BR. This demonstrates that the linear decoder captures the transition from movement planning to movement execution.

We now discuss this finding in more detail in the discussion:

Page 31, Line 5-11

"Furthermore, information about action only culminated after the movement onset. Thus, we cannot preclude that these processes mainly capture post-movement, rather than preparatory dynamics. However, we observed a clear build-up of action-specific information (cf. Figure 5g). This suggests that the observed dynamics track the internal transition from planning to the final movement execution, in line with a recent human iEEG-study²."

With respect to the "information hand-off" of action plans, we initially based our interpretation on (1) the phase-slope-index (Figure 4h) as well as (2) the temporal information of the average action-decoding traces (Figure 5g/h; left panel). In the previous revision of this manuscript, we further added the cross-correlation analysis, peak-decoding analysis as well as the cross-regional generalization analysis (see comment #5) to substantiate our interpretation.

5- The methods description of the cross-regional pattern analysis is not clear from the description of the methods and results section. I'm not sure how you can test classification using different sets of electrodes that were not used in the training of the LDA. As such, I cannot assess fig 5j and the "information handoff" claim.

We thank the reviewer for bringing this up and apologize for the lack of clarity in our description. As pointed out in our response to comment #4, we performed the cross-regional generalization analysis to directly assess the information hand-off between prefrontal and motor cortex.

The reviewer is entirely correct that this analysis would be impossible on different sets of electrodes. As outlined in our response above, we mitigated this problem by focusing on the relevant coding dimension in both regions.

Thus, we did not train and test the classifier on electrodes as features, but instead tested whether the neural information in the PFC action subspace generalizes to the motor cortex action subspace. We thereby tested if a discriminative (informative) pattern that was present in PFC was equally present in motor cortex.

The selection of the coding subspace enabled this analysis, which otherwise would have been impossible as correctly remarked by the reviewer. In the revised version of the manuscript, we made this more explicit in the results section and have further added details to the methods description.

Page 23, Line 7-11

"We trained a linear classifier on the activity of the PFC action subspace to discriminate reaction times and subsequently tested if the classifier generalized to the action subspace in motor cortex. Hence, we tested if a discriminative pattern that was present in the action-specific subspace in PFC was equally present in the action-specific subspace in motor cortex."

Page 51, Line 10-17

"Cross-regional pattern analysis. To quantify whether a discriminative action-specific pattern present in PFC is equally present in motor cortex, we trained a linear classifier on every time point in the action subspace (principal component maximally discriminating action) in PFC and subsequently applied it on every time point in the action subspace in motor cortex. Cross-validation was not necessary since training and testing datasets were independent. Finally, classification values were tested against chance level and corrected for multiple comparison using cluster-based permutation statistics."

Minor Points

1- Fig g and h right are not very useful without stats... the axis spans seem to be chosen to make a particular point rather than to reflect the data's structure.

We concur that the right panels in figure 5g/h were largely descriptive and only served illustrative purposes. We therefore decided to remove them from the manuscript.

2- Pg 32: "Here, we replicate this finding in humans, but in contrast to NHPs, we found no evidence that the human motor cortex encodes predictive context." This is not quite true, since the NHP work referenced provides evidence of pre-movement prediction regarding the specific movement (not context). This study only had one movement (button release) so this could not be assessed.

Thank you for bringing this mistake to our attention. We realized that our referencing was misleading. The replication with respect to a low-dimensional neural pattern in motor cortex referred to the following two references: Gallego et al.⁹ and Suresh et al.¹⁰. The statement "[...] but in contrast to NHPs, we found no evidence that the human motor cortex encodes predictive context" instead referred to Thura and Cisek^{11,12}, Wang et al.¹³ and Glaser et al.¹⁴. However, after carefully re-reading the referenced studies, we decided to nuance the statement that prior work in NHPs has revealed context-dependent coding in the (pre-)motor network. In particular, we now explicitly state that context-dependent coding has mainly been found in adjacent premotor areas, such as frontal eye fields or dorsal premotor-cortex. We changed the text in several instances:

Page 28, Line 21-23

"Previous work in NHP indicated that adjacent premotor structures, such as frontal eye fields^{15,16} or dorsal premotor cortex^{11,14}, might mediate context context-dependent decision-making."

Page 33, Line 13-17

"While previous evidence in NHP indicated that adjacent premotor structures, such as frontal eye fields^{15,16} or dorsal premotor cortex^{11,14} perform context-dependent computations, we found that neural trajectories in prefrontal, but not motor cortex, dissociated the current predictive context."

Page 33, Line 23 – Page 34, Line 9

"Previous work in NHP demonstrated that motor cortex exhibits a low-dimensional structure^{9,10}. Here, we replicate this finding in humans, but in contrast to prior work in NHPs that has revealed context-dependent computations in adjacent premotor structures^{11,14-16}, we found no evidence that the human motor cortex encodes predictive context. We observed that motor cortex relies on input from PFC, which encodes both context as well as the current action plan. However, due to the inherent limited coverage in intracranial recordings, we cannot preclude that specialized sub-regions within the human motor cortex (e.g. anterior or posterior parts of the supplementary motor area, frontal eye fields, premotor cortex) might also encode contextual information."

3- The correlation analysis in Figure 5e can also be confounded by the fact that high-uncertainty trials had longer RTs... the differences in correlation values could be a simple reflection of a larger number of points being included in the correlations of longer-RT trials.

Thank you for the opportunity to clarify. We time-locked the data from -0.5 to +0.3s with respect to the button release across all conditions. Therefore, the correlation analysis is not confounded by a difference in the number of time points considered. In the revised manuscript, we now make this point explicit.

Page 52, Line 14-18

"We computed the correlation coefficient between PC single-trials in PFC and motor cortex from -0.5s to 0.3s with respect to movement onset. This ensured that all trials contained an equal number of data samples, thereby avoiding potential confounds in the correlation value simply due to variable reaction times across trials."

4- PCA methods: There is not enough information in the methods to understand what was done with the PCA analysis. Were the time bins aligned to movement, or to task events? Were all trials (across all contexts) used to build the space, or were spaces built on context-specific activity? Which time bins were included in the PCA analysis?

We apologize for the lack of clarity on how we performed the PCA. We performed PCA on the movement-locked data (button release). For the revised version of the manuscript, we now also performed PCA on the data time-locked to the HLL (**Supplementary Figure 5**). We used all trials to build the PC-space. We have added this critical information into the methods section on dimensionality reduction.

Page 49, Line 12-19

"We used principal component analysis (PCA) to identify linearly uncorrelated population activity patterns and construct a low-dimensional manifold that is embedded in the neural state space spanned by the recorded depth electrodes. We performed PCA on a two-dimensional data matrix (channel x time, trial) locked to either the HLL or to the movement onset. All trials were used to construct the PC-space. The resulting matrix (component x time, trial) was then reshaped into a three-dimensional matrix (trial x component x time) which allowed us to perform single trial analysis in PC space."

5- Answers to reviewers mostly address their issues, except for Reviewer 2, point 1. All those analyses are also consistent with difficulty being the main factor. At the very least this should be discussed as a confound in the paper, rather than brushing it off.

Thank you for this comment. We did not intend to brush the argument off. We agree that it is not possible to entirely and unambiguously disentangle uncertainty from task-difficult, but we performed additional analyses to provide evidence in support of this consideration. We would like to point out that a shift along the internal response axis (criterion) is indicative of an alteration in the participant's decision strategy³⁷. However, we fully acknowledge that an additional paragraph is needed to address this limitation. In the revised version of the manuscript, we have added a paragraph emphasizing this limitation.

Page 39, Line 6-14

"It is important to acknowledge that the environment of intracranial EEG recordings precludes long experiments with many control conditions. Based on our design, we cannot fully disentangle behavioral uncertainty from overall task difficulty. We directly addressed this limitation by

calculating the SDT as well as linear ballistic accumulator models to quantify the participants' response strategy. However, the observed shift in criterion as a function of uncertainty could also be explained by an overall shift of the signal and noise distribution along the internal response axis that would not involve any change in participants' response strategy."

References

- 1 Kaufman, M. T. *et al.* The Largest Response Component in the Motor Cortex Reflects Movement Timing but Not Movement Type. *eNeuro* **3**, doi:10.1523/ENEURO.0085-16.2016 (2016).
- 2 Ter Wal, M. *et al.* Human stereoEEG recordings reveal network dynamics of decision-making in a rule-switching task. *Nat Commun* **11**, 3075, doi:10.1038/s41467-020-16854-w (2020).
- 3 Hauber, W. Involvement of basal ganglia transmitter systems in movement initiation. *Prog Neurobiol* **56**, 507-540, doi:10.1016/s0301-0082(98)00041-0 (1998).
- 4 Philipp, R. & Hoffmann, K. P. Arm movements induced by electrical microstimulation in the superior colliculus of the macaque monkey. *J Neurosci* **34**, 3350-3363, doi:10.1523/JNEUROSCI.0443-13.2014 (2014).
- 5 Pesaran, B., Nelson, M. J. & Andersen, R. A. Free choice activates a decision circuit between frontal and parietal cortex. *Nature* **453**, 406-409, doi:10.1038/nature06849 (2008).
- 6 Murakami, M., Vicente, M. I., Costa, G. M. & Mainen, Z. F. Neural antecedents of self-initiated actions in secondary motor cortex. *Nat Neurosci* **17**, 1574-1582, doi:10.1038/nn.3826 (2014).
- 7 Chiang, F. K., Wallis, J. D. & Rich, E. L. Cognitive strategies shift information from single neurons to populations in prefrontal cortex. *Neuron* **110**, 709-721 e704, doi:10.1016/j.neuron.2021.11.021 (2022).
- 8 Haller, M. *et al.* Persistent neuronal activity in human prefrontal cortex links perception and action. *Nat Hum Behav* **2**, 80-91, doi:10.1038/s41562-017-0267-2 (2018).
- 9 Gallego, J. A., Perich, M. G., Miller, L. E. & Solla, S. A. Neural Manifolds for the Control of Movement. *Neuron* **94**, 978-984, doi:10.1016/j.neuron.2017.05.025 (2017).
- 10 Suresh, A. K. *et al.* Neural population dynamics in motor cortex are different for reach and grasp. *Elife* **9**, doi:10.7554/eLife.58848 (2020).
- 11 Thura, D. & Cisek, P. Deliberation and commitment in the premotor and primary motor cortex during dynamic decision making. *Neuron* **81**, 1401-1416, doi:10.1016/j.neuron.2014.01.031 (2014).
- 12 Thura, D. & Cisek, P. Modulation of Premotor and Primary Motor Cortical Activity during Volitional Adjustments of Speed-Accuracy Trade-Offs. *J Neurosci* **36**, 938-956, doi:10.1523/JNEUROSCI.2230-15.2016 (2016).
- 13 Wang, M. *et al.* Macaque dorsal premotor cortex exhibits decision-related activity only when specific stimulus-response associations are known. *Nat Commun* **10**, 1793, doi:10.1038/s41467-019-09460-y (2019).
- 14 Glaser, J. I., Perich, M. G., Ramkumar, P., Miller, L. E. & Kording, K. P. Population coding of conditional probability distributions in dorsal premotor cortex. *Nat Commun* **9**, 1788, doi:10.1038/s41467-018-04062-6 (2018).

- 15 Mante, V., Sussillo, D., Shenoy, K. V. & Newsome, W. T. Context-dependent computation by recurrent dynamics in prefrontal cortex. *Nature* **503**, 78-84, doi:10.1038/nature12742 (2013).
- 16 Aoi, M. C., Mante, V. & Pillow, J. W. Prefrontal cortex exhibits multidimensional dynamic encoding during decision-making. *Nat Neurosci* **23**, 1410-1420, doi:10.1038/s41593-020-0696-5 (2020).
- 17 Iemi, L., Chaumon, M., Crouzet, S. M. & Busch, N. A. Spontaneous Neural Oscillations Bias Perception by Modulating Baseline Excitability. *J Neurosci* **37**, 807-819, doi:10.1523/JNEUROSCI.1432-16.2016 (2017).

Point-by-Point Reply: NCOMMS-22-05636

We would like to thank the reviewers for their thoughtful and constructive comments. Below, we provide point-by-point responses to all queries. All changes in the revised manuscript are highlighted with tracked changes.

Reply formatting guide:

Reviewer remarks: **bold, black.**

Author response: *blue.*

Changes in the manuscript: *italics, blue.*

Reviewer #1 (Remarks to the Authors)

The authors have done an excellent job in addressing my concerns. I have no further comments and congratulate the authors on a splendid paper.

We would like to thank the reviewer for the constructive review that helped to improve the manuscript.

Reviewer #2 (Remarks to the Authors)

I would like to thank the authors for addressing all of my comments. I agree with their responses and have no further comments on the revised manuscript. This is an impressive work that thoroughly investigated the activity and communication of PFC and motor cortex in their task with iEEG data.

Regarding comment 7 (orthogonality of context and action encoding), what I suggested is to find axes in neural space that maximally discriminate context and action independently (without relating them to PCs, which are granted to be orthogonal) and then compute their angle in neural space. But I agree that orthogonality is not a necessary claim, and I am satisfied with the authors' revision.

We would like to thank the reviewer for the valuable and insightful feedback that substantially improved the manuscript.

Reviewer #3 (Remarks to the Authors)

I think that the results from Figs 1-4 are solid, and the authors did an excellent job at addressing the reviewers' comments on these sections. However, Figs 5-6 have some serious methodological flaws that render them either hard to interpret, or just not valid.

We would like to thank the reviewer for the insightful and constructive comments on our manuscript and the detailed queries regarding figures 5 & 6. The key criticisms are a direct result of either imprecise wording or an insufficient explanation of the rationale and methods. We now provide a more detailed explanation of the analytical approach and the employed methods. In addition, we conducted several additional control analyses, which further substantiated our findings.

Major Points

1- The LL appears to trigger a large activity change. Uncertain trials have longer RTs. So the trajectory analyses could be affected by this difference (as pointed out by Reviewer 2 and acknowledged by the authors). The issue with the action decoder is that it captures different periods of the task for different categories (pre-LL for fast RTs, post-LL for slow RTs and a mixture for the mid RTs). To carry out the intended analysis the authors would need to match the RTs for the different contexts before comparing the activities. Unfortunately, this observation renders the rest of the analyses using this action space uninterpretable.

We would like to thank the reviewer for bringing up this point. We would like to divide our responses to query 1 into two sections to separately address the query on (1) large activity changes upon the lower limit (LL or HLL) and (2) the concern regarding the action decoder.

“The LL appears to trigger a large activity change. Uncertain trials have longer RTs. So the trajectory analyses could be affected by this difference (as pointed out by Reviewer 2 and acknowledged by the authors).”

We agree that a go-signal can often lead to large neural activity changes in motor tasks (see also point #2 below, i.e. Kaufman et al.¹). As outlined below, we first clarified the experimental design and previous analyses and then conducted additional analyses, which collectively demonstrate that the large activity change is not the result of reaching the LL.

1. Experimental design: The lower limit is present throughout every trial. The limit never changes its position, nor does it suddenly appear within each trial. Thus, participants could see the horizontal bar already at trial onset and therefore, prepare their motor response.

I think this part was well explained in the original submission, and I considered this in my comment. A go cue, in this case, could be thought of as the disappearance of the HLL line (occluded by the other ascending bar). A go cue does not need to be the appearance of a sensory stimulus, it can also be a change or disappearance of a sensory stimulus.

The early motor preparation was particularly evident in trials with no uncertainty (0% likelihood of a stop). In this condition, we did not observe a strong change in neural activity in human PFC (**Figure 2a**; **Figure 3a**; **Figure 5c**).

These figures do not show that there is no change in neural activity associated with the go-cue. Figure 2a shows that post-HLL there is a rise in

activity, Figure 3a is just a model fit, so even if there was a transient, it would not be seen in this model, and Figure 5c shows responses aligned to button response, and not HLL. So none of these make the point the authors are attempting to make here.

In order to better understand whether there might be a strong phase-locked response, we computed event-related potentials for the lower limit. As displayed in **Response Figure 1**, we did not observe a strong change in neural activity upon reaching the LL. This finding demonstrates that the lower limit does not trigger a large activity change.

Response Figure 1. Grand-average event-related potentials relative to the HLL. We computed the average ERP (mean \pm SEM) across participants time-locked to the lower limit. Note that there is no phase-locked change in activity when the target reaches the HLL, precluding that the HLL triggers a consistent and large change in neural activity.

As I see it, this plot shows a marked increase in variance post-HLL. This increased variance post-HLL indicates go-cue-triggered changes, which are reflected differently in different electrodes (some increasing and some decreasing activity). If anything, I think this plot proves that go-cue triggers an activity change, and this needs to be taken seriously in subsequent analyses.

2. Single-trial dynamics in uncertain trials (25% or 75% likelihood) exhibit a ramping pattern prior to the lower limit (**Figure 2c/d**; see below). This observation demonstrates that the observed activity patterns were not triggered by the LL, but reflected intrinsic, context-dependent dynamics that were already present prior to the LL.

I am unsure why this is brought up here. Yes, this (and Fig 3a) shows that there is a context-dependent ramping in the PFC prior to HLL, but it does not show that there

is no go-cue-triggered change, nor does it address the issue highlighted about the decoder using different periods.

3. We also time-locked the population response (PC1) in PFC and motor cortex to the LL to supplement our findings relative to the button release (**Supplementary Figure 5**). As visualized in **Supplementary Figure 5**, the population activity in PFC already ramps-up and dissociates the different conditions prior to the LL. This observation is also in line with our findings on univariate ramping dynamics (**Figure 2a/b**): No significant ramping was observed in PFC during trials with no uncertainty (0% likelihood), while significant ramping activity was present for trials with a 25% or 75% likelihood of stopping.

Supplementary Figure 5. Context-dependent population dynamics in PFC emerge prior to the HLL. **a**, Population activity in PFC as indexed by the first principal component reveals context-specific activation patterns prior to the onset of the HLL ($\text{sum}(F_{2,32}) = 637.4$, $P = 0.023$; cluster test). The single-colored horizontal lines show the temporal extent of a significant context-dependent dissociation. Two-colored horizontal lines indicate the temporal extent of significant clusters obtained from pairwise comparisons. **b**, Same as **(a)**, but for motor cortex. Note the similar, context-independent, temporal activation profile across all conditions in motor cortex ($\text{sum}(F_{2,26}) = 251.7$, $P = 0.073$).

This result helps to make the point the authors are trying to make with Figure 5c and d, that PFC is modulated by predictive context. However, to make the point that HLL does not trigger changes in activity the authors would need to show the other PCs (2 and above) to assess the claim that HLL does not trigger a change.

In sum, this set of findings demonstrates that the LL did not trigger a large activity change. On the contrary, these additional analyses demonstrate that the activity change reflects the context integration process that started prior to reaching the LL.

Unfortunately, I don't see the answers as sufficient to demonstrate that LL did not trigger activity changes, which could then confound many of the analyses.

We have added **Supplementary Figure 5** to the supplements and refer to it in the main text:

Page 19, Line 15-18

“We additionally time-locked the population activity to the HLL to ensure that context-dependent activity in PFC ramped up and was dissociable prior to the HLL and not a consequence of an activation change triggered by the HLL (Supplementary Fig. 5).”

Next, we addressed the second concern that was voiced in the first query.

“The issue with the action decoder is that it captures different periods of the task for different categories (pre-LL for fast RTs, post-LL for slow RTs and a mixture for the mid RTs). To carry out the intended analysis the authors would need to match the RTs for the different contexts before comparing the activities. Unfortunately, this observation renders the rest of the analyses using this action space uninterpretable.”

We performed the analysis as suggested by the reviewer as well as another additional control analysis to preclude that the action decoder is confounded by capturing different periods of the task. We will separately outline the two analyses:

1. We agree that the action decoder captures different periods of the task for different categories. However, the lower limit did not trigger large activity changes as outlined above. Therefore, it is unlikely that the action decoder is confounded by capturing different periods of the task. However, in order to exclude this possibility, we time-locked the data to the LL (which serves as the ‘go’ cue) and trained a linear decoder to discriminate reaction times (action). Above-chance decoding prior to the LL indicates that the decoder captures processes involved in the planning of action. As shown in **Supplementary Figure 6**, a linear decoder can successfully discriminate the reaction times prior to the lower limit. This finding indicates that action plans were initiated prior to reaching the lower limit.

Supplementary Figure 6. Action-specific information emerges prior to the HLL. a, Grand average decoding accuracy (mean \pm SEM) for action within PFC

when locked to the HLL. Horizontal lines indicate the extent of significant temporal clusters. **b**, same as **(a)** but for motor cortex. Same conventions as in **(a)**.

We have added **Supplementary Figure 6** to the supplements. We have further added an additional paragraph of this control analysis for clarification:

Page 21, Line 9-19

*“We performed an additional control analysis to preclude that the action-decoding was confounded by capturing different periods of the task (i.e., pre-HLL for fast RTs, post-HLL for slow RTs) when locked to the BR. Therefore, we additionally time-locked the population response to the HLL and repeated the classification analysis. In this case, the linear decoder captures the same periods of the task regardless of the respective RT. This analysis revealed a build-up of action-specific information prior to the HLL (**Supplementary Fig. 6**; PFC: $\text{sum}(t_{13}) = 1074.87$, $P < 0.001$, Cohen’s $d = 1.69$; Motor cortex: $\text{sum}(t_9) = 322.79$, $P < 0.001$, Cohen’s $d = 1.67$; cluster test), therefore, supporting the notion that the decoder captured action-planning processes.”*

Unfortunately, this analysis does not show what the authors intended to show. As it is shown in response figure 3 (copy-pasted below), RTs included in the analysis include a many trials in which subjects responded faster than 100ms after the HLL. These reaction times are too fast to be triggered by the go-cue, so these are predictive in nature. Thus, the pre-HLL decoding shown in Suppl Figure 6 is likely reflecting the execution of these very fast movements instead of action-planning processes that preceded HLL.

2. Finally, we implemented the proposed analysis by the reviewer and trained and tested the decoder on a subset of trials with matched RTs (time-locked to the HLL). Note this analysis was only feasible for the 25% context condition where a sufficient number of trials was available. As shown in **Response Figure 2**, a linear decoder can reliably discriminate the onset of movement with matched RTs (PFC: first cluster, $\text{sum}(t_{13}) = 483.98$, $P = 0.008$, Cohen’s $d = 1$; second cluster, $\text{sum}(t_{13}) = 241.98$, $P = 0.037$, Cohen’s $d = 1.52$; Motor cortex: $\text{sum}(t_9) = 246.039$, $P = 0.015$, Cohen’s $d = 0.94$; cluster test), indicating that the decoder is not confounded by possibly capturing distinct periods of the task.

Response Figure 2. Context-dependent decoding of action. **a**, Grand average decoding accuracy for action in PFC (mean \pm SEM) using only trials with a 25% likelihood. Horizontal lines indicate the extent of significant temporal clusters. **b**, same as **(a)** but for motor cortex.

Unfortunately, this is not the analysis I was suggesting. I am not even sure what “matching RTs” means if only one context was included (25%). What I meant was match RTs across contexts. The manuscript is making claims about the timing of information about context and action (RT), and the handoff of information. The proper analysis would need all 3 contexts RT-matched, so we could see the information about context and RT, like in Figure 5g.

Collectively, this set of finding demonstrates that action decoding is not driven by the inclusion of slightly different task periods. On the contrary, these analyses reveal that action information was present prior to the HLL, exhibited a gradual build-up and hence, most likely capturing action-planning processes.

These considerations are now also reflected in the discussion:

Page 31, Line 7-11

“However, we observed a build-up of action-specific information (cf. Figure 5g) that emerged prior to the HLL (Supplementary Fig. 6). This suggests that the observed dynamics track the internal transition from planning to the final movement execution, in line with a recent human iEEG-study².”

Given the issues raised above, I still think that the results presented have serious confounds, and my original comment stands.

2- Line 366: “This indicates that PFC, but not motor cortex, is gradually recruited as a function of behavioral uncertainty (Fig. 5e).” I don’t think your results should be interpreted this way. When aligned to button release (as in Fig5), the differences begin emerging around 0.2s before movement, which happens to coincide with the LL (lower limit) onset for the slowest RT trials. If PFC was gradually recruited, these differences should start emerging from the moment that the contextual cue is given (560-750ms before the LL). But that is not the case. Go-cues (like the LL) trigger a condition-invariant signal, or CIS (Kaufman et al. 2016, Guo et al. 2014, Inagaki et al. 2022). In light of the CIS, averaging trials, aligned to movement onset, could lead to the exact figure shown. So this analysis should not be interpreted as a gradual increase.

We thank the reviewer for this relevant query and bringing these papers to our attention. We believe that our description of the task structure caused this misunderstanding. We therefore briefly clarify the task structure again before we discuss our data in light of the CIS.

1. Task structure: Each trial started with a brief baseline period (500ms) followed by a contextual cue presented at the center of the screen (green, orange or red indicating the likelihood of a premature stop). Importantly, after receiving the cue, participants self-paced the trial start (average time to start the trial: $1.76s \pm 0.55s$; mean \pm SD). By pressing the space bar, the target would start moving upwards and reach the LL after 560 – 580ms. As a consequence, the time between the contextual cue and the moment the target reaches the LL could vary from trial to trial. Once the target reached the LL, participants were instructed to release the button. We have added this information into the methods section:

Page 38, Line 1-4

“Upon receiving the predictive cue, participants were able to start the trial in a self-paced manner by pressing the space bar on the keyboard (average time to start the trial: $1.76s \pm 0.55s$; mean \pm SD).”

This was clear to me from the original description. I'm not sure why the authors thought that there was a misunderstanding on this based on my comment above.

2. As correctly remarked, the context-dependent dissociation – when locked to the button release – emerges $\sim 0.2s$ prior to the actual movement onset. This approximately coincides with the lower limit. However, as can be seen in **Figure 1c**, reaction times strongly vary from trial to trial. Therefore, as depicted in **Response Figure 3**, the time between the movement onset and the moment the target reaches the LL is variable and jittered across trials. This temporal jitter renders it unlikely that our results can be fully explained by a change in neural activity that is time-locked to the LL.

Response Figure 3. Movement onset distribution with respect to HLL. The distribution depicts the movement onset timing relative to the HLL. The vertical dashed line reflects the average movement onset time. Note that the timing is jittered across trials.

I have addressed this point about HLL-triggered activity thoroughly above. However, this depiction of RT raises even more doubts. It would be useful to show the RT distribution for the different contexts. I imagine that the 0% accounts for most of the <500ms RTs shown here, whereas the 75% context probably has none of these. This asymmetry would of course be very problematic for all the analyses carried out.

More importantly, I am not sure how this observation (that there is RT variability) has any bearing on my comment about contextual cues having to emerge much earlier (the moment the cues are given).

I also just realized that Fig 1c above is not actually showing RTs, but rather something else (in the order of 700ms... maybe time from context cue to response?)

3. We also time-locked the population activity to the LL (see comment #1; **Supplementary Figure 5**). This analysis revealed that context-dependent activity already emerged ~ 0.2 s prior to the LL in PFC. This finding supports the univariate analyses where context-dependent activity is evident prior to the LL (**Figure 2a**; **Figure 3a**). In sum, this set of findings demonstrates that the LL did not trigger a large activity change. Instead, these findings highlight that the activity change reflects contextual processing that started prior to reaching the LL.

Collectively, these results are in line with a context-dependent activation in PFC, that is not evident in motor cortex. We carefully considered the suggested papers, in particular Kaufman et al.¹. CIS reflects a large and rapid change prior to the movement onset in dorsal premotor regions as well as in M1 that reflects more variance than condition-specific ("tuned") responses. The notion of CIS reflecting the internal transition from motor preparation to movement onset in motor cortex is fully in line with our findings. In the present dataset in humans, the principal motor component does not exhibit context-dependent activity. Therefore, we believe that our findings nicely complement and extend similar findings previously obtained in non-human primates. We now discuss our findings in light of condition-invariant signals in motor cortex:

Page 32, Line 3-14

“Neural activity in motor cortex revealed large, context-independent activity changes prior to the movement onset. This context-independent activity, previously also referred to as a condition-invariant signal (CIS)¹, typically reflects the largest response component in motor cortex¹. The present findings are compatible with a CIS in human motor cortex. The first principal component of the HFA-signal in motor cortex is (1) context-independent and (2) explaining the largest variance. An unresolved question is whether other areas might drive this sudden change in motor cortex activity. Previous studies have identified a large-scale network that might provide input to motor cortex, including subcortical^{3,4} and cortical structures^{5,6}. The present findings demonstrate that human PFC also modulates neural activity in motor cortex.”

In contrast to motor cortex activity, PFC population activity was strongly context-dependent (**Figure 5c**; **Supplementary Figure 5**). Therefore, the population activity in PFC cannot be explained by a CIS. We nuanced our phrasing of a gradual recruitment in PFC:

Page 19, Line 12-15

“This set of findings indicates that population activity in PFC is modulated by predictive context. In contrast, motor cortex population activity displayed context-invariant dynamics supporting a condition-invariant signal¹ (see Discussion).”

3- LDA analysis: This is a very unconventional way to carry out this analysis. The decoding latent space could be a high-dimensional space that spans multiple PCs. So normally these analyses are done on all dimensions at once (or all dimensions that explain more than 95% of variance), and not one PC dimension at a time, as was done here. I think that this analysis, as carried out, is not very meaningful. If the goal was to enforce orthogonality, you can impose this constraint in the higher-dimensional LDA spaces (although I’m not sure why you’d want to impose orthogonality).

We thank the reviewer for the comment and the excellent suggestion to run the analysis by using the full, high-dimensional space. We separate the response in two parts. First, we would like to clarify the rationale behind our analysis approach. Subsequently, we outline the results of the analysis when performed in high-dimensional space as suggested by the reviewer.

Rationale behind the analysis:

1. Using univariate analyses (**Figure 1-4**), we established that ramping, but not oscillatory dynamics are modulated by predictive context. Instead, oscillatory dynamics primarily supported the interplay between prefrontal and motor cortex. These findings then guided our multivariate analyses where we (1) examined the functional roles of oscillatory and ramping dynamics and (2) identified the population-level interaction between the prefrontal and motor cortices. Therefore, we first established low-dimensional subspaces that could dissociate contextual from action-specific processing. This approach subsequently allowed us to then test whether the principal dimension with the strongest oscillatory power would (1) encode aspects of contextual processing or action-planning and (2) would temporally synchronize the prefrontal-motor network. In order to mitigate the different orientation of the

multidimensional space of prefrontal and motor cortex, we constrained our analysis to the relevant one-dimensional subspace. As indicated by the reviewer, this might seem unconventional, however, it provides the basis to study information transfer between the relevant coding dimensions and to understand the role of oscillatory dynamics in the transfer of information on the population-level. We conducted several control analyses (i.e., decoding in high-dimensional space as suggested by the reviewer) to demonstrate that the relevant coding dimension is well approximated by a single dimension and that adding more dimensions does not necessarily improve the decoding performance.

2. Another reason that led us to constrain our decoding analysis to a low-dimensional subspace was the heterogeneous number of electrodes implanted in different patients. In turn, this would have resulted in a largely heterogeneous number of PCs that define a high-dimensional space. This is unfortunately an inherent limitation of intracranial recordings in humans.
3. However, we fully agree with the reviewer that the decoding latent space itself could be high-dimensional and thus span multiple PCs. Therefore, in the revised version of the manuscript, we now explicitly emphasize that our decoding analysis does not allow any inference regarding the dimensionality of the decoding latent space.

Page 34, Line 10-13

“Importantly, prefrontal population activity is high-dimensional in nature, where different operations are encoded in distinct subspaces. Yet, our analytical approach does not allow to draw any inference on the dimensionality of the decoding latent space, only on the overall dimensionality of the neural data.”

Noted on the rationale, and the new analysis. It makes sense.

Additional analyses:

Finally, we followed the reviewer’s suggestion and performed the analysis using all principal components cumulatively explaining more than 95% of the variance. This revealed comparable effects (**Supplementary Figure 8**) supporting our observation that contextual processing is PFC-dependent whereas action-related processing occurs in both PFC and motor cortex.

Supplementary Figure 8. Decoding analysis of context and action in high-dimensional space. a, Grand average decoding accuracy (mean \pm SEM) for

context and action in PFC. Classification was performed using all principal components cumulatively explaining more than 95% of the variance. Horizontal lines indicate the extent of significant temporal clusters (color coded for the respective feature). **b**, Grand average decoding accuracy (mean \pm SEM) for context and action in motor cortex. Only action, but not context, could be decoded in motor cortex. Same conventions as in **(a)**.

We have implemented these results into the supplements and directly refer to and discuss them in the main text.

Page 21, Line 19 – Page 22, Line 8

*“We performed additional control analyses demonstrating that the context-encoding and action-encoding subspaces capture dissociable processes related to contextual computations and the planning of actions (**Supplementary Fig. 7; Methods**). We also performed the decoding analyses using a high-dimensional neural space on all principal components cumulatively explaining more than 95% of the variance. This replicated our main findings that the human PFC encodes both context and action (**Supplementary Fig. 8**; context: first cluster, $sum(t_{16}) = 408.68$, $P = 0.002$, Cohen’s $d = 0.74$; second cluster, $sum(t_{16}) = 151.62$, $P = 0.039$, Cohen’s $d = 0.69$; action: $sum(t_{16}) = 1040.76$, $P < 0.001$, Cohen’s $d = 1.08$; cluster test) whereas motor cortex only encodes action (**Supplementary Fig. 8**; context: first cluster, $sum(t_{13}) = 39.46$, $P = 0.427$, Cohen’s $d = 0.66$; action: $sum(t_{13}) = 828.57$, $P < 0.001$, Cohen’s $d = 1.11$; cluster test).”*

To directly address the similarity between the decoding traces obtained from a high-dimensional vs. one-dimensional set, we adopted an analysis by Chiang et al.⁷ to find the optimal ensemble size for a decoder. We sorted the individual PCs according to their decoding accuracy and then iteratively added the sorted PCs to train the decoder. This analysis revealed that a high-dimensional space does not improve classification performance (**Response Figure 4**). Overall, the correlation between the decoding time series using multiple PCs and the decoding time series using only the most informative PC showed that the time courses are largely correlated (**Response Figure 5**).

Response Figure 4. Comparison of decoding performance in low- vs. high-dimensional space. We tested the optimal dimensionality for the linear classifier to decode context as well as action. We sorted individual PCs according to their decoding performance and then iteratively added the sorted PCs to the decoder. This revealed that decoding performance did not benefit from adding more (uninformative) components. Single-subject example, *Upper Left*: Depicts the magnitude of the decoding cluster as a function of the number of sorted PCs to train the decoder on context. *Lower Left*: Time-resolved context-decoding accuracies computed on distinct numbers of PCs. Note the strong temporal similarity of the decoding traces. *Upper Right*: Depicts the magnitude of the decoding cluster as a function of the number of sorted PCs to train the decoder on action. *Lower Left*: Time-resolved action-decoding accuracies computed on distinct numbers of PCs.

Response Figure 5. Group-level correlation between decoding traces resulting from quantification in low vs. high-dimensional spaces. We quantified the correlation between individual decoding traces (*left and middle panel*) to test their similarity. We observed a strong temporal correlation on the group-level (*right panel*) between the decoding traces independent of the number of PCs that were used to train the linear decoder.

Collectively, this set of findings demonstrates that our decoding analyses yields valid and interpretable results that replicated using a high-dimensional set of components.

These analyses indeed do show that this information lives in low-D spaces, and support the authors claim.

4- Fig 5i shows that most action peaks occur post-movement (BR), so these cannot be causal in the movement generation nor planning. As such this “serial unfolding” claim does not really rest in solid grounds. This also motivates pg 21: “This raises the question, whether and how the representation of the action plan in PFC is passed on to motor cortex for final action execution?”, which loses force when we realize the post-movement nature of these sequential signals.

We thank the reviewer for the comment. We would like to clarify that we did not intend to draw any conclusions or make any claims regarding causality – neither with respect to movement generation nor movement planning. We apologize if our wording was not clear-cut. To avoid further confusion on data interpretation, we

now explicitly state that we cannot draw any causal inference on the directional flow of information from movement planning to movement generation.

Page 31, Line 2-5

“However, due to the correlative nature of our study we cannot draw any causal inference or directionality nor exclude the possibility that the observed effects could be driven by a third structure that we did not record from.”

We would like to clarify that the serial unfolding claim only referred to the temporal segregation between context-integration in PFC and action-planning in both PFC and motor cortex. It did not refer to the temporal dynamics of action-related processes between PFC and motor cortex. In the revised version of the manuscript, we now make this explicit.

Page 22, Line 8-18

*“Having established that context and action can be decoded from the prefrontal-motor network, we next quantified their temporal dynamics by computing the peak-decoding latency. Peak-decoding accuracy revealed a distinct temporal pattern with initial information about contextual information in PFC (**Supplementary Fig. 9**; $P = 0.024$; Kruskal-Wallis; $-0.09 \pm 0.19s.$ with respect to BR; mean \pm SD), followed by action-related processes in PFC ($0.05 \pm 0.15s.$) and motor cortex ($0.11 \pm 0.12s.$), suggesting that contextual computations in PFC are completed prior to action-relevant computations. Although decoding accuracy for action reached its maximum right after the movement onset, we observed a clear build-up of action-related information over time, suggesting that this the decoder maps information with respect to action-planning.”*

However, we fully concur that it might be counterintuitive that the maximum decoding performance for action only culminates just after the movement onset. Yet, considering the human iEEG literature, this is not unexpected. For example, Haller et al.⁸ observed that the high-frequency activity in motor cortex peaks after the movement onset. Our results (**Figure 2b/5d**) substantiate this observation. Furthermore, Ter Wal et al.² used linear classifier on human iEEG data to decode left vs. right movement. The authors showed that decoding performance about the movement culminates after the movement onset.

There are at least two potential reasons for this observation:

- 1) As the computation of the HFA involves a filter in the temporal domain (i.e. from 70-150 Hz in non-overlapping 10 Hz wide bins), the temporal precision of the data is inevitably reduced as compared to single unit data, a necessary limitation of human intracranial recordings. As a direct consequence, the temporal uncertainty with respect to minima and maxima is increased.
- 2) Figure 2b/c in Ter Wal et al.² shows the decoding performance with respect to left vs. right movement. The decoding trace clearly builds up over time and yields sudden changes $\sim 0.2s$ prior to the movement onset. This fully matches our findings (**Figure 5g/h, left panels**) that action can be successfully decoded $\sim 0.2s$ prior to the BR. This demonstrates that the linear decoder captures the transition from movement planning to movement execution.

To be honest, I was not perplexed by the post-movement decoding peak. This is observed over and over in many brain regions. What I was raising is the use of this

metric as a means to talk about the serial unfolding, or any temporal metric. If anything, I would have expected the time of first significant decoding time-point as the relevant metric.

We now discuss this finding in more detail in the discussion:

Page 31, Line 5-11

*“Furthermore, information about action only culminated after the movement onset. Thus, we cannot preclude that these processes mainly capture post-movement, rather than preparatory dynamics. However, we observed a clear build-up of action-specific information (cf. **Figure 5g**). This suggests that the observed dynamics track the internal transition from planning to the final movement execution, in line with a recent human iEEG-study².”*

With respect to the “information hand-off” of action plans, we initially based our interpretation on (1) the phase-slope-index (**Figure 4h**) as well as (2) the temporal information of the average action-decoding traces (**Figure 5g/h; left panel**). In the previous revision of this manuscript, we further added the cross-correlation analysis, peak-decoding analysis as well as the cross-regional generalization analysis (see comment #5) to substantiate our interpretation.

Maybe this is discussed later, but as of now I see no reason to see any of these results as “information hand-off”

5- The methods description of the cross-regional pattern analysis is not clear from the description of the methods and results section. I’m not sure how you can test classification using different sets of electrodes that were not used in the training of the LDA. As such, I cannot assess fig 5j and the “information handoff” claim.

We thank the reviewer for bringing this up and apologize for the lack of clarity in our description. As pointed out in our response to comment #4, we performed the cross-regional generalization analysis to directly assess the information hand-off between prefrontal and motor cortex.

The reviewer is entirely correct that this analysis would be impossible on different sets of electrodes. As outlined in our response above, we mitigated this problem by focusing on the relevant coding dimension in both regions.

Thus, we did not train and test the classifier on electrodes as features, but instead tested whether the neural information in the PFC action subspace generalizes to the motor cortex action subspace. We thereby tested if a discriminative (informative) pattern that was present in PFC was equally present in motor cortex. The selection of the coding subspace enabled this analysis, which otherwise would have been impossible as correctly remarked by the reviewer. In the revised version of the manuscript, we made this more explicit in the results section and have further added details to the methods description.

Page 23, Line 7-11

“We trained a linear classifier on the activity of the PFC action subspace to discriminate reaction times and subsequently tested if the classifier generalized to the action subspace in motor cortex. Hence, we tested if a discriminative

pattern that was present in the action-specific subspace in PFC was equally present in the action-specific subspace in motor cortex.”

Page 51, Line 10-17

“Cross-regional pattern analysis. To quantify whether a discriminative action-specific pattern present in PFC is equally present in motor cortex, we trained a linear classifier on every time point in the action subspace (principal component maximally discriminating action) in PFC and subsequently applied it on every time point in the action subspace in motor cortex. Cross-validation was not necessary since training and testing datasets were independent. Finally, classification values were tested against chance level and corrected for multiple comparison using cluster-based permutation statistics.”

Thank you. This makes more sense (in terms of methods). However, I am still unclear on how this analysis supports information hand-off.

Minor Points

1- Fig g and h right are not very useful without stats... the axis spans seem to be chosen to make a particular point rather than to reflect the data’s structure.

We concur that the right panels in figure 5g/h were largely descriptive and only served illustrative purposes. We therefore decided to remove them from the manuscript.

Thanks.

2- Pg 32: “Here, we replicate this finding in humans, but in contrast to NHPs, we found no evidence that the human motor cortex encodes predictive context.” This is not quite true, since the NHP work referenced provides evidence of pre-movement prediction regarding the specific movement (not context). This study only had one movement (button release) so this could not be assessed.

Thank you for bringing this mistake to our attention. We realized that our referencing was misleading. The replication with respect to a low-dimensional neural pattern in motor cortex referred to the following two references: Gallego et al.⁹ and Suresh et al.¹⁰. The statement “[...] but in contrast to NHPs, we found no evidence that the human motor cortex encodes predictive context” instead referred to Thura and Cisek^{11,12}, Wang et al.¹³ and Glaser et al.¹⁴. However, after carefully re-reading the referenced studies, we decided to nuance the statement that prior work in NHPs has revealed context-dependent coding in the (pre-)motor network. In particular, we now explicitly state that context-dependent coding has mainly been found in adjacent premotor areas, such as frontal eye fields or dorsal premotor-cortex. We changed the text in several instances:

Page 28, Line 21-23

“Previous work in NHP indicated that adjacent premotor structures, such as frontal eye fields^{15,16} or dorsal premotor cortex^{11,14}, might mediate context context-dependent decision-making.”

Page 33, Line 13-17

“While previous evidence in NHP indicated that adjacent premotor structures, such as frontal eye fields^{15,16} or dorsal premotor cortex^{11,14} perform context-dependent computations, we found that neural trajectories in prefrontal, but not motor cortex, dissociated the current predictive context.”

Page 33, Line 23 – Page 34, Line 9

“Previous work in NHP demonstrated that motor cortex exhibits a low-dimensional structure^{9,10}. Here, we replicate this finding in humans, but in contrast to prior work in NHPs that has revealed context-dependent computations in adjacent premotor structures^{11,14-16}, we found no evidence that the human motor cortex encodes predictive context. We observed that motor cortex relies on input from PFC, which encodes both context as well as the current action plan. However, due to the inherent limited coverage in intracranial recordings, we cannot preclude that specialized sub-regions within the human motor cortex (e.g. anterior or posterior parts of the supplementary motor area, frontal eye fields, premotor cortex) might also encode contextual information.”

Thanks for the clarifications

3- The correlation analysis in Figure 5e can also be confounded by the fact that high-uncertainty trials had longer RTs... the differences in correlation values could be a simple reflection of a larger number of points being included in the correlations of longer-RT trials.

Thank you for the opportunity to clarify. We time-locked the data from -0.5 to +0.3s with respect to the button release across all conditions. Therefore, the correlation analysis is not confounded by a difference in the number of time points considered. In the revised manuscript, we now make this point explicit.

Page 52, Line 14-18

“We computed the correlation coefficient between PC single-trials in PFC and motor cortex from -0.5s to 0.3s with respect to movement onset. This ensured that all trials contained an equal number of data samples, thereby avoiding potential confounds in the correlation value simply due to variable reaction times across trials.”

Noted, thanks.

4- PCA methods: There is not enough information in the methods to understand what was done with the PCA analysis. Were the time bins aligned to movement, or to task events? Were all trials (across all contexts) used to build the space, or were spaces built on context-specific activity? Which time bins were included in the PCA analysis?

We apologize for the lack of clarity on how we performed the PCA. We performed PCA on the movement-locked data (button release). For the revised version of the manuscript, we now also performed PCA on the data time-locked to the HLL (**Supplementary Figure 5**). We used all trials to build the PC-space. We have added this critical information into the methods section on dimensionality reduction.

Page 49, Line 12-19

“We used principal component analysis (PCA) to identify linearly uncorrelated population activity patterns and construct a low-dimensional manifold that is

embedded in the neural state space spanned by the recorded depth electrodes. We performed PCA on a two-dimensional data matrix (channel x time, trial) locked to either the HLL or to the movement onset. All trials were used to construct the PC-space. The resulting matrix (component x time, trial) was then reshaped into a three-dimensional matrix (trial x component x time) which allowed us to perform single trial analysis in PC space.”

Noted, thanks

5- Answers to reviewers mostly address their issues, except for Reviewer 2, point 1. All those analyses are also consistent with difficulty being the main factor. At the very least this should be discussed as a confound in the paper, rather than brushing it off.

Thank you for this comment. We did not intend to brush the argument off. We agree that it is not possible to entirely and unambiguously disentangle uncertainty from task-difficult, but we performed additional analyses to provide evidence in support of this consideration. We would like to point out that a shift along the internal response axis (criterion) is indicative of an alteration in the participant’s decision strategy¹⁷. However, we fully acknowledge that an additional paragraph is needed to address this limitation. In the revised version of the manuscript, we have added a paragraph emphasizing this limitation.

Page 39, Line 6-14

“It is important to acknowledge that the environment of intracranial EEG recordings precludes long experiments with many control conditions. Based on our design, we cannot fully disentangle behavioral uncertainty from overall task difficulty. We directly addressed this limitation by calculating the SDT as well as linear ballistic accumulator models to quantify the participants’ response strategy. However, the observed shift in criterion as a function of uncertainty could also be explained by an overall shift of the signal and noise distribution along the internal response axis that would not involve any change in participants’ response strategy.”

Thank you for the addition

References

- 1 Kaufman, M. T. *et al.* The Largest Response Component in the Motor Cortex Reflects Movement Timing but Not Movement Type. *eNeuro* **3**, doi:10.1523/ENEURO.0085-16.2016 (2016).
- 2 Ter Wal, M. *et al.* Human stereoEEG recordings reveal network dynamics of decision-making in a rule-switching task. *Nat Commun* **11**, 3075, doi:10.1038/s41467-020-16854-w (2020).
- 3 Hauber, W. Involvement of basal ganglia transmitter systems in movement initiation. *Prog Neurobiol* **56**, 507-540, doi:10.1016/s0301-0082(98)00041-0 (1998).
- 4 Philipp, R. & Hoffmann, K. P. Arm movements induced by electrical microstimulation in the superior colliculus of the macaque monkey. *J Neurosci* **34**, 3350-3363, doi:10.1523/JNEUROSCI.0443-13.2014 (2014).

- 5 Pesaran, B., Nelson, M. J. & Andersen, R. A. Free choice activates a decision
circuit between frontal and parietal cortex. *Nature* **453**, 406-409,
doi:10.1038/nature06849 (2008).
- 6 Murakami, M., Vicente, M. I., Costa, G. M. & Mainen, Z. F. Neural antecedents
of self-initiated actions in secondary motor cortex. *Nat Neurosci* **17**, 1574-
1582, doi:10.1038/nn.3826 (2014).
- 7 Chiang, F. K., Wallis, J. D. & Rich, E. L. Cognitive strategies shift information
from single neurons to populations in prefrontal cortex. *Neuron* **110**, 709-721
e704, doi:10.1016/j.neuron.2021.11.021 (2022).
- 8 Haller, M. *et al.* Persistent neuronal activity in human prefrontal cortex links
perception and action. *Nat Hum Behav* **2**, 80-91, doi:10.1038/s41562-017-
0267-2 (2018).
- 9 Gallego, J. A., Perich, M. G., Miller, L. E. & Solla, S. A. Neural Manifolds for
the Control of Movement. *Neuron* **94**, 978-984,
doi:10.1016/j.neuron.2017.05.025 (2017).
- 10 Suresh, A. K. *et al.* Neural population dynamics in motor cortex are different
for reach and grasp. *Elife* **9**, doi:10.7554/eLife.58848 (2020).
- 11 Thura, D. & Cisek, P. Deliberation and commitment in the premotor and
primary motor cortex during dynamic decision making. *Neuron* **81**, 1401-1416,
doi:10.1016/j.neuron.2014.01.031 (2014).
- 12 Thura, D. & Cisek, P. Modulation of Premotor and Primary Motor Cortical
Activity during Volitional Adjustments of Speed-Accuracy Trade-Offs. *J*
Neurosci **36**, 938-956, doi:10.1523/JNEUROSCI.2230-15.2016 (2016).
- 13 Wang, M. *et al.* Macaque dorsal premotor cortex exhibits decision-related
activity only when specific stimulus-response associations are known. *Nat*
Commun **10**, 1793, doi:10.1038/s41467-019-09460-y (2019).
- 14 Glaser, J. I., Perich, M. G., Ramkumar, P., Miller, L. E. & Kording, K. P.
Population coding of conditional probability distributions in dorsal premotor
cortex. *Nat Commun* **9**, 1788, doi:10.1038/s41467-018-04062-6 (2018).
- 15 Mante, V., Sussillo, D., Shenoy, K. V. & Newsome, W. T. Context-dependent
computation by recurrent dynamics in prefrontal cortex. *Nature* **503**, 78-84,
doi:10.1038/nature12742 (2013).
- 16 Aoi, M. C., Mante, V. & Pillow, J. W. Prefrontal cortex exhibits
multidimensional dynamic encoding during decision-making. *Nat Neurosci* **23**,
1410-1420, doi:10.1038/s41593-020-0696-5 (2020).
- 17 Iemi, L., Chaumon, M., Crouzet, S. M. & Busch, N. A. Spontaneous Neural
Oscillations Bias Perception by Modulating Baseline Excitability. *J Neurosci*
37, 807-819, doi:10.1523/JNEUROSCI.1432-16.2016 (2017).

Reviewers' Comments:

Reviewer #3:

Remarks to the Author:

Point-by-Point Reply: NCOMMS-22-05636B

We would like to thank the reviewers for their thoughtful and constructive comments. Below, we provide point-by-point responses to all queries. All changes in the revised manuscript are highlighted with tracked changes.

Reply formatting guide:

Reviewer remarks revision #2: **bold, black.**

Author response revision #2: *blue.*

Changes in the manuscript revision #2: *blue italics.*

Revision #3:

We highlight the additional comments of reviewer 3, that were embedded in our second rebuttal letter, in **red**. We now address these comments within the respective context (highlighted in **green**). Conversely, changes in the manuscript are now highlighted in *green italics*.

Reviewer #1 (Remarks to the Authors)

The authors have done an excellent job in addressing my concerns. I have no further comments and congratulate the authors on a splendid paper.

We would like to thank the reviewer for the constructive review that helped to improve the manuscript.

Reviewer #2 (Remarks to the Authors)

I would like to thank the authors for addressing all of my comments. I agree with their responses and have no further comments on the revised manuscript. This is an impressive work that thoroughly investigated the activity and communication of PFC and motor cortex in their task with iEEG data.

Regarding comment 7 (orthogonality of context and action encoding), what I suggested is to find axes in neural space that maximally discriminate context and action independently (without relating them to PCs, which are granted to be orthogonal) and then compute their angle in neural space. But I agree that orthogonality is not a necessary claim, and I am satisfied with the authors' revision.

We would like to thank the reviewer for the valuable and insightful feedback that substantially improved the manuscript.

Reviewer #3 (Remarks to the Authors)

I think that the results from Figs 1-4 are solid, and the authors did an excellent job at addressing the reviewers' comments on these sections. However, Figs 5-6 have some serious methodological flaws that render them either hard to interpret, or just not valid.

We would like to thank the reviewer for the insightful and constructive comments on our manuscript and the detailed queries regarding figures 5 & 6. The key criticisms are a direct result of either imprecise wording or an insufficient explanation of the rationale and methods. We now provide a more detailed explanation of the analytical approach and the employed methods. In addition, we conducted several additional control analyses, which further substantiated our findings.

Major Points

1- The LL appears to trigger a large activity change. Uncertain trials have longer RTs. So the trajectory analyses could be affected by this difference (as pointed out by Reviewer 2 and acknowledged by the authors). The issue with the action decoder is that it captures different periods of the task for different categories (pre-LL for fast RTs, post-LL for slow RTs and a mixture for the mid RTs). To carry out the intended analysis the authors would need to match the RTs for the different contexts before comparing the activities. Unfortunately, this observation renders the rest of the analyses using this action space uninterpretable.

We would like to thank the reviewer for bringing up this point. We would like to divide our responses to query 1 into two sections to separately address the query on (1) large activity changes upon the lower limit (LL or HLL) and (2) the concern regarding the action decoder.

"The LL appears to trigger a large activity change. Uncertain trials have longer RTs. So the trajectory analyses could be affected by this difference (as pointed out by Reviewer 2 and acknowledged by the authors)."

We agree that a go-signal can often lead to large neural activity changes in motor tasks (see also point #2 below, i.e. Kaufman et al.¹). As outlined below, we first clarified the experimental design and previous analyses and then conducted additional analyses, which collectively demonstrate that the large activity change is not the result of reaching the LL.

1. Experimental design: The lower limit is present throughout every trial. The limit never changes its position, nor does it suddenly appear within each trial. Thus, participants could see the horizontal bar already at trial onset and therefore, prepare their motor response.

Reviewer comment #1:

I think this part was well explained in the original submission, and I considered this in my comment. A go cue, in this case, could be thought of as the disappearance of the HLL line (occluded by the other ascending bar). A go cue does not need to be the appearance of a sensory stimulus, it can also be a change or disappearance of a sensory stimulus.

Our response to #1:

We fully agree with the reviewer that a go cue does not need to be the appearance of a sensory stimulus. However, it is not correct that the lower limit was occluded by the ascending bar as the lower limit was visible throughout the entire trial. Hence, the only sensory information that was dynamically changing during the trial was the ascending bar; the rest remained constant.

Further additional analyses that we demonstrate below in fact clarify that there were no prominent changes in neural activity triggered by the lower limit. However, we added the following paragraph to the discussion in order to provide a balanced view:

Page 34

"Furthermore, while we did not employ a designated go-cue in our experimental paradigm, the lower limit (cf. Fig. 1a) might still resemble a go-cue and could potentially trigger condition-invariant activity changes. Consequently, based on the current experimental paradigm, we cannot fully disentangle neural activity that reflects context-dependent processing from neural activity potentially triggered by the lower limit. However, the fact that context-dependent dynamics already evolved (cf. Fig. 2a & Supplementary Fig. 5a) and neural dynamics mainly ramped-up prior to lower limit suggests that the observed neural activity patterns in PFC were not solely triggered by the lower limit, but reflected intrinsic, context-dependent dynamics prior to the lower limit."

The early motor preparation was particularly evident in trials with no uncertainty (0% likelihood of a stop). In this condition, we did not observe a strong change in neural activity in human PFC (Figure 2a; Figure 3a; Figure 5c).

Reviewer comment #2:

These figures do not show that there is no change in neural activity associated with the go-cue. Figure 2a shows that post-HLL there is a rise in activity, Figure 3a is just a model fit, so even if there was a transient, it would not be seen in this model, and Figure 5c shows responses aligned to button response, and not HLL. So none of these make the point the authors are attempting to make here.

Our response to #2:

We respectfully disagree. First, the change in activity upon the lower limit should not be specific to the 0% condition, but should instead be observable across all conditions (0%, 25% & 75% likelihood of stopping). However, as can be clearly seen in Figure 2a, the rise in activity for trials with a 25% and 75% likelihood of stopping does not coincide with the lower limit, but occurs before that. Furthermore, we computed the average HFA activity before and after the lower limit for the 0% condition to test whether the HFA activity significantly differed between the two time periods. This revealed that HFA activity did not change from before to after the lower limit (Response Fig. 1); hence rendering it very unlikely that the lower limit triggered large activity changes. Instead, this finding supports our statement that the human PFC was not 'recruited' during trials with no uncertainty (in line with no significant changes in the ramping slope; cf. Figure 3a). In comparison, and in

line with what the reviewer proposed, activity in motor cortex was condition-invariant and revealed strong ramping dynamics even during trials with no uncertainty. Hence, we concluded that early motor preparation was particularly evident in trials with no uncertainty, since neural activity in PFC did not show significant activation changes.

Response Figure 1. We computed the average HFA before (-0.5 to -0.015) and after (0 to 0.35) the lower limit. This revealed that the HFA did not significantly change from pre- to post-lower-limit (Wilcoxon rank sum test; $P = 0.61$).

In order to better understand whether there might be a strong phase-locked response, we computed event-related potentials for the lower limit. As displayed in **Response Figure 1**, we did not observe a strong change in neural activity upon reaching the LL. This finding demonstrates that the lower limit does not trigger a large activity change.

Response Figure 1. Grand-average event-related potentials relative to the HLL. We computed the average ERP (mean \pm SEM) across participants time-locked to the lower limit. Note that there is no phase-locked change in activity when the target reaches the HLL, precluding that the HLL triggers a consistent and large change in neural activity.

Reviewer comment #3:

As I see it, this plot shows a marked increase in variance post-HLL. This increased variance post-HLL indicates go-cue-triggered changes, which are reflected differently in different electrodes (some increasing and some decreasing activity). If anything, I think this plot proves that go-cue triggers an activity change, and this needs to be taken seriously in subsequent analyses.

Our response to #3:

We thank the reviewer for this comment. We here aimed to make the point that there is no phasic response upon the lower limit. Therefore, we considered the time between -500ms to 0ms prior to the lower limit as 'baseline' and subtracted the mean of each channel from the time series. Therefore, it is trivial that the variance upon the lower limit increases as compared to 'baseline'. We re-computed the grand-average ERP without centering our data (**Response Fig. 2**). We directly tested whether there would be a significant increase in variance (across subjects) from pre- to post-lower limit using a Levene's test. This revealed that there is no statistically significant increase in variance upon the lower limit (Levene's test; $P = 0.578$).

Response Figure 2. The variance of event-related potentials did not increase from pre- to post-lower limit. Furthermore, there is no phasic response upon the lower limit.

We also performed a Levene's test on single-subject data to test the reviewer's statement that the presumed increase in variance mirrors the variance across channels with some increasing and some decreasing activity. This revealed that the variance across channels did not differ between pre- and post-lower limit in 15/17 participants (Binomial test; $P = 0.0023$; **Response Fig. 3**). Hence, we conclude that there is neither a phase-locked response nor an increase in variance upon the lower limit.

Response Figure 3. The variance of event-related potentials did not increase from pre- to post-lower limit on a single-subject level.

- Single-trial dynamics in uncertain trials (25% or 75% likelihood) exhibit a ramping pattern prior to the lower limit (**Figure 2c/d**; see below). This observation demonstrates that the observed activity patterns were not triggered by the LL, but reflected intrinsic, context-dependent dynamics that were already present prior to the LL.

Reviewer comment #4:

I am unsure why this is brought up here. Yes, this (and Fig 3a) shows that there is a context-dependent ramping in the PFC prior to HLL, but it does not show that there is no go-cue-triggered change, nor does it address the issue highlighted about the decoder using different periods.

Our response to #4:

Our intention was to highlight that single-trial dynamics show context-dependent ramping prior to the lower limit, but do not show a go-cue-triggered change. Visual inspection is certainly subjective and we agree that single-trial examples should not be used to make any strong claims. However, as shown in **Response Figure 1**, the activity prior and after the lower limit did not

significantly differ for trials with 0% uncertainty (in trials with 25% and 75% uncertainty, the activity already ramps up before; cf. Figure 2a & Supplementary Fig. 5). Hence, the single-trial dynamics support our statement and therefore also address the presumed issue highlighted about the decoder using different periods as the lower limit did not trigger substantial changes in activity.

3. We also time-locked the population response (PC1) in PFC and motor cortex to the LL to supplement our findings relative to the button release (**Supplementary Figure 5**). As visualized in **Supplementary Figure 5**, the population activity in PFC already ramps-up and dissociates the different conditions prior to the LL. This observation is also in line with our findings on univariate ramping dynamics (**Figure 2a/b**): No significant ramping was observed in PFC during trials with no uncertainty (0% likelihood), while significant ramping activity was present for trials with a 25% or 75% likelihood of stopping.

Supplementary Figure 5. Context-dependent population dynamics in PFC emerge prior to the HLL. **a**, Population activity in PFC as indexed by the first principal component reveals context-specific activation patterns prior to the onset of the HLL ($\text{sum}(F_{2,32}) = 637.4, P = 0.023$; cluster test). The single-colored horizontal lines show the temporal extent of a significant context-dependent dissociation. Two-colored horizontal lines indicate the temporal extent of significant clusters obtained from pairwise comparisons. **b**, Same as (a), but for motor cortex. Note the similar, context-independent, temporal activation profile across all conditions in motor cortex ($\text{sum}(F_{2,26}) = 251.7, P = 0.073$).

Reviewer comment #5:

This result helps to make the point the authors are trying to make with Figure 5c and d, that PFC is modulated by predictive context. However, to make the point that HLL does not trigger changes in activity the authors would need to show the other PCs (2 and above) to assess the claim that HLL does not trigger a change.

Our response to #5:

We thank the reviewer for this remark. The main reason why we have chosen PC1 here was based on the reviewer's comment #2 where he/she claims that the lower limit triggers a large condition-invariant signal. Hence, we reasoned that if there is such a strong condition-invariant signal ("largest response component in motor cortex [...]", Kaufman et al. 2016), we should observe this signal in the first PC. We followed the reviewer's suggestion and performed the analysis now on multiple PCs (1 to 5). None of the PCs showed an activity change upon the lower limit (**Supplementary Fig. 6**).

Moreover, we computed the first derivative of the signal which reflects the rate of change in activation. If the lower-limit would have triggered strong changes in neural activity, this should be

captured by the first derivative. However, we did not find support of this claim (**Supplementary Fig. 6**). Therefore, there is no statistical evidence to conclude that the lower-limit in this task triggered strong changes in neural activity.

Supplementary Figure 6. Population dynamics quantified by the first 5 principal components do not show a change in activity triggered by the lower limit. **a**, The neural dynamics in PFC as indexed by the principal component time-series (1 to 5) do not show a time-locked change upon the lower limit. **b**, We computed the temporal derivative of the traces in (a). This revealed that there is no strong rate of change in neural activity upon the lower limit.

In sum, this set of findings demonstrates that the LL did not trigger a large activity change. On the contrary, these additional analyses demonstrate that the activity change reflects the context integration process that started prior to reaching the LL.

Reviewer comment #6:

Unfortunately, I don't see the answers as sufficient to demonstrate that LL did not trigger activity changes, which could then confound many of the analyses.

Our response to #6:

We thank the reviewer for the thorough assessment. However, we respectfully disagree. We implemented several additional analyses to address the concerns. We outline all the changes in detail below. However, we would also like to highlight that the current results were obtained in human participants in the context of clinical recordings; hence, only limited recording time was available. We discuss the study limitations in the discussion and are convinced that future work in other model species that do not suffer from the inherent limitations of human intracranial recordings, such as invasive recordings in non-human primates, may address the remaining details.

In sum, our results demonstrate that:

1. There is no significant change upon the lower limit upon the 0% condition (**Response Fig. 1**). The changes for the 25% and 75% conditions occur much earlier with respect to the lower limit (cf. Fig. 2a & Supplementary Fig. 5). If there would be a strong change in activation upon the lower limit, this should be clearly visible across all trials.
2. The study by Kaufman et al. (2016) showed that the condition-invariant signal is the strongest component in motor cortex. While we have no doubts that this is true, the fact that activity in motor cortex ramps up prior to the lower limit (cf. Fig. 2b & Supplementary Fig. 5b) rather supports the notion of preparatory motor activity.
3. We demonstrated that there is no phasic response upon the lower limit using event-related potentials. We also performed additional control analyses showing that the variance does also not increase from pre- to post-lower limit (**Response Fig. 2 & 3**).

4. We showed that none of the PCs shows a clear-cut response upon the lower limit (**Supplementary Fig. 6**).

In addition to the changes to the manuscript in the previous revision, we now add **Supplementary Figure 6** to the manuscript in order to substantiate our claims:

Page 19

*"We additionally time-locked the population activity to the HLL to ensure that context-dependent activity in PFC ramped up and was dissociable prior to the HLL and not a consequence of an activation change triggered by the HLL (**Supplementary Fig. 5/6**)."*

We have added **Supplementary Figure 5** to the supplements and refer to it in the main text:

Page 19, Line 15-18

*"We additionally time-locked the population activity to the HLL to ensure that context-dependent activity in PFC ramped up and was dissociable prior to the HLL and not a consequence of an activation change triggered by the HLL (**Supplementary Fig. 5**)."*

Next, we addressed the second concern that was voiced in the first query.

"The issue with the action decoder is that it captures different periods of the task for different categories (pre-LL for fast RTs, post-LL for slow RTs and a mixture for the mid RTs). To carry out the intended analysis the authors would need to match the RTs for the different contexts before comparing the activities. Unfortunately, this observation renders the rest of the analyses using this action space uninterpretable."

We performed the analysis as suggested by the reviewer as well as another additional control analysis to preclude that the action decoder is confounded by capturing different periods of the task. We will separately outline the two analyses:

1. We agree that the action decoder captures different periods of the task for different categories. However, the lower limit did not trigger large activity changes as outlined above. Therefore, it is unlikely that the action decoder is confounded by capturing different periods of the task. However, in order to exclude this possibility, we time-locked the data to the LL (which serves as the 'go' cue) and trained a linear decoder to discriminate reaction times (action). Above-chance decoding prior to the LL indicates that the decoder captures processes involved in the planning of action. As shown in **Supplementary Figure 6**, a linear decoder can successfully discriminate the reaction times prior to the lower limit. This finding indicates that action plans were initiated prior to reaching the lower limit.

Supplementary Figure 6. Action-specific information emerges prior to the HLL. **a**, Grand average decoding accuracy (mean \pm SEM) for action within PFC when locked to the HLL. Horizontal lines indicate the extent of significant temporal clusters. **b**, same as **(a)** but for motor cortex. Same conventions as in **(a)**.

We have added **Supplementary Figure 6** to the supplements. We have further added an additional paragraph of this control analysis for clarification:

Page 21, Line 9-19

*"We performed an additional control analysis to preclude that the action-decoding was confounded by capturing different periods of the task (i.e., pre-HLL for fast RTs, post-HLL for slow RTs) when locked to the BR. Therefore, we additionally time-locked the population response to the HLL and repeated the classification analysis. In this case, the linear decoder captures the same periods of the task regardless of the respective RT. This analysis revealed a build-up of action-specific information prior to the HLL (**Supplementary Fig. 6**; PFC: $\text{sum}(t_{13}) = 1074.87$, $P < 0.001$, Cohen's $d = 1.69$; Motor cortex: $\text{sum}(t_9) = 322.79$, $P < 0.001$, Cohen's $d = 1.67$; cluster test), therefore, supporting the notion that the decoder captured action-planning processes."*

Reviewer comment #7:

Unfortunately, this analysis does not show what the authors intended to show. As it is shown in response figure 3 (copy-pasted below), RTs included in the analysis include a many trials in which subjects responded faster than 100ms after the HLL. These reaction times are too fast to be triggered by the go-cue, so these are predictive in nature. Thus, the pre-HLL decoding shown in Suppl Figure 6 is likely reflecting the execution of these very fast movements instead of action-planning processes that preceded HLL.

Our response to #7:

We thank the reviewer for this remark, since this is exactly a key finding that we were highlighting. The reviewer is concerned about the presumed activation change upon the lower limit. Therefore, we sought to test whether action-related information was decodable prior to the lower limit which would speak in favor of action-planning processes. As reported above, we demonstrated that reaction times were successfully decodable prior to the lower limit. The reviewer correctly remarks that there are *"many trials in which subjects responded faster than 100ms after the HLL"* and that these reaction times are *"too fast to be triggered by the go-cue"* and that these reaction times are *"predictive in nature"*.

At this point, we would like to emphasize that this is a key finding of our study and the main reason that the experiment was designed as such. In trials with zero uncertainty, participants could already prepare the movement and respond once the bar reached the lower limit; hence resulting in very fast reaction times which are, as correctly remarked, *"predictive in nature"*.

We agree that the decoding might capture direct action, rather than solely action-planning processes. However, this is also not critical for the main message that the current paper conveys; namely that contextual information is integrated in the human prefrontal cortex by means of a ramping processing mode whereas the inter-areal communication to guide context-dependent action is facilitated by means of an oscillatory communication mode. In the current study, we won't be able to fully disentangle preparatory signatures from direct motor commands as the transition between the two reflects a continuum. However, we would like to emphasize that the classifier was trained to discriminate fast, medium and slow reaction times. If the classifier would only decode the execution of very fast movements (33% of all reaction times), then the classifier should fail to predict reaction times above chance. We would like to emphasize that we already acknowledged the limitation, that we cannot fully separate action-planning from movement-execution based on the current experimental design, in the previous round of revision.

Page 31

"Furthermore, information about action only culminated after the movement onset. Thus, we cannot preclude that these processes mainly capture post-movement, rather than preparatory dynamics. However, we observed a clear build-up of action-specific information (cf. Fig. 5g). This suggests that the observed dynamics track the internal transition from planning to the final movement execution, in line with a recent human iEEG-study²."

Page 20

"Thus, any neural activity that dissociates reaction times prior to the actual movement execution (and that is not primarily driven by early context-integration signals) reflects the internal transition from the planning to the execution of a movement."

Page 22

"Although decoding accuracy for action reached its maximum right after the movement onset, we observed a clear build-up of action-related information over time, suggesting that the decoder captures the direct transition between action-planning and action-execution."

2. Finally, we implemented the proposed analysis by the reviewer and trained and tested the decoder on a subset of trials with matched RTs (time-locked to the HLL). Note this analysis was only feasible for the 25% context condition where a sufficient number of trials was available. As shown in **Response Figure 2**, a linear decoder can reliably discriminate the onset of movement with matched RTs (PFC: first cluster, $\text{sum}(t_{13}) = 483.98$, $P = 0.008$, Cohen's $d = 1$; second cluster, $\text{sum}(t_{13}) = 241.98$, $P = 0.037$, Cohen's $d = 1.52$; Motor cortex: $\text{sum}(t_9) = 246.039$, $P = 0.015$, Cohen's $d = 0.94$; cluster test), indicating that the decoder is not confounded by possibly capturing distinct periods of the task.

Response Figure 2. Context-dependent decoding of action. **a**, Grand average decoding accuracy for action in PFC (mean \pm SEM) using only trials with a 25% likelihood. Horizontal lines indicate the extent of significant temporal clusters. **b**, same as **(a)** but for motor cortex.

Reviewer comment #8:

Unfortunately, this is not the analysis I was suggesting. I am not even sure what "matching RTs" means if only one context was included (25%). What I meant was match RTs across contexts. The manuscript is making claims about the timing of information about context and action (RT), and the handoff of information. The proper analysis would need all 3 contexts RT-matched, so we could see the information about context and RT, like in Figure 5g.

Our response to #8:

We apologize for misunderstanding the reviewer's suggestion. However, the reviewer's main concern all along is that the action decoder is confounded by capturing distinct periods of the task. While we are convinced that we have provided ample evidence that the lower limit does not trigger substantial activity changes, we tested this notion in another complementary way.

We only used the reaction times (RTs) within the 25% condition in order to reduce the variance of the RT-distribution. This minimizes the uncertainty that the action-decoder simply picks-up different periods of the task since the width of the distribution is narrowed. We then demonstrated that the classifier is still able to successfully predict reaction times even within one context condition only, clearly demonstrating that the decoder captures action- and not context-specific activity (see also first round of revision, reviewer #2, comment #6 where we show that we cannot decode action from the 'context-subspace' or vice versa). Moreover, if the action decoder would be fully driven by simply capturing 'different time periods of the task', then the same decoding traces should have been emerged for context-decoding since reaction times strongly differ between context conditions. This is not the case (cf. Fig. 5g; Supplementary Fig. 9) and has also been previously addressed in the first round of revision (reviewer #2, comment #6).

We fully agree that the reviewer's suggestion is an excellent idea. However, this is unfortunately not possible simply due to the fact that reaction times strongly differ between context conditions (the main hypothesis of the manuscript) and the resulting number of trials that could be used for the analysis would be too low. This is a direct consequence of clinical recording settings and an inherent limitation of human intracranial recordings. We have added this limitation into the main text:

Page 34-35

"However, it is worth noting that we cannot rule out some alternative hypotheses due to inherent limitations of human intracranial recordings and the currently employed experimental design. First, due to the inherent limited coverage in intracranial recordings, we cannot preclude that

*specialized sub-regions within the human motor cortex (e.g. anterior or posterior parts of the supplementary motor area, frontal eye fields, premotor cortex) might also encode contextual information. Second, based on the current experimental design, we cannot fully dissociate between preparatory- and movement-related computations. This is a direct consequence of the fast-paced task where the transition between preparation and movement reflects a continuum. Furthermore, while we did not employ a designated go-cue in our experimental paradigm, the lower limit (cf. **Fig. 1a**) might still resemble a go-cue and could potentially trigger condition-invariant activity changes. Consequently, based on the current experimental paradigm, we cannot fully disentangle neural activity that reflects context-dependent processing from neural activity potentially triggered by the lower limit. However, the fact that context-dependent dynamics already evolved (cf. **Fig. 2a & Supplementary Fig. 5a**) and neural dynamics mainly ramped-up prior to lower limit suggests that the observed neural activity patterns in PFC were not solely triggered by the lower limit, but reflected intrinsic, context-dependent dynamics prior to the lower limit. Finally, prefrontal population activity is high-dimensional in nature, where different operations are encoded in distinct subspaces. Yet, our analytical approach does not allow to draw any inference on the dimensionality of the decoding latent space, only on the overall dimensionality of the neural data. However, our findings strongly support the notion that the high-dimensional prefrontal functional architecture constitutes a substrate for flexible goal-directed behavior and that simultaneous processing in separate coding dimensions maximizes information-coding capacity of the underlying population³⁻⁵.”*

Collectively, this set of finding demonstrates that action decoding is not driven by the inclusion of slightly different task periods. On the contrary, these analyses reveal that action information was present prior to the HLL, exhibited a gradual build-up and hence, most likely capturing action-planning processes.

These considerations are now also reflected in the discussion:

Page 31, Line 7-11

*“However, we observed a build-up of action-specific information (cf. **Figure 5g**) that emerged prior to the HLL (**Supplementary Fig. 6**). This suggests that the observed dynamics track the internal transition from planning to the final movement execution, in line with a recent human iEEG-study².”*

Reviewer comment #9:

Given the issues raised above, I still think that the results presented have serious confounds, and my original comment stands.

Our response to #9:

We are thankful for the reviewer’s constructive comments and the time taken to review the manuscript. The concerns were valid and needed to be addressed. However, based on the additional analyses, we are convinced that the presented evidence demonstrates that there is no confound with respect to the action decoder, since the lower limit did not trigger substantial changes in activity. Hence, even though the decoder picks up slightly different periods with respect to the lower limit, it does not confound the analysis. However, since the recordings were obtained in the clinical context, we now discuss the inherent study limitations in the discussion. In sum, we are convinced that the results provide an important contribution, since the demonstrate that population-based context and action coding (as previously observed in model species, such as non-human primates) are also evident in the human brain.

2- Line 366: "This indicates that PFC, but not motor cortex, is gradually recruited as a function of behavioral uncertainty (Fig. 5e)." I don't think your results should be interpreted this way. When aligned to button release (as in Fig5), the differences begin emerging around 0.2s before movement, which happens to coincide with the LL (lower limit) onset for the slowest RT trials. If PFC was gradually recruited, these differences should start emerging from the moment that the contextual cue is given (560-750ms before the LL). But that is not the case. Go-cues (like the LL) trigger a condition-invariant signal, or CIS (Kaufman et al. 2016, Guo et al. 2014, Inagaki et al. 2022). In light of the CIS, averaging trials, aligned to movement onset, could lead to the exact figure shown. So this analysis should not be interpreted as a gradual increase.

We thank the reviewer for this relevant query and bringing these papers to our attention. We believe that our description of the task structure caused this misunderstanding. We therefore briefly clarify the task structure again before we discuss our data in light of the CIS.

1. Task structure: Each trial started with a brief baseline period (500ms) followed by a contextual cue presented at the center of the screen (green, orange or red indicating the likelihood of a premature stop). Importantly, after receiving the cue, participants self-paced the trial start (average time to start the trial: $1.76s \pm 0.55s$; mean \pm SD). By pressing the space bar, the target would start moving upwards and reach the LL after 560 – 580ms. As a consequence, the time between the contextual cue and the moment the target reaches the LL could vary from trial to trial. Once the target reached the LL, participants were instructed to release the button. We have added this information into the methods section:

Page 38, Line 1-4

"Upon receiving the predictive cue, participants were able to start the trial in a self-paced manner by pressing the space bar on the keyboard (average time to start the trial: $1.76s \pm 0.55s$; mean \pm SD)."

Reviewer comment #10:

This was clear to me from the original description. I'm not sure why the authors thought that there was a misunderstanding on this based on my comment above.

Our response to #10:

We thought there was a misunderstanding because the reviewer stated that: *"If the PFC was gradually recruited, these differences should start emerging from the moment that the contextual cue is given (560-750ms before the LL)."* This statement is incorrect and that is why we thought there was a misunderstanding.

As mentioned above, after receiving the contextual cue, participants self-paced the trial start. Therefore, the contextual cue is not necessarily given 560-760ms prior to the lower limit, but as stated above: *"Upon receiving the predictive cue, participants were able to start the trial in a self-paced manner by pressing the space bar on the keyboard (average time to start the trial: $1.76s \pm 0.55s$; mean \pm SD)."* Hence, it is not surprising to see that the effects did not directly emerge upon the moment the participants pressed the button and self-started the trial, since the time between the appearance of the contextual cue and the self-paced start was variable.

2. As correctly remarked, the context-dependent dissociation – when locked to the button release – emerges ~0.2s prior to the actual movement onset. This approximately coincides with the lower limit. However, as can be seen in **Figure 1c**, reaction times strongly vary from trial to trial. Therefore, as depicted in **Response Figure 3**, the time between the movement onset and the moment the target reaches the LL is variable and jittered across trials. This temporal jitter renders it unlikely that our results can be fully explained by a change in neural activity that is time-locked to the LL.

Response Figure 3. Movement onset distribution with respect to HLL. The distribution depicts the movement onset timing relative to the HLL. The vertical dashed line reflects the average movement onset time. Note that the timing is jittered across trials.

Reviewer comment #11:

I have addressed this point about HLL-triggered activity thoroughly above. However, this depiction of RT raises even more doubts. It would be useful to show the RT distribution for the different contexts. I imagine that the 0% accounts for most of the <500ms RTs shown here, whereas the 75% context probably has none of these. This asymmetry would of course be very problematic for all the analyses carried out.

More importantly, I am not sure how this observation (that there is RT variability) has any bearing on my comment about contextual cues having to emerge much earlier (the moment the cues are given).

Our response to #11:

First, we do not understand why this depiction should raise any doubts as it is simply a different depiction of Figure 1c. The reviewer asks for an RT distribution for the different contexts. This is what we have depicted in Figure 1c and clearly described in the results section: “*We found that reaction times (RT) gradually increased as a function of uncertainty (Fig. 1c).*” This was in fact the main manipulation of the task. We manipulated the uncertainty by providing different probabilities with which the bar would prematurely stop prior to the lower limit. Hence, in the 0% condition, there was no uncertainty and the participants could already prepare the movement, resulting in fast reaction times. In contrast, when participants were provided with the orange or red

cue (25% and 75% probability of stop, respectively), reaction times were slower because the bar could still prematurely stop. Second, this observation has an important bearing on the following reviewer's comment:

"This indicates that PFC, but not motor cortex, is gradually recruited as a function of behavioral uncertainty (Fig. 5e)." I don't think your results should be interpreted this way. When aligned to button release (as in Fig5), the differences begin emerging around 0.2s before movement, which happens to coincide with the LL (lower limit) onset for the slowest RT trials."

The reviewer's argument here is that the differences in population activity that we observe between the different context-conditions are driven by the presumed activity change upon the lower limit. While we have already addressed this point thoroughly above, the variability of RTs renders it even more unlikely that Fig. 5c is driven by activity changes upon the lower limit. If it was true that the results are a simple consequence of activity changes upon the lower limit, one would have expected a temporal lag in the responses shown in Fig. 5c simply due to different RTs between context-conditions (i.e. activity in 75% condition increasing first, then 25% condition, then 0% condition because the distance to the lower limit decreases).

Reviewer comment #12:

I also just realized that Fig 1c above is not actually showing RTs, but rather something else (in the order of 700ms... maybe time from context cue to response?)

Our response to #12:

We thank the reviewer for this remark and thank the reviewer for highlighting this inconsistency. Figure 1c shows RTs but it reflects the time between the self-paced trial start and the reaction. We changed Figure 1c to reflect the time between the lower limit and the reaction times.

3. We also time-locked the population activity to the LL (see comment #1; **Supplementary Figure 5**). This analysis revealed that context-dependent activity already emerged $\sim 0.2s$ prior to the LL in PFC. This finding supports the univariate analyses where context-dependent activity is evident prior to the LL (**Figure 2a**; **Figure 3a**). In sum, this set of findings demonstrates that the LL did not trigger a large activity change. Instead, these findings highlight that the activity change reflects contextual processing that started prior to reaching the LL.

Collectively, these results are in line with a context-dependent activation in PFC, that is not evident in motor cortex. We carefully considered the suggested papers, in particular Kaufman et al.¹. CIS reflects a large and rapid change prior to the movement onset in dorsal premotor regions as well as in M1 that reflects more variance than condition-specific (“tuned”) responses. The notion of CIS reflecting the internal transition from motor preparation to movement onset in motor cortex is fully in line with our findings. In the present dataset in humans, the principal motor component does not exhibit context-dependent activity. Therefore, we believe that our findings nicely complement and extend similar findings previously obtained in non-human primates. We now discuss our findings in light of condition-invariant signals in motor cortex:

Page 32, Line 3-14

"Neural activity in motor cortex revealed large, context-independent activity changes prior to the movement onset. This context-independent activity, previously also referred to as a condition-invariant signal (CIS)², typically reflects the largest response component in motor cortex². The present findings are compatible with a CIS in human motor cortex. The first principal component of the HFA-signal in motor cortex is (1) context-independent and (2) explaining the largest variance. An unresolved question is whether other areas might drive this sudden change in motor cortex activity. Previous studies have identified a large-scale network that might provide input to motor cortex, including subcortical^{6,7} and cortical structures^{8,9}. The present findings demonstrate that human PFC also modulates neural activity in motor cortex."

In contrast to motor cortex activity, PFC population activity was strongly context-dependent (**Figure 5c; Supplementary Figure 5**). Therefore, the population activity in PFC cannot be explained by a CIS. We nuanced our phrasing of a gradual recruitment in PFC:

Page 19, Line 12-15

"This set of findings indicates that population activity in PFC is modulated by predictive context. In contrast, motor cortex population activity displayed context-invariant dynamics supporting a condition-invariant signal² (see Discussion)."

3- LDA analysis: This is a very unconventional way to carry out this analysis. The decoding latent space could be a high-dimensional space that spans multiple PCs. So normally these analyses are done on all dimensions at once (or all dimensions that explain more than 95% of variance), and not one PC dimension at a time, as was done here. I think that this analysis, as carried out, is not very meaningful. If the goal was to enforce orthogonality, you can impose this constraint in the higher-dimensional LDA spaces (although I'm not sure why you'd want to impose orthogonality).

We thank the reviewer for the comment and the excellent suggestion to run the analysis by using the full, high-dimensional space. We separate the response in two parts. First, we would like to clarify the rationale behind our analysis approach. Subsequently, we outline the results of the analysis when performed in high-dimensional space as suggested by the reviewer.

Rationale behind the analysis:

1. Using univariate analyses (**Figure 1-4**), we established that ramping, but not oscillatory dynamics are modulated by predictive context. Instead, oscillatory

dynamics primarily supported the interplay between prefrontal and motor cortex. These findings then guided our multivariate analyses where we (1) examined the functional roles of oscillatory and ramping dynamics and (2) identified the population-level interaction between the prefrontal and motor cortices. Therefore, we first established low-dimensional subspaces that could dissociate contextual from action-specific processing. This approach subsequently allowed us to then test whether the principal dimension with the strongest oscillatory power would (1) encode aspects of contextual processing or action-planning and (2) would temporally synchronize the prefrontal-motor network. In order to mitigate the different orientation of the multidimensional space of prefrontal and motor cortex, we constrained our analysis to the relevant one-dimensional subspace. As indicated by the reviewer, this might seem unconventional, however, it provides the basis to study information transfer between the relevant coding dimensions and to understand the role of oscillatory dynamics in the transfer of information on the population-level. We conducted several control analyses (i.e., decoding in high-dimensional space as suggested by the reviewer) to demonstrate that the relevant coding dimension is well approximated by a single dimension and that adding more dimensions does not necessarily improve the decoding performance.

2. Another reason that led us to constrain our decoding analysis to a low-dimensional subspace was the heterogenous number of electrodes implanted in different patients. In turn, this would have resulted in a largely heterogenous number of PCs that define a high-dimensional space. This is unfortunately an inherent limitation of intracranial recordings in humans.
3. However, we fully agree with the reviewer that the decoding latent space itself could be high-dimensional and thus span multiple PCs. Therefore, in the revised version of the manuscript, we now explicitly emphasize that our decoding analysis does not allow any inference regarding the dimensionality of the decoding latent space.

Page 34, Line 10-13

"Importantly, prefrontal population activity is high-dimensional in nature, where different operations are encoded in distinct subspaces. Yet, our analytical approach does not allow to draw any inference on the dimensionality of the decoding latent space, only on the overall dimensionality of the neural data."

Reviewer comment #13:

Noted on the rationale, and the new analysis. It makes sense.

Our response to #13:

Thanks.

Additional analyses:

Finally, we followed the reviewer's suggestion and performed the analysis using all principal components cumulatively explaining more than 95% of the variance. This revealed comparable effects (**Supplementary Figure 8**) supporting our observation that contextual processing is PFC-dependent whereas action-related processing occurs in both PFC and motor cortex.

Supplementary Figure 8. Decoding analysis of context and action in high-dimensional space. **a**, Grand average decoding accuracy (mean \pm SEM) for context and action in PFC. Classification was performed using all principal components cumulatively explaining more than 95% of the variance. Horizontal lines indicate the extent of significant temporal clusters (color coded for the respective feature). **b**, Grand average decoding accuracy (mean \pm SEM) for context and action in motor cortex. Only action, but not context, could be decoded in motor cortex. Same conventions as in (a).

We have implemented these results into the supplements and directly refer to and discuss them in the main text.

Page 21, Line 19 – Page 22, Line 8

"We performed additional control analyses demonstrating that the context-encoding and action-encoding subspaces capture dissociable processes related to contextual computations and the planning of actions (Supplementary Fig. 7; Methods). We also performed the decoding analyses using a high-dimensional neural space on all principal components cumulatively explaining more than 95% of the variance. This replicated our main findings that the human PFC encodes both context and action (Supplementary Fig. 8; context: first cluster, $\text{sum}(t_{16}) = 408.68$, $P = 0.002$, Cohen's $d = 0.74$; second cluster, $\text{sum}(t_{16}) = 151.62$, $P = 0.039$, Cohen's $d = 0.69$; action: $\text{sum}(t_{16}) = 1040.76$, $P < 0.001$, Cohen's $d = 1.08$; cluster test) whereas motor cortex only encodes action (Supplementary Fig. 8; context: first cluster, $\text{sum}(t_{13}) = 39.46$, $P = 0.427$, Cohen's $d = 0.66$; action: $\text{sum}(t_{13}) = 828.57$, $P < 0.001$, Cohen's $d = 1.11$; cluster test)."

To directly address the similarity between the decoding traces obtained from a high-dimensional vs. one-dimensional set, we adopted an analysis by Chiang et al.¹⁰ to find the optimal ensemble size for a decoder. We sorted the individual PCs according to their decoding accuracy and then iteratively added the sorted PCs to train the decoder. This analysis revealed that a high-dimensional space does not improve classification performance (Response Figure 4). Overall, the correlation between the decoding time series using multiple PCs and the decoding time series using only the most informative PC showed that the time courses are largely correlated (Response Figure 5).

Response Figure 4. Comparison of decoding performance in low- vs. high-dimensional space. We tested the optimal dimensionality for the linear classifier to decode context as well as action. We sorted individual PCs according to their decoding performance and then iteratively added the sorted PCs to the decoder. This revealed that decoding performance did not benefit from adding more (uninformative) components. Single-subject example, *Upper Left*: Depicts the magnitude of the decoding cluster as a function of the number of sorted PCs to train the decoder on context. *Lower Left*: Time-resolved context-decoding accuracies computed on distinct numbers of PCs. Note the strong temporal similarity of the decoding traces. *Upper Right*: Depicts the magnitude of the decoding cluster as a function of the number of sorted PCs to train the decoder on action. *Lower Left*: Time-resolved action-decoding accuracies computed on distinct numbers of PCs.

Response Figure 5. Group-level correlation between decoding traces resulting from quantification in low vs. high-dimensional spaces. We quantified the correlation between individual decoding traces (*left and middle panel*) to test their similarity. We observed a strong temporal correlation on the group-level (*right panel*) between the decoding traces independent of the number of PCs that were used to train the linear decoder.

Collectively, this set of findings demonstrates that our decoding analyses yields valid and interpretable results that replicated using a high-dimensional set of components.

Reviewer comment #14:

These analyses indeed do show that this information lives in low-D spaces, and support the authors claim.

Our response to #14:

We would like to thank the reviewer again for the suggestion to perform this analysis in high-dimensional space as well, which strongly substantiated our claims.

4- Fig 5i shows that most action peaks occur post-movement (BR), so these cannot be causal in the movement generation nor planning. As such this “serial unfolding” claim

does not really rest in solid grounds. This also motivates pg 21: “This raises the question, whether and how the representation of the action plan in PFC is passed on to motor cortex for final action execution?”, which loses force when we realize the post-movement nature of these sequential signals.

We thank the reviewer for the comment. We would like to clarify that we did not intend to draw any conclusions or make any claims regarding causality – neither with respect to movement generation nor movement planning. We apologize if our wording was not clear-cut. To avoid further confusion on data interpretation, we now explicitly state that we cannot draw any causal inference on the directional flow of information from movement planning to movement generation.

Page 31, Line 2-5

"However, due to the correlative nature of our study we cannot draw any causal inference or directionality nor exclude the possibility that the observed effects could be driven by a third structure that we did not record from."

We would like to clarify that the serial unfolding claim only referred to the temporal segregation between context-integration in PFC and action-planning in both PFC and motor cortex. It did not refer to the temporal dynamics of action-related processes between PFC and motor cortex. In the revised version of the manuscript, we now make this explicit.

Page 22, Line 8-18

*"Having established that context and action can be decoded from the prefrontal-motor network, we next quantified their temporal dynamics by computing the peak-decoding latency. Peak-decoding accuracy revealed a distinct temporal pattern with initial information about contextual information in PFC (**Supplementary Fig. 9**; $P = 0.024$; Kruskal-Wallis; $-0.09 \pm 0.19s.$ with respect to BR; mean \pm SD), followed by action-related processes in PFC ($0.05 \pm 0.15s.$) and motor cortex ($0.11 \pm 0.12s.$), suggesting that contextual computations in PFC are completed prior to action-relevant computations. Although decoding accuracy for action reached its maximum right after the movement onset, we observed a clear build-up of action-related information over time, suggesting that this the decoder maps information with respect to action-planning."*

However, we fully concur that it might be counterintuitive that the maximum decoding performance for action only culminates just after the movement onset. Yet, considering the human iEEG literature, this is not unexpected. For example, Haller et al.¹¹ observed that the high-frequency activity in motor cortex peaks after the movement onset. Our results (**Figure 2b/5d**) substantiate this observation. Furthermore, Ter Wal et al.² used linear classifier on human iEEG data to decode left vs. right movement. The authors showed that decoding performance about the movement culminates after the movement onset.

There are at least two potential reasons for this observation:

- 1) As the computation of the HFA involves a filter in the temporal domain (i.e. from 70-150 Hz in non-overlapping 10 Hz wide bins), the temporal precision of the data is inevitably reduced as compared to single unit data, a necessary limitation of human intracranial recordings. As a direct consequence, the temporal uncertainty with respect to minima and maxima is increased.

- 2) Figure 2b/c in Ter Wal et al.² shows the decoding performance with respect to left vs. right movement. The decoding trace clearly builds up over time and yields sudden changes $\sim 0.2s$ prior to the movement onset. This fully matches our findings (**Figure 5g/h, left panels**) that action can be successfully decoded $\sim 0.2s$ prior to the BR. This demonstrates that the linear decoder captures the transition from movement planning to movement execution.

Reviewer comment #15:

To be honest, I was not perplexed by the post-movement decoding peak. This is observed over and over in many brain regions. What I was raising is the use of this metric as a means to talk about the serial unfolding, or any temporal metric. If anything, I would have expected the time of first significant decoding time-point as the relevant metric.

Our response to #15:

We apologize, but this was not well-phrased. The reviewer clearly stated that "*Fig 5j shows that most action peaks occur post-movement (BR), so these cannot be causal in the movement generation nor planning*". Therefore, we argued that the time series of decoding accuracy can still capture action-planning processes even though they culminate post-movement. Furthermore, as mentioned above, the 'serial-unfolding' referred to the temporal segregation between context-integration and action-related processes. Figure 5g and Supplementary Figure 8 demonstrate that successful decoding of context precedes successful decoding of action. We interpreted this as a 'serial unfolding' from context to action. To further substantiate this, we computed the peak-decoding accuracies and demonstrated that contextual integration precedes to action-related processes (Supplementary Figure 9). Nevertheless, we fully agree that we cannot entirely preclude that the processes we capture might reflect post-movement, rather than preparatory dynamics and therefore added this as a limitation in the main manuscript (see below).

We now discuss this finding in more detail in the discussion:

Page 31, Line 5-11

"Furthermore, information about action only culminated after the movement onset. Thus, we cannot preclude that these processes mainly capture post-movement, rather than preparatory dynamics. However, we observed a clear build-up of action-specific information (cf. Figure 5g). This suggests that the observed dynamics track the internal transition from planning to the final movement execution, in line with a recent human iEEG-study²."

With respect to the "information hand-off" of action plans, we initially based our interpretation on (1) the phase-slope-index (**Figure 4h**) as well as (2) the temporal information of the average action-decoding traces (**Figure 5g/h; left panel**). In the previous revision of this manuscript, we further added the cross-correlation analysis, peak-decoding analysis as well as the cross-regional generalization analysis (see comment #5) to substantiate our interpretation.

Reviewer comment #16:

Maybe this is discussed later, but as of now I see no reason to see any of these results as "information hand-off"

Our response to #16:

We fully concur and elaborate on this issue below. We omitted the term from the revised manuscript.

5- The methods description of the cross-regional pattern analysis is not clear from the description of the methods and results section. I'm not sure how you can test classification using different sets of electrodes that were not used in the training of the LDA. As such, I cannot assess fig 5j and the "information handoff" claim.

We thank the reviewer for bringing this up and apologize for the lack of clarify in our description. As pointed out in our response to comment #4, we performed the cross-regional generalization analysis to directly assess the information hand-off between prefrontal and motor cortex.

The reviewer is entirely correct that this analysis would be impossible on different sets of electrodes. As outlined in our response above, we mitigated this problem by focusing on the relevant coding dimension in both regions.

Thus, we did not train and test the classifier on electrodes as features, but instead tested whether the neural information in the PFC action subspace generalizes to the motor cortex action subspace. We thereby tested if a discriminative (informative) pattern that was present in PFC was equally present in motor cortex.

The selection of the coding subspace enabled this analysis, which otherwise would have been impossible as correctly remarked by the reviewer. In the revised version of the manuscript, we made this more explicit in the results section and have further added details to the methods description.

Page 23, Line 7-11

"We trained a linear classifier on the activity of the PFC action subspace to discriminate reaction times and subsequently tested if the classifier generalized to the action subspace in motor cortex. Hence, we tested if a discriminative pattern that was present in the action-specific subspace in PFC was equally present in the action-specific subspace in motor cortex."

Page 51, Line 10-17

"Cross-regional pattern analysis. To quantify whether a discriminative action-specific pattern present in PFC is equally present in motor cortex, we trained a linear classifier on every time point in the action subspace (principal component maximally discriminating action) in PFC and subsequently applied it on every time point in the action subspace in motor cortex. Cross-validation was not necessary since training and testing datasets were independent. Finally, classification values were tested against chance level and corrected for multiple comparison using cluster-based permutation statistics."

Reviewer comment #17:

Thank you. This makes more sense (in terms of methods). However, I am still unclear on how this analysis supports information hand-off.

Our response to #17:

Thank you for having us provided the opportunity to better clarify the rationale behind this analysis. With respect to the 'information hand-off': As mentioned previously, we based our claim that there is an information hand-off from PFC to motor cortex upon three findings. (1) The phase-slope index analysis, which is a measure of directional connectivity, revealed that there is a context-dependent directional interaction from PFC to motor cortex. (2) The temporal dynamics of the decoding accuracies revealed that context could be successfully decoded in PFC prior to action in both PFC and motor cortex. Hence, we interpreted this as an 'information hand-off' from PFC to motor cortex. (3) We then aimed to test whether there is also an 'hand-off' of

action-specific information and therefore employed the cross-regional pattern analysis. This analysis revealed that that a discriminative pattern that was present in the action-specific subspace in PFC was equally present in the action-specific subspace in motor cortex, but with a temporal lag, suggesting that this information initially emerged in PFC and was passed onto motor cortex.

However, we realized that this interpretation is clearly subject to debate and hence nuanced our language and removed claims regarding the 'information hand-off'.

Page 4

*"We describe a striking functional dissociation between population activity and network oscillations where human PFC encodes predictive context and the current action plan in distinct subspaces using a continuous processing regime, **while theta oscillations mediate the inter-areal communication between PFC and motor cortex to guide context-dependent actions.**"*

Page 23

"Finally, to infer whether action-related information generalizes from PFC to motor cortex prior, we conducted a cross-regional pattern analysis (Methods)."

Page 23-24

"In sum, this set of findings demonstrates that the human PFC integrates contextual information at early stages using a low-dimensional subspace and devises them into an appropriate action plan that can be read-out by motor cortex for final action-execution."

Page 27

"[...] indicating a functional role of theta oscillations to mediate the cross-regional generalization of action plans from prefrontal to motor cortex."

Page 29

"Furthermore, our results reveal that (V) theta synchrony temporally coordinates action-encoding population subspaces, thereby mediating the cross-regional generalization of action plans from prefrontal to motor cortex (Fig. 6)."

Minor Points

1- Fig g and h right are not very useful without stats... the axis spans seem to be chosen to make a particular point rather than to reflect the data's structure.

We concur that the right panels in figure 5g/h were largely descriptive and only served illustrative purposes. We therefore decided to remove them from the manuscript.

Reviewer comment #18:

Thanks.

2- Pg 32: "Here, we replicate this finding in humans, but in contrast to NHPs, we found no evidence that the human motor cortex encodes predictive context." This is not quite true, since the NHP work referenced provides evidence of pre-movement prediction regarding the specific movement (not context). This study only had one movement (button release) so this could not be assessed.

Thank you for bringing this mistake to our attention. We realized that our referencing was misleading. The replication with respect to a low-dimensional neural pattern in motor cortex referred to the following two references: Gallego et al.¹² and Suresh et al.¹³. The statement "[...] but in contrast to NHPs, we found no evidence that the human motor cortex encodes predictive context" instead referred to Thura and Cisek^{14,15}, Wang et al.¹⁶ and Glaser et al.¹⁷. However, after carefully re-reading the referenced studies, we decided to nuance the statement that prior work in NHPs has revealed context-dependent coding in the (pre-)motor network. In particular, we now explicitly state that context-dependent coding has mainly been found in adjacent premotor areas, such as frontal eye fields or dorsal premotor-cortex. We changed the text in several instances:

Page 28, Line 21-23

"Previous work in NHP indicated that adjacent premotor structures, such as frontal eye fields^{5,18} or dorsal premotor cortex^{14,17}, might mediate context context-dependent decision-making."

Page 33, Line 13-17

"While previous evidence in NHP indicated that adjacent premotor structures, such as frontal eye fields^{5,18} or dorsal premotor cortex^{14,17} perform context-dependent computations, we found that neural trajectories in prefrontal, but not motor cortex, dissociated the current predictive context."

Page 33, Line 23 – Page 34, Line 9

"Previous work in NHP demonstrated that motor cortex exhibits a low-dimensional structure^{12,13}. Here, we replicate this finding in humans, but in contrast to prior work in NHPs that has revealed context-dependent computations in adjacent premotor structures^{5,14,17,18}, we found no evidence that the human motor cortex encodes predictive context. We observed that motor cortex relies on input from PFC, which encodes both context as well as the current action plan. However, due to the inherent limited coverage in intracranial recordings, we cannot preclude that specialized sub-regions within the human motor cortex (e.g. anterior or posterior parts of the supplementary motor area, frontal eye fields, premotor cortex) might also encode contextual information."

Reviewer comment #19:

Thanks for the clarifications.

3- The correlation analysis in Figure 5e can also be confounded by the fact that high-uncertainty trials had longer RTs... the differences in correlation values could be a simple reflection of a larger number of points being included in the correlations of longer-RT trials.

Thank you for the opportunity to clarify. We time-locked the data from -0.5 to +0.3s with respect to the button release across all conditions. Therefore, the correlation analysis is not confounded by a difference in the number of time points considered. In the revised manuscript, we now make this point explicit.

Page 52, Line 14-18

"We computed the correlation coefficient between PC single-trials in PFC and motor cortex from -0.5s to 0.3s with respect to movement onset. This ensured that all trials contained an equal number of data samples, thereby avoiding potential confounds in the correlation value simply due to variable reaction times across trials."

Reviewer comment #20:

Noted, thanks.

4- PCA methods: There is not enough information in the methods to understand what was done with the PCA analysis. Were the time bins aligned to movement, or to task events? Were all trials (across all contexts) used to build the space, or were spaces built on context-specific activity? Which time bins were included in the PCA analysis?

We apologize for the lack of clarity on how we performed the PCA. We performed PCA on the movement-locked data (button release). For the revised version of the manuscript, we now also performed PCA on the data time-locked to the HLL (**Supplementary Figure 5**). We used all trials to build the PC-space. We have added this critical information into the methods section on dimensionality reduction.

Page 49, Line 12-19

"We used principal component analysis (PCA) to identify linearly uncorrelated population activity patterns and construct a low-dimensional manifold that is embedded in the neural state space spanned by the recorded depth electrodes. We performed PCA on a two-dimensional data matrix (channel x time, trial) locked to either the HLL or to the movement onset. All trials were used to construct the PC-space. The resulting matrix (component x time, trial) was then reshaped into a three-dimensional matrix (trial x component x time) which allowed us to perform single trial analysis in PC space."

Reviewer comment #21:

Noted, thanks.

5- Answers to reviewers mostly address their issues, except for Reviewer 2, point 1. All those analyses are also consistent with difficulty being the main factor. At the very least this should be discussed as a confound in the paper, rather than brushing it off.

Thank you for this comment. We did not intend to brush the argument off. We agree that it is not possible to entirely and unambiguously disentangle uncertainty from task-difficult, but we performed additional analyses to provide evidence in support of this consideration. We would like to point out that a shift along the internal response axis (criterion) is indicative of an alteration in the participant's decision strategy¹⁹. However, we fully acknowledge that an additional paragraph is needed to address this limitation. In the revised version of the manuscript, we have added a paragraph emphasizing this limitation.

Page 39, Line 6-14

"It is important to acknowledge that the environment of intracranial EEG recordings precludes long experiments with many control conditions. Based on our design, we cannot fully disentangle behavioral uncertainty from overall task difficulty. We directly addressed this limitation by calculating the SDT as well as linear ballistic accumulator models to quantify the participants' response strategy. However, the observed shift in criterion as a function of uncertainty could also be explained by an overall shift of the signal and noise distribution along the internal response axis that would not involve any change in participants' response strategy."

Reviewer comment #22:

Thank you for the addition.

References

- 1 Kaufman, M. T. *et al.* The Largest Response Component in the Motor Cortex Reflects Movement Timing but Not Movement Type. *eNeuro* **3**, doi:10.1523/ENEURO.0085-16.2016 (2016).
- 2 Ter Wal, M. *et al.* Human stereoEEG recordings reveal network dynamics of decision-making in a rule-switching task. *Nat Commun* **11**, 3075, doi:10.1038/s41467-020-16854-w (2020).
- 3 Rigotti, M. *et al.* The importance of mixed selectivity in complex cognitive tasks. *Nature* **497**, 585-590, doi:10.1038/nature12160 (2013).
- 4 Parthasarathy, A. *et al.* Mixed selectivity morphs population codes in prefrontal cortex. *Nat Neurosci* **20**, 1770-1779, doi:10.1038/s41593-017-0003-2 (2017).
- 5 Aoi, M. C., Mante, V. & Pillow, J. W. Prefrontal cortex exhibits multidimensional dynamic encoding during decision-making. *Nat Neurosci* **23**, 1410-1420, doi:10.1038/s41593-020-0696-5 (2020).
- 6 Hauber, W. Involvement of basal ganglia transmitter systems in movement initiation. *Prog Neurobiol* **56**, 507-540, doi:10.1016/s0301-0082(98)00041-0 (1998).
- 7 Philipp, R. & Hoffmann, K. P. Arm movements induced by electrical microstimulation in the superior colliculus of the macaque monkey. *J Neurosci* **34**, 3350-3363, doi:10.1523/JNEUROSCI.0443-13.2014 (2014).
- 8 Pesaran, B., Nelson, M. J. & Andersen, R. A. Free choice activates a decision circuit between frontal and parietal cortex. *Nature* **453**, 406-409, doi:10.1038/nature06849 (2008).
- 9 Murakami, M., Vicente, M. I., Costa, G. M. & Mainen, Z. F. Neural antecedents of self-initiated actions in secondary motor cortex. *Nat Neurosci* **17**, 1574-1582, doi:10.1038/nn.3826 (2014).
- 10 Chiang, F. K., Wallis, J. D. & Rich, E. L. Cognitive strategies shift information from single neurons to populations in prefrontal cortex. *Neuron* **110**, 709-721 e704, doi:10.1016/j.neuron.2021.11.021 (2022).
- 11 Haller, M. *et al.* Persistent neuronal activity in human prefrontal cortex links perception and action. *Nat Hum Behav* **2**, 80-91, doi:10.1038/s41562-017-0267-2 (2018).
- 12 Gallego, J. A., Perich, M. G., Miller, L. E. & Solla, S. A. Neural Manifolds for the Control of Movement. *Neuron* **94**, 978-984, doi:10.1016/j.neuron.2017.05.025 (2017).
- 13 Suresh, A. K. *et al.* Neural population dynamics in motor cortex are different for reach and grasp. *Elife* **9**, doi:10.7554/eLife.58848 (2020).
- 14 Thura, D. & Cisek, P. Deliberation and commitment in the premotor and primary motor cortex during dynamic decision making. *Neuron* **81**, 1401-1416, doi:10.1016/j.neuron.2014.01.031 (2014).
- 15 Thura, D. & Cisek, P. Modulation of Premotor and Primary Motor Cortical Activity during Volitional Adjustments of Speed-Accuracy Trade-Offs. *J Neurosci* **36**, 938-956, doi:10.1523/JNEUROSCI.2230-15.2016 (2016).
- 16 Wang, M. *et al.* Macaque dorsal premotor cortex exhibits decision-related activity only when specific stimulus-response associations are known. *Nat Commun* **10**, 1793, doi:10.1038/s41467-019-09460-y (2019).
- 17 Glaser, J. I., Perich, M. G., Ramkumar, P., Miller, L. E. & Kording, K. P. Population coding of conditional probability distributions in dorsal premotor cortex. *Nat Commun* **9**, 1788, doi:10.1038/s41467-018-04062-6 (2018).

- 18 Mante, V., Sussillo, D., Shenoy, K. V. & Newsome, W. T. Context-dependent computation by recurrent dynamics in prefrontal cortex. *Nature* **503**, 78-84, doi:10.1038/nature12742 (2013).
- 19 lemi, L., Chaumon, M., Crouzet, S. M. & Busch, N. A. Spontaneous Neural Oscillations Bias Perception by Modulating Baseline Excitability. *J Neurosci* **37**, 807-819, doi:10.1523/JNEUROSCI.1432-16.2016 (2017).

Point-by-Point Reply: NCOMMS-22-05636B

We would like to thank the reviewers for their thoughtful and constructive comments. Below, we provide point-by-point responses to all queries. All changes in the revised manuscript are highlighted with tracked changes.

Reply formatting guide:

Reviewer remarks revision #2: **bold, black**.

Author response revision #2: *blue*.

Changes in the manuscript revision #2: *blue italics*.

Revision #3:

We highlight the additional comments of reviewer 3, that were embedded in our second rebuttal letter, in *red*. We now address these comments within the respective context (highlighted in *green*).

Conversely, changes in the manuscript are now highlighted in *green italics*.

Reviewer #3 comments added in purple

Reviewer #1 (Remarks to the Authors)

The authors have done an excellent job in addressing my concerns. I have no further comments and congratulate the authors on a splendid paper.

We would like to thank the reviewer for the constructive review that helped to improve the manuscript.

Reviewer #2 (Remarks to the Authors)

I would like to thank the authors for addressing all of my comments. I agree with their responses and have no further comments on the revised manuscript. This is an impressive work that thoroughly investigated the activity and communication of PFC and motor cortex in their task with *iEEG* data.

Regarding comment 7 (orthogonality of context and action encoding), what I suggested is to find axes in neural space that maximally discriminate context and action independently (without relating them to PCs, which are granted to be orthogonal) and then compute their angle in neural space. But I agree that orthogonality is not a necessary claim, and I am satisfied with the authors' revision.

We would like to thank the reviewer for the valuable and insightful feedback that substantially improved the manuscript.

Reviewer #3 (Remarks to the Authors)

I think that the results from Figs 1-4 are solid, and the authors did an excellent job at addressing the reviewers' comments on these sections. However, Figs 5-6 have some serious methodological flaws that render them either hard to interpret, or just not valid.

We would like to thank the reviewer for the insightful and constructive comments on our manuscript and the detailed queries regarding figures 5 & 6. The key criticisms are a direct result of either imprecise wording or an insufficient explanation of the rationale and methods. We now provide a more detailed explanation of the analytical approach and the employed methods. In addition, we conducted several additional control analyses, which further substantiated our findings.

Major Points

1- The LL appears to trigger a large activity change. Uncertain trials have longer RTs. So the trajectory analyses could be affected by this difference (as pointed out by Reviewer 2 and acknowledged by the authors). The issue with the action decoder is that it captures different periods of the task for different categories (pre-LL for fast RTs, post-LL for slow RTs and a mixture for the mid RTs). To carry out the intended analysis the authors would need to match the RTs for the different contexts before comparing the activities. Unfortunately, this observation renders the rest of the analyses using this action space uninterpretable.

We would like to thank the reviewer for bringing up this point. We would like to divide our responses to query 1 into two sections to separately address the query on (1) large activity changes upon the lower limit (LL or HLL) and (2) the concern regarding the action decoder.

"The LL appears to trigger a large activity change. Uncertain trials have longer RTs. So the trajectory analyses could be affected by this difference (as pointed out by Reviewer 2 and acknowledged by the authors)."

We agree that a go-signal can often lead to large neural activity changes in motor tasks (see also point #2 below, i.e. Kaufman et al.³). As outlined below, we first clarified the experimental design and previous analyses and then conducted additional analyses, which collectively demonstrate that the large activity change is not the result of reaching the LL.

1. Experimental design: The lower limit is present throughout every trial. The limit never changes its position, nor does it suddenly appear within each trial. Thus, participants could see the horizontal bar already at trial onset and therefore, prepare their motor response.

Reviewer comment #1:

I think this part was well explained in the original submission, and I considered this in my comment. A go cue, in this case, could be thought of as the disappearance of the HLL line (occluded by the other ascending bar). A go cue does not need to be the appearance of a sensory stimulus, it can also be a change or disappearance of a sensory stimulus.

Our response to #1:

We fully agree with the reviewer that a go cue does not need to be the appearance of a sensory stimulus. However, it is not correct that the lower limit was occluded by the ascending bar as the lower limit was visible throughout the entire trial. Hence, the only sensory information that was dynamically changing during the trial was the ascending bar; the rest remained constant.

Noted on this. This clarification, while useful, does not imply that this task did not have a "designated go-cue". If it helps, you can think of the gap between the ascending bar and the

LL disappearing as the go-cue. The key point here is that the subjects received an instruction to initiate the movement as soon as certain sensory event occurred, which is the very definition of a go-cue. I would suggest that the authors add a statement in the paragraph highlighted below to clarify this.

Further additional analyses that we demonstrate below in fact clarify that there were no prominent changes in neural activity triggered by the lower limit. However, we added the following paragraph to the discussion in order to provide a balanced view:

Page 34

"Furthermore, while we did not employ a designated go-cue in our experimental paradigm, the lower limit (cf. Fig. 1a) might still resemble a go-cue and could potentially trigger condition-invariant activity changes. Consequently, based on the current experimental paradigm, we cannot fully disentangle neural activity that reflects context-dependent processing from neural activity potentially triggered by the lower limit. However, the fact that context-dependent dynamics already evolved (cf. Fig. 2a & Supplementary Fig. 5a) and neural dynamics mainly ramped-up prior to lower limit suggests that the observed neural activity patterns in PFC were not solely triggered by the lower limit, but reflected intrinsic, context-dependent dynamics prior to the lower limit."

Thank you for adding this clarification.

The early motor preparation was particularly evident in trials with no uncertainty (0% likelihood of a stop). In this condition, we did not observe a strong change in neural activity in human PFC (Figure 2a; Figure 3a; Figure 5c).

Reviewer comment #2:

These figures do not show that there is no change in neural activity associated with the go-cue. Figure 2a shows that post-HLL there is a rise in activity, Figure 3a is just a model fit, so even if there was a transient, it would not be seen in this model, and Figure 5c shows responses aligned to button response, and not HLL. So none of these make the point the authors are attempting to make here.

Our response to #2:

We respectfully disagree. First, the change in activity upon the lower limit should not be specific to the 0% condition, but should instead be observable across all conditions (0%, 25% & 75% likelihood of stopping).

Yes, agreed, and I didn't mean to imply otherwise. With one caveat:

POINT 1. It should be observable across all conditions, so long as *the go-cue triggers* the movement. In this task, many (most?) of the movements were not triggered by the HLL (in particular for the 0% condition). To assess HLL-triggered activity, only HLL-triggered saccades should have been assessed.

However, as can be clearly seen in Figure 2a, the rise in activity for trials with a 25% and 75% likelihood of stopping does not coincide with the lower limit, but occurs before that. Indeed, Figure 2a shows that around 300ms before the HLL there is an increase in activity for the 75% condition. This observation, however, is consistent with an HLL-triggered change in activity, since there could simultaneously exist pre-HLL and HLL-triggered changes in activity.

Furthermore, we computed the average HFA activity before and after the lower limit for the 0% condition to test whether the HFA activity significantly differed between the two time periods. This revealed that HFA activity did not change from before to after the lower

limit (**Response Fig. 1**); hence rendering it very unlikely that the lower limit triggered large activity changes.

As discussed in POINT 1 above, this observation does not address the question, since most of the 0% condition saccades are initiated prior to the HLL.

Instead, this finding supports our statement that the human PFC was not 'recruited' during trials with no uncertainty (in line with no significant changes in the ramping slope; cf. Figure 3a).

I think this result is solid.

In comparison, and in line with what the reviewer proposed, activity in motor cortex was condition-invariant and revealed strong ramping dynamics even during trials with no uncertainty. Hence, we concluded that early motor preparation was particularly evident in trials with no uncertainty, since neural activity in PFC did not show significant activation changes.

Response Figure 1. We computed the average HFA before (-0.5 to -0.01s) and after (0 to 0.3s) the lower limit. This revealed that the HFA did not significantly change from pre- to post-lower-limit (Wilcoxon rank sum test; $P = 0.61$).

As discussed above, it could be argued that the 0% condition is the least important condition to look at for this analysis, since subjects are not waiting for the HLL (go-cue) in this condition to initiate their movements (as revealed by their extremely fast RTs - page 6, line 4: mean RT was 43.9ms**). Normally, RTs are >150ms after a go-cue in non-predictive movements. That being said, I presume that in a subset of 0% trials the subjects did indeed wait for the go-cue (I don't have a sense of this since the authors have not provided a distribution of RTs for each condition, but rather only provided a distribution across all conditions – response to reviewers – and a distribution of subject averages – Fig 1c). If that is the case, then some HLL-triggered activity could be observed. Based on Figure 2 (copied below) there does appear to be an early peak in the green trace (0%). Response Figure 1 above likely missed this because it used an average activity for the post-HLL of 0-300, which is too large for the potential HLL-triggered activity (which appear to occur around 50-200ms after the go-cue).

I think the most obvious place to look at pre- vs post-HLL HFA should be made in the most uncertain condition, which is the condition in which subjects are most likely to wait for the HLL to initiate their movement. Based on Fig 2a, copied below, the conclusions would be quite different:

** although see comment below regarding the inconsistency of this RT value and those shown in the plot

In order to better understand whether there might be a strong phase-locked response, we computed event-related potentials for the lower limit. As displayed in **Response Figure 1**, we did not observe a strong change in neural activity upon reaching the LL. This finding demonstrates that the lower limit does not trigger a large activity change.

Response Figure 1. Grand-average event-related potentials relative to the HLL. We computed the average ERP (mean \pm SEM) across participants time-locked to the lower limit. Note that there is no phase-locked change in activity when the target reaches the HLL, precluding that the HLL triggers a consistent and large change in neural activity.

Reviewer comment #3:

As I see it, this plot shows a marked increase in variance post-HLL. This increased variance post-HLL indicates go-cue-triggered changes, which are reflected differently in different electrodes (some increasing and some decreasing activity). If anything, I think this plot proves that go-cue triggers an activity change, and this needs to be taken seriously in subsequent analyses.

Our response to #3:

We thank the reviewer for this comment. We here aimed to make the point that there is no phasic response upon the lower limit. Therefore, we considered the time between -500ms to 0ms prior to the lower limit as 'baseline' and subtracted the mean of each channel from the time series. Therefore, it is trivial that the variance upon the lower limit increases as compared to 'baseline'. We re-computed the grand-average ERP without centering our data (**Response Fig. 2**). We directly tested whether there would be a significant increase in variance (across subjects) from

pre- to post-lower limit using a Levene's test. This revealed that there is no statistically significant increase in variance upon the lower limit (Levene's test; $P = 0.578$).

Response Figure 2. The variance of event-related potentials did not increase from pre- to post-lower limit. Furthermore, there is no phasic response upon the lower limit.

We also performed a Levene's test on single-subject data to test the reviewer's statement that the presumed increase in variance mirrors the variance across channels with some increasing and some decreasing activity. This revealed that the variance across channels did not differ between pre- and post-lower limit in 15/17 participants (Binomial test; $P = 0.0023$; **Response Fig. 3**). Hence, we conclude that there is neither a phase-locked response nor an increase in variance upon the lower limit.

Response Figure 3. The variance of event-related potentials did not increase from pre- to post-lower limit on a single-subject level.

I thank the authors for the effort placed in this answer. However, it is a whole set of analyses, for which there aren't very many details to be able to thoroughly assess. For instance, which trials were included? (only 0% or all?). Was Levene's test done on the normalized values? Or the second plot (un-normalized?). Given the large variability in the values of the different subjects (y-axis in single-subjects plots ranging from 1-20uV), I would hope the test was done on the normalized values (although based on the description it appears that it was done on the un-normalized values instead). Further, while interesting to see the single-subject plots, these are noisy and likely underpowered for statistical assessments, so I am not sure that we can use the lack of stats significance observed in the majority as a strong argument.

2. Single-trial dynamics in uncertain trials (25% or 75% likelihood) exhibit a ramping pattern prior to the lower limit (**Figure 2c/d**; see below). This observation demonstrates that the observed activity patterns were not triggered by the LL, but reflected intrinsic, context-dependent dynamics that were already present prior to the LL.

Reviewer comment #4:

I am unsure why this is brought up here. Yes, this (and Fig 3a) shows that there is a context-dependent ramping in the PFC prior to HLL, but it does not show that there is no go-cue-triggered change, nor does it address the issue highlighted about the decoder using different periods.

Our response to #4:

Our intention was to highlight that single-trial dynamics show context-dependent ramping prior to the lower limit, but do not show a go-cue-triggered change. Visual inspection is certainly subjective and we agree that single-trial examples should not be used to make any strong claims. However, as shown in **Response Figure 1**, the activity prior and after the lower limit did not significantly differ for trials with 0% uncertainty (in trials with 25% and 75% uncertainty, the activity already ramps up before; cf. Figure 2a & Supplementary Fig. 5). Hence, the single-trial dynamics support our statement and therefore also address the presumed issue highlighted about the decoder using different periods as the lower limit did not trigger substantial changes in activity.

As discussed above, I have my reservations regarding response figure 1. The observation that the 25% and 75% uncertainty already ramps up does not imply that there is no HLL-triggered change in activity. They can both co-exist. And indeed, single trial dynamics should not be used as evidence in this case (I am sure you can find single-trial examples in your data that would support any claim you'd like to make).

3. We also time-locked the population response (PC₁) in PFC and motor cortex to the LL to supplement our findings relative to the button release (**Supplementary Figure 5**). As visualized in **Supplementary Figure 5**, the population activity in PFC already ramps-up and dissociates the different conditions prior to the LL. This observation is also in line with our findings on univariate ramping dynamics (**Figure 2a/b**): No significant ramping was

observed in PFC during trials with no uncertainty (0% likelihood), while significant ramping activity was present for trials with a 25% or 75% likelihood of stopping.

Supplementary Figure 5. Context-dependent population dynamics in PFC emerge prior to the HLL. **a**, Population activity in PFC as indexed by the first principal component reveals context-specific activation patterns prior to the onset of the HLL (sum($F_{2,32}$) = 637.4, $P = 0.023$; cluster test). The single-colored horizontal lines show the temporal extent of a significant context-dependent dissociation. Two-colored horizontal lines indicate the temporal extent of significant clusters obtained from pairwise comparisons. **b**, Same as (a), but for motor cortex. Note the similar, context-independent, temporal activation profile across all conditions in motor cortex (sum($F_{2,26}$) = 251.7, $P = 0.073$).

Reviewer comment #5:

This result helps to make the point the authors are trying to make with Figure 5c and d, that PFC is modulated by predictive context. However, to make the point that HLL does not trigger changes in activity the authors would need to show the other PCs (2 and above) to assess the claim that HLL does not trigger a change.

Our response to #5:

We thank the reviewer for this remark. The main reason why we have chosen PC1 here was based on the reviewer's comment #2 where he/she claims that the lower limit triggers a large condition-invariant signal. Hence, we reasoned that if there is such a strong condition-invariant signal ("largest response component in motor cortex [...]"; Kaufman et al. 2016), we should observe this signal in the first PC. We followed the reviewer's suggestion and performed the analysis now on multiple PCs (1 to 5). None of the PCs showed an activity change upon the lower limit (**Supplementary Fig. 6**).

Moreover, we computed the first derivative of the signal which reflects the rate of change in activation. If the lower-limit would have triggered strong changes in neural activity, this should be captured by the first derivative. However, we did not find support of this claim (**Supplementary Fig. 6**). Therefore, there is no statistical evidence to conclude that the lower-limit in this task triggered strong changes in neural activity.

Supplementary Figure 6. Population dynamics quantified by the first 5 principal components do not show a change in activity triggered by the lower limit. a, The neural dynamics in PFC as indexed by the principal component time-series (1 to 5) do not show a time-locked change upon the lower limit. **b,** We computed the temporal derivative of the traces in (a). This revealed that there is no strong rate of change in neural activity upon the lower limit.

Again, this is useful, but with limited description of the analysis it is hard to interpret... were all conditions included? Were all RTs included? PC3 looks like it contains a sharp change after HLL. How is this change deemed non-significant?

In sum, this set of findings demonstrates that the LL did not trigger a large activity change. On the contrary, these additional analyses demonstrate that the activity change reflects the context integration process that started prior to reaching the LL.

Reviewer comment #6:

Unfortunately, I don't see the answers as sufficient to demonstrate that LL did not trigger activity changes, which could then confound many of the analyses.

Our response to #6:

We thank the reviewer for the thorough assessment. However, we respectfully disagree. We implemented several additional analyses to address the concerns. We outline all the changes in detail below. However, we would also like to highlight that the current results were obtained in human participants in the context of clinical recordings; hence, only limited recording time was available. We discuss the study limitations in the discussion and are convinced that future work in other model species that do not suffer from the inherent limitations of human intracranial recordings, such as invasive recordings in non-human primates, may address the remaining details.

In sum, our results demonstrate that:

1. There is no significant change upon the lower limit upon the 0% condition (**Response Fig. 1**). The changes for the 25% and 75% conditions occur much earlier with respect to the lower limit (cf. Fig. 2a & Supplementary Fig. 5). If there would be a strong change in activation upon the lower limit, this should be clearly visible across all trials.
2. The study by Kaufman et al. (2016) showed that the condition-invariant signal is the strongest component in motor cortex. While we have no doubts that this is true, the fact that activity in motor cortex ramps up prior to the lower limit (cf. Fig. 2b & Supplementary Fig. 5b) rather supports the notion of preparatory motor activity.
3. We demonstrated that there is no phasic response upon the lower limit using event-related potentials. We also performed additional control analyses showing that the variance does also not increase from pre- to post-lower limit (**Response Fig. 2 & 3**).

- We showed that none of the PCs shows a clear-cut response upon the lower limit (Supplementary Fig. 6).

In addition to the changes to the manuscript in the previous revision, we now add Supplementary Figure 6 to the manuscript in order to substantiate our claims:

Page 19

"We additionally time-locked the population activity to the HLL to ensure that context-dependent activity in PFC ramped up and was dissociable prior to the HLL and not a consequence of an activation change triggered by the HLL (Supplementary Fig. 5/6)."

I thank the authors for their thorough responses. To be honest, I am torn here. I do appreciate the limitations of human studies (and the enormous benefits as well). I appreciate the considerable effort placed in addressing the concerns raised. However, I am still worried that these analyses may be mixing different cognitive operations within the same analysis, and assigning the found differences to "contextual differences", when in fact they may reflect something else (i.e. motor preparation for long RT trials vs motor execution for short RT trials). Even if it is true that the HLL does not trigger an activity change (which, as discussed above, is not clear), the concern I raised originally would stand... given the differences in RT across conditions, aligning responses to the HLL will inevitably lead to analyses that match different mental states (i.e. pre and post movement).

To give a very specific example, let's consider the -0.2 time point in Fig2a:

As seen in Fig1c, for 0% uncertainty trials (green) subjects were very fast (average RTs <100ms), which means that the HFA data at -0.2 includes a majority of trials in which the subjects are preparing the movement execution. On the other hand, for 75% uncertain trials subjects were comparatively slow (RTs >100ms), which means that the HFA data at -0.2 includes a majority of trials in which the subjects *have not* prepared the movement execution, but rather are preparing a potential movement pending the HLL. So this (and future plots) are comparing very different cognitive states as if they were comparable.

We have added **Supplementary Figure 5** to the supplements and refer to it in the main text:

Page 19, Line 15-18

"We additionally time-locked the population activity to the HLL to ensure that context-dependent activity in PFC ramped up and was dissociable prior to the HLL and not a consequence of an activation change triggered by the HLL (Supplementary Fig. 5)."

Next, we addressed the second concern that was voiced in the first query.

"The issue with the action decoder is that it captures different periods of the task for different categories (pre-LL for fast RTs, post-LL for slow RTs and a mixture for the mid RTs). To carry out the intended analysis the authors would need to match the RTs for the different contexts before

comparing the activities. Unfortunately, this observation renders the rest of the analyses using this action space uninterpretable.”

We performed the analysis as suggested by the reviewer as well as another additional control analysis to preclude that the action decoder is confounded by capturing different periods of the task. We will separately outline the two analyses:

1. We agree that the action decoder captures different periods of the task for different categories. However, the lower limit did not trigger large activity changes as outlined above. Therefore, it is unlikely that the action decoder is confounded by capturing different periods of the task. However, in order to exclude this possibility, we time-locked the data to the LL (which serves as the ‘go’ cue) and trained a linear decoder to discriminate reaction times (action). Above-chance decoding prior to the LL indicates that the decoder captures processes involved in the planning of action. As shown in **Supplementary Figure 6**, a linear decoder can successfully discriminate the reaction times prior to the lower limit. This finding indicates that action plans were initiated prior to reaching the lower limit.

Supplementary Figure 6. Action-specific information emerges prior to the HLL. **a**, Grand average decoding accuracy (mean \pm SEM) for action within PFC when locked to the HLL. Horizontal lines indicate the extent of significant temporal clusters. **b**, same as **(a)** but for motor cortex. Same conventions as in **(a)**.

We have added **Supplementary Figure 6** to the supplements. We have further added an additional paragraph of this control analysis for clarification:

Page 21, Line 9-19

*“We performed an additional control analysis to preclude that the action-decoding was confounded by capturing different periods of the task (i.e., pre-HLL for fast RTs, post-HLL for slow RTs) when locked to the BR. Therefore, we additionally time-locked the population response to the HLL and repeated the classification analysis. In this case, the linear decoder captures the same periods of the task regardless of the respective RT. This analysis revealed a build-up of action-specific information prior to the HLL (**Supplementary Fig. 6**; PFC: $\text{sum}(t_{13}) = 1074.87$, $P < 0.001$, Cohen’s $d = 1.69$; Motor cortex: $\text{sum}(t_9) = 322.79$, $P < 0.001$, Cohen’s $d = 1.67$; cluster test), therefore, supporting the notion that the decoder captured action-planning processes.”*

Reviewer comment #7:

Unfortunately, this analysis does not show what the authors intended to show. As it is shown in response figure 3, RTs included in the analysis include a many trials in which subjects responded faster than 100ms after the HLL. These reaction times are too fast to be triggered by the go-cue, so these are predictive in nature. Thus, the pre-HLL decoding shown in Suppl Figure 6 is likely reflecting the execution of these very fast movements instead of action-planning processes that preceded HLL.

Our response to #7:

We thank the reviewer for this remark, since this is exactly a key finding that we were highlighting. The reviewer is concerned about the presumed activation change upon the lower limit. Therefore, we sought to test whether action-related information was decodable prior to the lower limit which would speak in favor of action-planning processes. As reported above, we demonstrated that reaction times were successfully decodable prior to the lower limit. The reviewer correctly remarks that there are "*many trials in which subjects responded faster than 100ms after the HLL*" and that these reaction times are "*too fast to be triggered by the go-cue*" and that these reaction times are "*predictive in nature*".

At this point, we would like to emphasize that this is a key finding of our study and the main reason that the experiment was designed as such. In trials with zero uncertainty, participants could already prepare the movement and respond once the bar reached the lower limit; hence resulting in very fast reaction times which are, as correctly remarked, "*predictive in nature*".

I understand that the task was designed to have predictive movements in the zero-uncertainty condition. However, this is not a finding per-se, since this could be expected based on the task. It seems to me that it is trivial that RTs can be "decoded", given that for slow RTs the subjects are not executing the motor commands prior to the HLL, whereas for fast RTs they are.

We agree that the decoding might capture direct action, rather than solely action-planning processes. However, this is also not critical for the main message that the current paper conveys;

I think this is my issue with this figure: why is it included if it is not critical to the main message? I am still not sure what we learned with this additional figure.

namely that contextual information is integrated in the human prefrontal cortex by means of a ramping processing mode whereas the inter-areal communication to guide context-dependent action is facilitated by means of an oscillatory communication mode.

The previous figures attempt to make these points (and we discuss them separately in elsewhere), but this decoding figures does not contribute, as far as I can tell, to either of these conclusions.

In the current study, we won't be able to fully disentangle preparatory signatures from direct motor commands as the transition between the two reflects a continuum.

Not necessarily, as the work of Churchland, Shenoy, Svoboda and others shows. There may be an abrupt transition from preparing a movement to executing it, and this abrupt transition occurs around ~100ms prior to the movement execution.

However, we would like to emphasize that the classifier was trained to discriminate fast, medium and slow reaction times. If the classifier would only decode the execution of very fast movements (33% of all reaction times), then the classifier should fail to predict reaction times above chance.

I don't understand the point being made here. The LDA classifier dissociates 3 categories (fast, mid, slow). How could the classifier only decode fast movements?

We would like to emphasize that we already acknowledged the limitation, that we cannot fully separate action-planning from movement-execution based on the current experimental design, in the previous round of revision.

Page 31

"Furthermore, information about action only culminated after the movement onset. Thus, we cannot preclude that these processes mainly capture post-movement, rather than preparatory dynamics. However, we observed a clear build-up of action-specific information (cf. Fig. 5g). This suggests that the observed dynamics track the internal transition from planning to the final movement execution, in line with a recent human iEEG-study²."

Page 20

"Thus, any neural activity that dissociates reaction times prior to the actual movement execution (and that is not primarily driven by early context-integration signals) reflects the internal transition from the planning to the execution of a movement."

Page 22

"Although decoding accuracy for action reached its maximum right after the movement onset, we observed a clear build-up of action-related information over time, suggesting that the decoder captures the direct transition between action-planning and action-execution."

2. Finally, we implemented the proposed analysis by the reviewer and trained and tested the decoder on a subset of trials with matched RTs (time-locked to the HLL). Note this analysis was only feasible for the 25% context condition where a sufficient number of trials was available. As shown in **Response Figure 2**, a linear decoder can reliably discriminate the onset of movement with matched RTs (PFC: first cluster, $\text{sum}(t_{13}) = 483.98$, $P = 0.008$, Cohen's $d = 1$; second cluster, $\text{sum}(t_{13}) = 241.98$, $P = 0.037$, Cohen's $d = 1.52$; Motor cortex: $\text{sum}(t_9) = 246.039$, $P = 0.015$, Cohen's $d = 0.94$; cluster test), indicating that the decoder is not confounded by possibly capturing distinct periods of the task.

Response Figure 2. Context-dependent decoding of action. **a**, Grand average decoding accuracy for action in PFC (mean \pm SEM) using only trials with a 25% likelihood. Horizontal lines indicate the extent of significant temporal clusters. **b**, same as (a) but for motor cortex.

Reviewer comment #8:

Unfortunately, this is not the analysis I was suggesting. I am not even sure what "matching RTs" means if only one context was included (25%). What I meant was match RTs across contexts. The manuscript is making claims about the timing of information about context and action (RT), and the handoff of information. The proper analysis would need all 3 contexts RT-matched, so we could see the information about context and RT, like in Figure 5g.

Our response to #8:

We apologize for misunderstanding the reviewer's suggestion. However, the reviewer's main concern all along is that the action decoder is confounded by capturing distinct periods of the task. While we are convinced that we have provided ample evidence that the lower limit does not trigger substantial activity changes, we tested this notion in another complementary way.

We only used the reaction times (RTs) within the 25% condition in order to reduce the variance of the RT-distribution.

Unfortunately, you have not shown the RT distributions of any of the conditions, so it is hard for me to evaluate this claim.

This minimizes the uncertainty that the action-decoder simply picks-up different periods of the task since the width of the distribution is narrowed. We then demonstrated that the classifier is still able to successfully predict reaction times even within one context condition only, clearly demonstrating that the decoder captures action- and not context-specific activity (see also first round of revision, reviewer #2, comment #6 where we show that we cannot decode action from the 'context-subspace' or vice versa). Moreover, if the action decoder would be fully driven by simply capturing 'different time periods of the task', then the same decoding traces should have been emerged for context-decoding since reaction times strongly differ between context conditions. This is not the case (cf. Fig. 5g; Supplementary Fig. 9) and has also been previously addressed in the first round of revision (reviewer #2, comment #6).

This claim rests on the orthogonalization method employed. The authors selected an unconventional method, which is to carry out LDA separately on different PCs, and use the highest decoding performance for RT and context as the coding dimension. This analysis ignores the potentially important information found in the non-top dimensions. A more conventional approach would be to use the full PC space to run the LDA and impose an orthogonality constraint on the hyperplanes. Alternatively, they could have run an LDA for RT on the full space, and then a second LDA for context in the residuals (and vice versa). It is perfectly possible that the analyses that I outline above would lead to context signals closer to HLL.

We fully agree that the reviewer's suggestion is an excellent idea. However, this is unfortunately not possible simply due to the fact that reaction times strongly differ between context conditions (the main hypothesis of the manuscript) and the resulting number of trials that could be used for the analysis would be too low. This is a direct consequence of clinical recording settings and an inherent limitation of human intracranial recordings. We have added this limitation into the main text:

Page 34-35

"However, it is worth noting that we cannot rule out some alternative hypotheses due to inherent limitations of human intracranial recordings and the currently employed experimental design. First, due to the inherent limited coverage in intracranial recordings, we cannot preclude that specialized sub-regions within the human motor cortex (e.g. anterior or posterior parts of the supplementary motor area, frontal eye fields, premotor cortex) might also encode contextual information. Second, based on the current experimental design, we cannot fully dissociate between preparatory- and movement-related computations. This is a direct consequence of the fast-paced task where the transition between preparation and movement reflects a continuum. Furthermore, while we did not employ a designated go-cue in our experimental paradigm, the lower limit (cf. Fig. 1a) might still resemble a go-cue and could potentially trigger condition-invariant activity changes. Consequently, based on the current experimental paradigm, we cannot fully disentangle neural activity that reflects context-dependent processing from neural

activity potentially triggered by the lower limit. However, the fact that context-dependent dynamics already evolved (cf. **Fig. 2a** & **Supplementary Fig. 5a**) and neural dynamics mainly ramped-up prior to lower limit suggests that the observed neural activity patterns in PFC were not solely triggered by the lower limit, but reflected intrinsic, context-dependent dynamics prior to the lower limit. Finally, prefrontal population activity is high-dimensional in nature, where different operations are encoded in distinct subspaces. Yet, our analytical approach does not allow to draw any inference on the dimensionality of the decoding latent space, only on the overall dimensionality of the neural data. However, our findings strongly support the notion that the high-dimensional prefrontal functional architecture constitutes a substrate for flexible goal-directed behavior and that simultaneous processing in separate coding dimensions maximizes information-coding capacity of the underlying population³⁻⁵.”

This is fine. However, if there are limitations in the methods, the conclusions should be consistent with these limitations.

Collectively, this set of finding demonstrates that action decoding is not driven by the inclusion of slightly different task periods. On the contrary, these analyses reveal that action information was present prior to the HLL, exhibited a gradual build-up and hence, most likely capturing action-planning processes.

These considerations are now also reflected in the discussion:

Page 31, Line 7-11

*“However, we observed a build-up of action-specific information (cf. **Figure 5g**) that emerged prior to the HLL (**Supplementary Fig. 6**). This suggests that the observed dynamics track the internal transition from planning to the final movement execution, in line with a recent human iEEG-study².”*

Reviewer comment #9:

Given the issues raised above, I still think that the results presented have serious confounds, and my original comment stands.

Our response to #9:

We are thankful for the reviewer’s constructive comments and the time taken to review the manuscript. The concerns were valid and needed to be addressed. However, based on the additional analyses, we are convinced that the presented evidence demonstrates that there is no confound with respect to the action decoder, since the lower limit did not trigger substantial changes in activity. Hence, even though the decoder picks up slightly different periods with respect to the lower limit, it does not confound the analysis. However, since the recordings were obtained in the clinical context, we now discuss the inherent study limitations in the discussion. In sum, we are convinced that the results provide an important contribution, since the demonstrate that population-based context and action coding (as previously observed in model species, such as non-human primates) are also evident in the human brain.

Given the considerations above, unfortunately I remain unconvinced, and my concern stands.

- 1- BR-aligned analyses may suffer from confounds of HLL-triggered activity (I am willing to suspend my disbelief here, but even if there are no HLL-triggered changes in activity, points 2 and 3 stand).
- 2- HLL-aligned analyses suffer from confounds of comparing different cognitive processes (movement planning vs movement execution).
- 3- The decoding analysis does not add much to the main conclusions of the paper.

2- Line 366: "This indicates that PFC, but not motor cortex, is gradually recruited as a function of behavioral uncertainty (Fig. 5e)." I don't think your results should be interpreted this way. When aligned to button release (as in Fig5), the differences begin emerging around 0.2s before movement, which happens to coincide with the LL (lower limit) onset for the slowest RT trials. If PFC was gradually recruited, these differences should start emerging from the moment that the contextual cue is given (560-750ms before the LL). But that is not the case. Go-cues (like the LL) trigger a condition-invariant signal, or CIS (Kaufman et al. 2016, Guo et al. 2014, Inagaki et al. 2022). In light of the CIS, averaging trials, aligned to movement onset, could lead to the exact figure shown. So this analysis should not be interpreted as a gradual increase.

We thank the reviewer for this relevant query and bringing these papers to our attention. We believe that our description of the task structure caused this misunderstanding. We therefore briefly clarify the task structure again before we discuss our data in light of the CIS.

1. Task structure: Each trial started with a brief baseline period (500ms) followed by a contextual cue presented at the center of the screen (green, orange or red indicating the likelihood of a premature stop). Importantly, after receiving the cue, participants self-paced the trial start (average time to start the trial: $1.76s \pm 0.55s$; mean \pm SD). By pressing the space bar, the target would start moving upwards and reach the LL after 560 – 580ms. As a consequence, the time between the contextual cue and the moment the target reaches the LL could vary from trial to trial. Once the target reached the LL, participants were instructed to release the button. We have added this information into the methods section:

Page 38, Line 1-4

"Upon receiving the predictive cue, participants were able to start the trial in a self-paced manner by pressing the space bar on the keyboard (average time to start the trial: $1.76s \pm 0.55s$; mean \pm SD)."

Reviewer comment #10:

This was clear to me from the original description. I'm not sure why the authors thought that there was a misunderstanding on this based on my comment above.

Our response to #10:

We thought there was a misunderstanding because the reviewer stated that: *"If the PFC was gradually recruited, these differences should start emerging from the moment that the contextual cue is given (560-750ms before the LL)."* This statement is incorrect and that is why we thought there was a misunderstanding.

As mentioned above, after receiving the contextual cue, participants self-paced the trial start. Therefore, the contextual cue is not necessarily given 560-760ms prior to the lower limit, but as stated above: *"Upon receiving the predictive cue, participants were able to start the trial in a self-paced manner by pressing the space bar on the keyboard (average time to start the trial: $1.76s \pm 0.55s$; mean \pm SD)."* Hence, it is not surprising to see that the effects did not directly emerge upon the moment the participants pressed the button and self-started the trial, since the time between the appearance of the contextual cue and the self-paced start was variable.

Indeed I did not realize that the bar started moving at a variable time after the context cue. This, however, does not preclude the comment above. Again, I will suspend my

disbelief about the HLL-triggered signal for a minute, to highlight that the problem still persists.

A subject that views a 0% uncertainty trial, will press the spacebar and see that the bar starts ascending towards the LL. During this period, the subject knows that there will be a movement as soon as the bar reaches the LL. This context, which exists throughout, and is presumably reflected in the activity somewhere in the brain (quite likely in the PFC, as has been observed in numerous previous studies in NHPs), means that subjects will start preparing for an impending movement. None of these signals are identified in the current study, since the context predicting signals only emerge ~300-200ms prior to LL. This is fine, since it may reflect species-, task-, region-, or recording-method differences. Regardless, cognitively, the context is present in the subject.

A similar description applies to the 25% and 75% contexts, the only difference being that the task is “different” for these cases... the subjects have to be *more alert*, and *pay closer attention* to the bar movement (this is my guess as to why the uncertain contexts – 25% and 75% - show the ramping activity, while the certain context doesn’t... because the ramping activity reflects task engagement/alertness/selective attention, rather than context per-se. But this is just speculation). Again, here, the task rule (context) should be maintained online somewhere, even early on (after the pressing of the context cue presentation), but this is not observed here.

Back to the original point... if there were context-specific signals in the brain, they should be present from the moment that the context is shown to the subjects. If the context signals only emerge seconds later, I would argue that these are not context signals, but rather something else that correlates with context (such as attention).

2. As correctly remarked, the context-dependent dissociation – when locked to the button release – emerges ~0.2s prior to the actual movement onset. This approximately coincides with the lower limit. However, as can be seen in **Figure 1c**, reaction times strongly vary from trial to trial. Therefore, as depicted in **Response Figure 3**, the time between the movement onset and the moment the target reaches the LL is variable and jittered across trials. This temporal jitter renders it unlikely that our results can be fully explained by a change in neural activity that is time-locked to the LL.

Response Figure 3. Movement onset distribution with respect to HLL. The distribution depicts the movement onset timing relative to the HLL. The vertical dashed line reflects the average movement onset time. Note that the timing is jittered across trials.

Reviewer comment #11:

I have addressed this point about HLL-triggered activity thoroughly above. However, this depiction of RT raises even more doubts. It would be useful to show the RT distribution for the different contexts. I imagine that the 0% accounts for most of the <500ms RTs shown here, whereas the 75% context probably has none of these. This asymmetry would of course be very problematic for all the analyses carried out.

More importantly, I am not sure how this observation (that there is RT variability) has any bearing on my comment about contextual cues having to emerge much earlier (the moment the cues are given).

Our response to #11:

First, we do not understand why this depiction should raise any doubts as it is simply a different depiction of Figure 1c. The reviewer asks for an RT distribution for the different contexts. This is what we have depicted in Figure 1c and clearly described in the results section: *"We found that reaction times (RT) gradually increased as a function of uncertainty (Fig. 1c)."* This was in fact the main manipulation of the task.

We manipulated the uncertainty by providing different probabilities with which the bar would prematurely stop prior to the lower limit. Hence, in the 0% condition, there was no uncertainty and the participants could already prepare the movement, resulting in fast reaction times. In contrast, when participants were provided with the orange or red cue (25% and 75% probability of stop, respectively), reaction times were slower because the bar could still prematurely stop.

Yes, I have no issues with this design. The question I raised is because Response Figure 3 shows the RT distribution. Figure 1c shows the distribution of single subject *averages*, which is not the same. With the figures shown so far, I don't know how often subjects waited for the HLL (rather than predict its timing) across the different contexts.

Second, this observation has an important bearing on the following reviewer's comment:

"This indicates that PFC, but not motor cortex, is gradually recruited as a function of behavioral uncertainty (Fig. 5e)." I don't think your results should be interpreted this way. When aligned to button release (as in Fig5), the differences begin emerging around 0.2s before movement, which happens to coincide with the LL (lower limit) onset for the slowest RT trials."

The reviewer's argument here is that the differences in population activity that we observe between the different context-conditions are driven by the presumed activity change upon the lower limit. While we have already addressed this point thoroughly above, the variability of RTs renders it even more unlikely that Fig. 5c is driven by activity changes upon the lower limit. If it was true that the results are a simple consequence of activity changes upon the lower limit, one would have expected a temporal lag in the responses shown in Fig. 5c simply due to different RTs between context-conditions (i.e. activity in 75% condition increasing first, then 25% condition, then 0% condition because the distance to the lower limit decreases).

Agreed. And it does seem that it is the case (at least with visual inspection):

The deviation from baseline for the green line (0%) deviates from baseline ~100ms after the other 2 conditions. However, I will refer here to my previous response highlighting that the concern does not rest on the existence of HLL-triggered signals.

Reviewer comment #12:

I also just realized that Fig 1c above is not actually showing RTs, but rather something else (in the order of 700ms... maybe time from context cue to response?)

Our response to #12:

We thank the reviewer for this remark and thank the reviewer for highlighting this inconsistency. Figure 1c shows RTs but it reflects the time between the self-paced trial start and the reaction. We changed Figure 1c to reflect the time between the lower limit and the reaction times.

This is still not consistent with the values described in the text (and the distribution shown in the Response Figure 3. This figure shows subject average values in the order of 80-100ms (depending on condition), but the text says (page 6, line 4) that the mean RT was 43.9ms. And then, response figure 3 shows an average of ~90ms.

- We also time-locked the population activity to the LL (see comment #1; **Supplementary Figure 5**). This analysis revealed that context-dependent activity already emerged ~0.2s prior to the LL in PFC. This finding supports the univariate analyses where context-dependent activity is evident prior to the LL (**Figure 2a**; **Figure 3a**). In sum, this set of findings demonstrates that the LL did not trigger a large activity change. Instead, these findings highlight that the activity change reflects contextual processing that started prior to reaching the LL.

As mentioned before, this result could be driven by the fact that for green (0%) the movement execution activity has started, whereas for the others it has not yet.

Collectively, these results are in line with a context-dependent activation in PFC, that is not evident in motor cortex. We carefully considered the suggested papers, in particular Kaufman et al.¹. CIS reflects a large and rapid change prior to the movement onset in dorsal premotor regions

as well as in M1 that reflects more variance than condition-specific (“tuned”) responses. The notion of CIS reflecting the internal transition from motor preparation to movement onset in motor cortex is fully in line with our findings. In the present dataset in humans, the principal motor component does not exhibit context-dependent activity. Therefore, we believe that our findings nicely complement and extent similar findings previously obtained in non-human primates. We now discuss our findings in light of condition-invariant signals in motor cortex:

Page 32, Line 3-14

“Neural activity in motor cortex revealed large, context-independent activity changes prior to the movement onset. This context-independent activity, previously also referred to as a condition-invariant signal (CIS)¹, typically reflects the largest response component in motor cortex². The present findings are compatible with a CIS in human motor cortex. The first principal component of the HFA-signal in motor cortex is (1) context-independent and (2) explaining the largest variance. An unresolved question is whether other areas might drive this sudden change in motor cortex activity. Previous studies have identified a large-scale network that might provide input to motor cortex, including subcortical^{6,7} and cortical structures^{8,9}. The present findings demonstrate that human PFC also modulates neural activity in motor cortex.”

In contrast to motor cortex activity, PFC population activity was strongly context-dependent (**Figure 5c; Supplementary Figure 5**). Therefore, the population activity in PFC cannot be explained by a CIS. We nuanced our phrasing of a gradual recruitment in PFC:

Page 19, Line 12-15

“This set of findings indicates that population activity in PFC is modulated by predictive context. In contrast, motor cortex population activity displayed context-invariant dynamics supporting a condition-invariant signal¹ (see Discussion).”

3- LDA analysis: This is a very unconventional way to carry out this analysis. The decoding latent space could be a high-dimensional space that spans multiple PCs. So normally these analyses are done on all dimensions at once (or all dimensions that explain more than 95% of variance), and not one PC dimension at a time, as was done here. I think that this analysis, as carried out, is not very meaningful. If the goal was to enforce orthogonality, you can impose this constraint in the higher-dimensional LDA spaces (although I’m not sure why you’d want to impose orthogonality).

We thank the reviewer for the comment and the excellent suggestion to run the analysis by using the full, high-dimensional space. We separate the response in two parts. First, we would like to clarify the rationale behind our analysis approach. Subsequently, we outline the results of the analysis when performed in high-dimensional space as suggested by the reviewer.

Rationale behind the analysis:

1. Using univariate analyses (**Figure 1-4**), we established that ramping, but not oscillatory dynamics are modulated by predictive context. Instead, oscillatory dynamics primarily supported the interplay between prefrontal and motor cortex. These findings then guided our multivariate analyses where we (1) examined the functional roles of oscillatory and ramping dynamics and (2) identified the population-level interaction between the prefrontal and motor cortices. Therefore, we first established low-dimensional subspaces that could dissociate contextual from action-specific processing. This approach subsequently allowed us to then test whether the principal dimension with the strongest oscillatory power would (1) encode aspects of contextual processing or action-planning and (2) would temporally synchronize the prefrontal-motor network. In order to mitigate the different orientation of the multidimensional space of prefrontal and motor cortex, we constrained our analysis to the relevant one-dimensional subspace. As indicated by

the reviewer, this might seem unconventional, however, it provides the basis to study information transfer between the relevant coding dimensions and to understand the role of oscillatory dynamics in the transfer of information on the population-level. We conducted several control analyses (i.e., decoding in high-dimensional space as suggested by the reviewer) to demonstrate that the relevant coding dimension is well approximated by a single dimension and that adding more dimensions does not necessarily improve the decoding performance.

2. Another reason that led us to constrain our decoding analysis to a low-dimensional subspace was the heterogenous number of electrodes implanted in different patients. In turn, this would have resulted in a largely heterogenous number of PCs that define a high-dimensional space. This is unfortunately an inherent limitation of intracranial recordings in humans.
3. However, we fully agree with the reviewer that the decoding latent space itself could be high-dimensional and thus span multiple PCs. Therefore, in the revised version of the manuscript, we now explicitly emphasize that our decoding analysis does not allow any inference regarding the dimensionality of the decoding latent space.

Page 34, Line 10-13

"Importantly, prefrontal population activity is high-dimensional in nature, where different operations are encoded in distinct subspaces. Yet, our analytical approach does not allow to draw any inference on the dimensionality of the decoding latent space, only on the overall dimensionality of the neural data."

Reviewer comment #13:

Noted on the rationale, and the new analysis. It makes sense.

Our response to #13:

Thanks.

Additional analyses:

Finally, we followed the reviewer's suggestion and performed the analysis using all principal components cumulatively explaining more than 95% of the variance. This revealed comparable effects (**Supplementary Figure 8**) supporting our observation that contextual processing is PFC-dependent whereas action-related processing occurs in both PFC and motor cortex.

Supplementary Figure 8. Decoding analysis of context and action in high-dimensional space.

a, Grand average decoding accuracy (mean \pm SEM) for context and action in PFC. Classification was performed using all principal components cumulatively explaining more than 95% of the variance. Horizontal lines indicate the extent of significant temporal clusters (color coded for the

respective feature). **b**, Grand average decoding accuracy (mean \pm SEM) for context and action in motor cortex. Only action, but not context, could be decoded in motor cortex. Same conventions as in (a).

We have implemented these results into the supplements and directly refer to and discuss them in the main text.

Page 21, Line 19 – Page 22, Line 8

"We performed additional control analyses demonstrating that the context-encoding and action-encoding subspaces capture dissociable processes related to contextual computations and the planning of actions (Supplementary Fig. 7; Methods). We also performed the decoding analyses using a high-dimensional neural space on all principal components cumulatively explaining more than 95% of the variance. This replicated our main findings that the human PFC encodes both context and action (Supplementary Fig. 8; context: first cluster, $\text{sum}(t_{16}) = 408.68$, $P = 0.002$, Cohen's $d = 0.74$; second cluster, $\text{sum}(t_{16}) = 151.62$, $P = 0.039$, Cohen's $d = 0.69$; action: $\text{sum}(t_{16}) = 1040.76$, $P < 0.001$, Cohen's $d = 1.08$; cluster test) whereas motor cortex only encodes action (Supplementary Fig. 8; context: first cluster, $\text{sum}(t_{13}) = 39.46$, $P = 0.427$, Cohen's $d = 0.66$; action: $\text{sum}(t_{13}) = 828.57$, $P < 0.001$, Cohen's $d = 1.11$; cluster test)."

To directly address the similarity between the decoding traces obtained from a high-dimensional vs. one-dimensional set, we adopted an analysis by Chiang et al.¹⁰ to find the optimal ensemble size for a decoder. We sorted the individual PCs according to their decoding accuracy and then iteratively added the sorted PCs to train the decoder. This analysis revealed that a high-dimensional space does not improve classification performance (Response Figure 4). Overall, the correlation between the decoding time series using multiple PCs and the decoding time series using only the most informative PC showed that the time courses are largely correlated (Response Figure 5).

Response Figure 4. Comparison of decoding performance in low- vs. high-dimensional space. We tested the optimal dimensionality for the linear classifier to decode context as well as action. We sorted individual PCs according to their decoding performance and then iteratively added the sorted PCs to the decoder. This revealed that decoding performance did not benefit from adding more (uninformative) components. Single-subject example, *Upper Left*: Depicts the magnitude of the decoding cluster as a function of the number of sorted PCs to train the decoder on context. *Lower Left*: Time-resolved context-decoding accuracies computed on distinct numbers of PCs. Note the strong temporal similarity of the decoding traces. *Upper Right*: Depicts the magnitude

of the decoding cluster as a function of the number of sorted PCs to train the decoder on action.
Lower Left: Time-resolved action-decoding accuracies computed on distinct numbers of PCs.

Response Figure 5. Group-level correlation between decoding traces resulting from quantification in low vs. high-dimensional spaces. We quantified the correlation between individual decoding traces (*left and middle panel*) to test their similarity. We observed a strong temporal correlation on the group-level (*right panel*) between the decoding traces independent of the number of PCs that were used to train the linear decoder.

Collectively, this set of findings demonstrates that our decoding analyses yields valid and interpretable results that replicated using a high-dimensional set of components.

Reviewer comment #14:

These analyses indeed do show that this information lives in low-D spaces, and support the authors claim.

Our response to #14:

We would like to thank the reviewer again for the suggestion to perform this analysis in high-dimensional space as well, which strongly substantiated our claims.

4- Fig 5i shows that most action peaks occur post-movement (BR), so these cannot be causal in the movement generation nor planning. As such this “serial unfolding” claim does not really rest in solid grounds. This also motivates pg 21: “This raises the question, whether and how the representation of the action plan in PFC is passed on to motor cortex for final action execution?”, which loses force when we realize the post-movement nature of these sequential signals.

We thank the reviewer for the comment. We would like to clarify that we did not intend to draw any conclusions or make any claims regarding causality – neither with respect to movement generation nor movement planning. We apologize if our wording was not clear-cut. To avoid further confusion on data interpretation, we now explicitly state that we cannot draw any causal inference on the directional flow of information from movement planning to movement generation.

Page 31, Line 2-5

“However, due to the correlative nature of our study we cannot draw any causal inference or directionality nor exclude the possibility that the observed effects could be driven by a third structure that we did not record from.”

We would like to clarify that the serial unfolding claim only referred to the temporal segregation between context-integration in PFC and action-planning in both PFC and motor cortex. It did not

refer to the temporal dynamics of action-related processes between PFC and motor cortex. In the revised version of the manuscript, we now make this explicit.

Page 22, Line 8-18

*"Having established that context and action can be decoded from the prefrontal-motor network, we next quantified their temporal dynamics by computing the peak-decoding latency. Peak-decoding accuracy revealed a distinct temporal pattern with initial information about contextual information in PFC (**Supplementary Fig. 9**; $P = 0.024$; Kruskal-Wallis; $-0.09 \pm 0.19s$. with respect to BR; mean \pm SD), followed by action-related processes in PFC ($0.05 \pm 0.15s$.) and motor cortex ($0.11 \pm 0.12s$.), suggesting that contextual computations in PFC are completed prior to action-relevant computations. Although decoding accuracy for action reached its maximum right after the movement onset, we observed a clear build-up of action-related information over time, suggesting that this the decoder maps information with respect to action-planning."*

However, we fully concur that it might be counterintuitive that the maximum decoding performance for action only culminates just after the movement onset. Yet, considering the human iEEG literature, this is not unexpected. For example, Haller et al.¹¹ observed that the high-frequency activity in motor cortex peaks after the movement onset. Our results (**Figure 2b/5d**) substantiate this observation. Furthermore, Ter Wal et al.² used linear classifier on human iEEG data to decode left vs. right movement. The authors showed that decoding performance about the movement culminates after the movement onset.

There are at least two potential reasons for this observation:

- 1) As the computation of the HFA involves a filter in the temporal domain (i.e. from 70-150 Hz in non-overlapping 10 Hz wide bins), the temporal precision of the data is inevitably reduced as compared to single unit data, a necessary limitation of human intracranial recordings. As a direct consequence, the temporal uncertainty with respect to minima and maxima is increased.
- 2) Figure 2b/c in Ter Wal et al.² shows the decoding performance with respect to left vs. right movement. The decoding trace clearly builds up over time and yields sudden changes $\sim 0.2s$ prior to the movement onset. This fully matches our findings (**Figure 5g/h, left panels**) that action can be successfully decoded $\sim 0.2s$ prior to the BR. This demonstrates that the linear decoder captures the transition from movement planning to movement execution.

Reviewer comment #15:

To be honest, I was not perplexed by the post-movement decoding peak. This is observed over and over in many brain regions. What I was raising is the use of this metric as a means to talk about the serial unfolding, or any temporal metric. If anything, I would have expected the time of first significant decoding time-point as the relevant metric.

Our response to #15:

We apologize, but this was not well-phrased. The reviewer clearly stated that "Fig 5i shows that most action peaks occur post-movement (BR), so these cannot be causal in the movement generation nor planning". Therefore, we argued that the time series of decoding accuracy can still capture action-planning processes even though they culminate post-movement. Furthermore, as mentioned above, the 'serial-unfolding' referred to the temporal segregation between context-integration and action-related processes. Figure 5g and Supplementary Figure 8 demonstrate that successful decoding of context precedes successful decoding of action. We interpreted this as a 'serial unfolding' from context to action. To further substantiate this, we computed the peak-decoding accuracies and demonstrated that contextual integration precedes to action-related processes (Supplementary Figure 9). Nevertheless, we fully agree that we cannot entirely

preclude that the processes we capture might reflect post-movement, rather than preparatory dynamics and therefore added this as a limitation in the main manuscript (see below).

We now discuss this finding in more detail in the discussion:

Page 31, Line 5-11

"Furthermore, information about action only culminated after the movement onset. Thus, we cannot preclude that these processes mainly capture post-movement, rather than preparatory dynamics. However, we observed a clear build-up of action-specific information (cf. Figure 5g). This suggests that the observed dynamics track the internal transition from planning to the final movement execution, in line with a recent human iEEG-study²."

With respect to the "information hand-off" of action plans, we initially based our interpretation on (1) the phase-slope-index (Figure 4h) as well as (2) the temporal information of the average action-decoding traces (Figure 5g/h; left panel). In the previous revision of this manuscript, we further added the cross-correlation analysis, peak-decoding analysis as well as the cross-regional generalization analysis (see comment #5) to substantiate our interpretation.

Reviewer comment #16:

Maybe this is discussed later, but as of now I see no reason to see any of these results as "information hand-off"

Our response to #16:

We fully concur and elaborate on this issue below. We omitted the term from the revised manuscript.

Noted, and thank you for the changes.

5- The methods description of the cross-regional pattern analysis is not clear from the description of the methods and results section. I'm not sure how you can test classification using different sets of electrodes that were not used in the training of the LDA. As such, I cannot assess fig 5j and the "information handoff" claim.

We thank the reviewer for bringing this up and apologize for the lack of clarify in our description. As pointed out in our response to comment #4, we performed the cross-regional generalization analysis to directly assess the information hand-off between prefrontal and motor cortex.

The reviewer is entirely correct that this analysis would be impossible on different sets of electrodes. As outlined in our response above, we mitigated this problem by focusing on the relevant coding dimension in both regions.

Thus, we did not train and test the classifier on electrodes as features, but instead tested whether the neural information in the PFC action subspace generalizes to the motor cortex action subspace. We thereby tested if a discriminative (informative) pattern that was present in PFC was equally present in motor cortex.

The selection of the coding subspace enabled this analysis, which otherwise would have been impossible as correctly remarked by the reviewer. In the revised version of the manuscript, we made this more explicit in the results section and have further added details to the methods description.

Page 23, Line 7-11

"We trained a linear classifier on the activity of the PFC action subspace to discriminate reaction times and subsequently tested if the classifier generalized to the action subspace in motor cortex."

Hence, we tested if a discriminative pattern that was present in the action-specific subspace in PFC was equally present in the action-specific subspace in motor cortex."

Page 51, Line 10-17

"Cross-regional pattern analysis. To quantify whether a discriminative action-specific pattern present in PFC is equally present in motor cortex, we trained a linear classifier on every time point in the action subspace (principal component maximally discriminating action) in PFC and subsequently applied it on every time point in the action subspace in motor cortex. Cross-validation was not necessary since training and testing datasets were independent. Finally, classification values were tested against chance level and corrected for multiple comparison using cluster-based permutation statistics."

Reviewer comment #17:

Thank you. This makes more sense (in terms of methods). However, I am still unclear on how this analysis supports information hand-off.

Our response to #17:

Thank you for having us provided the opportunity to better clarify the rationale behind this analysis. With respect to the 'information hand-off': As mentioned previously, we based our claim that there is an information hand-off from PFC to motor cortex upon three findings. (1) The phase-slope index analysis, which is a measure of directional connectivity, revealed that there is a context-dependent directional interaction from PFC to motor cortex. (2) The temporal dynamics of the decoding accuracies revealed that context could be successfully decoded in PFC prior to action in both PFC and motor cortex. Hence, we interpreted this as an 'information hand-off' from PFC to motor cortex. (3) We then aimed to test whether there is also an 'hand-off' of action-specific information and therefore employed the cross-regional pattern analysis. This analysis revealed that that a discriminative pattern that was present in the action-specific subspace in PFC was equally present in the action-specific subspace in motor cortex, but with a temporal lag, suggesting that this information initially emerged in PFC and was passed onto motor cortex.

However, we realized that this interpretation is clearly subject to debate and hence nuanced our language and removed claims regarding the 'information hand-off'.

Page 4

*"We describe a striking functional dissociation between population activity and network oscillations where human PFC encodes predictive context and the current action plan in distinct subspaces using a continuous processing regime, **while theta oscillations mediate the inter-areal communication between PFC and motor cortex to guide context-dependent actions.**"*

Page 23

"Finally, to infer whether action-related information generalizes from PFC to motor cortex prior, we conducted a cross-regional pattern analysis (Methods)."

Page 23-24

"In sum, this set of findings demonstrates that the human PFC integrates contextual information at early stages using a low-dimensional subspace and devises them into an appropriate action plan that can be read-out by motor cortex for final action-execution."

Page 27

"[...] indicating a functional role of theta oscillations to mediate the cross-regional generalization of action plans from prefrontal to motor cortex."

Page 29

"Furthermore, our results reveal that (V) theta synchrony temporally coordinates action-encoding population subspaces, thereby mediating the cross-regional generalization of action plans from prefrontal to motor cortex (Fig. 6)."

I would like to thank the authors for these changes, which now accurately reflect the conclusions of these analyses.

Minor Points

1- Fig g and h right are not very useful without stats... the axis spans seem to be chosen to make a particular point rather than to reflect the data's structure.

We concur that the right panels in figure 5g/h were largely descriptive and only served illustrative purposes. We therefore decided to remove them from the manuscript.

Reviewer comment #18:

Thanks.

2- Pg 32: "Here, we replicate this finding in humans, but in contrast to NHPs, we found no evidence that the human motor cortex encodes predictive context." This is not quite true, since the NHP work referenced provides evidence of pre-movement prediction regarding the specific movement (not context). This study only had one movement (button release) so this could not be assessed.

Thank you for bringing this mistake to our attention. We realized that our referencing was misleading. The replication with respect to a low-dimensional neural pattern in motor cortex referred to the following two references: Gallego et al.¹² and Suresh et al.¹³. The statement "[...] but in contrast to NHPs, we found no evidence that the human motor cortex encodes predictive context" instead referred to Thura and Cisek^{14,15}, Wang et al.¹⁶ and Glaser et al.¹⁷. However, after carefully re-reading the referenced studies, we decided to nuance the statement that prior work in NHPs has revealed context-dependent coding in the (pre-)motor network. In particular, we now explicitly state that context-dependent coding has mainly been found in adjacent premotor areas, such as frontal eye fields or dorsal premotor-cortex. We changed the text in several instances:

Page 28, Line 21-23

"Previous work in NHP indicated that adjacent premotor structures, such as frontal eye fields^{5,18} or dorsal premotor cortex^{14,17}, might mediate context context-dependent decision-making."

Page 33, Line 13-17

"While previous evidence in NHP indicated that adjacent premotor structures, such as frontal eye fields^{5,18} or dorsal premotor cortex^{14,17} perform context-dependent computations, we found that neural trajectories in prefrontal, but not motor cortex, dissociated the current predictive context."

Page 33, Line 23 – Page 34, Line 9

"Previous work in NHP demonstrated that motor cortex exhibits a low-dimensional structure^{12,13}. Here, we replicate this finding in humans, but in contrast to prior work in NHPs that has revealed context-dependent computations in adjacent premotor structures^{5,14,17,18}, we found no evidence that the human motor cortex encodes predictive context. We observed that motor cortex relies on input from PFC, which encodes both context as well as the current action plan. However, due

to the inherent limited coverage in intracranial recordings, we cannot preclude that specialized sub-regions within the human motor cortex (e.g. anterior or posterior parts of the supplementary motor area, frontal eye fields, premotor cortex) might also encode contextual information."

Reviewer comment #19:
Thanks for the clarifications.

3- The correlation analysis in Figure 5e can also be confounded by the fact that high-uncertainty trials had longer RTs... the differences in correlation values could be a simple reflection of a larger number of points being included in the correlations of longer-RT trials.

Thank you for the opportunity to clarify. We time-locked the data from -0.5 to +0.3s with respect to the button release across all conditions. Therefore, the correlation analysis is not confounded by a difference in the number of time points considered. In the revised manuscript, we now make this point explicit.

Page 52, Line 14-18

"We computed the correlation coefficient between PC single-trials in PFC and motor cortex from -0.5s to 0.3s with respect to movement onset. This ensured that all trials contained an equal number of data samples, thereby avoiding potential confounds in the correlation value simply due to variable reaction times across trials."

Reviewer comment #20:
Noted, thanks.

4- PCA methods: There is not enough information in the methods to understand what was done with the PCA analysis. Were the time bins aligned to movement, or to task events? Were all trials (across all contexts) used to build the space, or were spaces built on context-specific activity? Which time bins were included in the PCA analysis?

We apologize for the lack of clarity on how we performed the PCA. We performed PCA on the movement-locked data (button release). For the revised version of the manuscript, we now also performed PCA on the data time-locked to the HLL (**Supplementary Figure 5**). We used all trials to build the PC-space. We have added this critical information into the methods section on dimensionality reduction.

Page 49, Line 12-19

"We used principal component analysis (PCA) to identify linearly uncorrelated population activity patterns and construct a low-dimensional manifold that is embedded in the neural state space spanned by the recorded depth electrodes. We performed PCA on a two-dimensional data matrix (channel x time, trial) locked to either the HLL or to the movement onset. All trials were used to construct the PC-space. The resulting matrix (component x time, trial) was then reshaped into a three-dimensional matrix (trial x component x time) which allowed us to perform single trial analysis in PC space."

Reviewer comment #21:
Noted, thanks.

5- Answers to reviewers mostly address their issues, except for Reviewer 2, point 1. All those analyses are also consistent with difficulty being the main factor. At the very least this should be discussed as a confound in the paper, rather than brushing it off.

Thank you for this comment. We did not intend to brush the argument off. We agree that it is not possible to entirely and unambiguously disentangle uncertainty from task-difficult, but we performed additional analyses to provide evidence in support of this consideration. We would like to point out that a shift along the internal response axis (criterion) is indicative of an alteration in the participant's decision strategy³⁹. However, we fully acknowledge that an additional paragraph is needed to address this limitation. In the revised version of the manuscript, we have added a paragraph emphasizing this limitation.

Page 39, Line 6-14

"It is important to acknowledge that the environment of intracranial EEG recordings precludes long experiments with many control conditions. Based on our design, we cannot fully disentangle behavioral uncertainty from overall task difficulty. We directly addressed this limitation by calculating the SDT as well as linear ballistic accumulator models to quantify the participants' response strategy. However, the observed shift in criterion as a function of uncertainty could also be explained by an overall shift of the signal and noise distribution along the internal response axis that would not involve any change in participants' response strategy."

Reviewer comment #22:
Thank you for the addition.

References

- 1 Kaufman, M. T. *et al.* The Largest Response Component in the Motor Cortex Reflects Movement Timing but Not Movement Type. *eNeuro* **3**, doi:10.1523/ENEURO.0085-16.2016 (2016).
- 2 Ter Wal, M. *et al.* Human stereoEEG recordings reveal network dynamics of decision-making in a rule-switching task. *Nat Commun* **11**, 3075, doi:10.1038/s41467-020-16854-w (2020).
- 3 Rigotti, M. *et al.* The importance of mixed selectivity in complex cognitive tasks. *Nature* **497**, 585-590, doi:10.1038/nature12160 (2013).
- 4 Parthasarathy, A. *et al.* Mixed selectivity morphs population codes in prefrontal cortex. *Nat Neurosci* **20**, 1770-1779, doi:10.1038/s41593-017-0003-2 (2017).
- 5 Aoi, M. C., Mante, V. & Pillow, J. W. Prefrontal cortex exhibits multidimensional dynamic encoding during decision-making. *Nat Neurosci* **23**, 1410-1420, doi:10.1038/s41593-020-0696-5 (2020).
- 6 Hauber, W. Involvement of basal ganglia transmitter systems in movement initiation. *Prog Neurobiol* **56**, 507-540, doi:10.1016/s0301-0082(98)00041-0 (1998).
- 7 Philipp, R. & Hoffmann, K. P. Arm movements induced by electrical microstimulation in the superior colliculus of the macaque monkey. *J Neurosci* **34**, 3350-3363, doi:10.1523/JNEUROSCI.0443-13.2014 (2014).
- 8 Pesaran, B., Nelson, M. J. & Andersen, R. A. Free choice activates a decision circuit between frontal and parietal cortex. *Nature* **453**, 406-409, doi:10.1038/nature06849 (2008).
- 9 Murakami, M., Vicente, M. I., Costa, G. M. & Mainen, Z. F. Neural antecedents of self-initiated actions in secondary motor cortex. *Nat Neurosci* **17**, 1574-1582, doi:10.1038/nn.3826 (2014).
- 10 Chiang, F. K., Wallis, J. D. & Rich, E. L. Cognitive strategies shift information from single neurons to populations in prefrontal cortex. *Neuron* **110**, 709-721 e704, doi:10.1016/j.neuron.2021.11.021 (2022).
- 11 Haller, M. *et al.* Persistent neuronal activity in human prefrontal cortex links perception and action. *Nat Hum Behav* **2**, 80-91, doi:10.1038/s41562-017-0267-2 (2018).
- 12 Gallego, J. A., Perich, M. G., Miller, L. E. & Solla, S. A. Neural Manifolds for the Control of Movement. *Neuron* **94**, 978-984, doi:10.1016/j.neuron.2017.05.025 (2017).
- 13 Suresh, A. K. *et al.* Neural population dynamics in motor cortex are different for reach and grasp. *Elife* **9**, doi:10.7554/eLife.58848 (2020).
- 14 Thura, D. & Cisek, P. Deliberation and commitment in the premotor and primary motor cortex during dynamic decision making. *Neuron* **81**, 1401-1416, doi:10.1016/j.neuron.2014.01.031 (2014).
- 15 Thura, D. & Cisek, P. Modulation of Premotor and Primary Motor Cortical Activity during Volitional Adjustments of Speed-Accuracy Trade-Offs. *J Neurosci* **36**, 938-956, doi:10.1523/JNEUROSCI.2230-15.2016 (2016).
- 16 Wang, M. *et al.* Macaque dorsal premotor cortex exhibits decision-related activity only when specific stimulus-response associations are known. *Nat Commun* **10**, 1793, doi:10.1038/s41467-019-09460-y (2019).
- 17 Glaser, J. I., Perich, M. G., Ramkumar, P., Miller, L. E. & Kording, K. P. Population coding of conditional probability distributions in dorsal premotor cortex. *Nat Commun* **9**, 1788, doi:10.1038/s41467-018-04062-6 (2018).
- 18 Mante, V., Sussillo, D., Shenoy, K. V. & Newsome, W. T. Context-dependent computation by recurrent dynamics in prefrontal cortex. *Nature* **503**, 78-84, doi:10.1038/nature12742 (2013).
- 19 Iemi, L., Chaumon, M., Crouzet, S. M. & Busch, N. A. Spontaneous Neural Oscillations Bias Perception by Modulating Baseline Excitability. *J Neurosci* **37**, 807-819, doi:10.1523/JNEUROSCI.1432-16.2016 (2017).

Point-by-Point Reply: NCOMMS-22-05636-Z

We would like to thank the reviewer for their thoughtful and detailed assessment of our revised manuscript. Below, we provide point-by-point responses to all remaining queries. All changes in the revised manuscript are highlighted with tracked changes.

Reply formatting guide:

Reviewer remarks: **bold, black**.

Author response: **blue**.

Author response in revision #3: **green**.

Changes in the manuscript: *italics, blue*.

Please note that the reviewer directly added their comments into the rebuttal. Therefore, some of their comments in the current revision directly refer to our responses from the previous revision. Hence, we incorporated our excerpts from previous answers in some instances to contextualize the reviewers' comments.

We would also like to highlight that several queries concern the same matter, hence, we provide cross-references between the different queries. In addition, we incorporated several changes into the main manuscript to nuance our statements and provide a balanced view when multiple interpretations are possible. These changes are not listed in their entirety in this rebuttal, but are highlighted with tracked changes in the revised manuscript to facilitate readability.

Reviewer #3

Comment #1:

Our response in revision #3:

"We fully agree with the reviewer that a go cue does not need to be the appearance of a sensory stimulus. However, it is not correct that the lower limit was occluded by the ascending bar as the lower limit was visible throughout the entire trial. Hence, the only sensory information that was dynamically changing during the trial was the ascending bar; the rest remained constant."

Noted on this. This clarification, while useful, does not imply that this task did not have a "designated go-cue". If it helps, you can think of the gap between the ascending bar and the LL disappearing as the go-cue. The key point here is that the subjects received an instruction to initiate the movement as soon as certain sensory event occurred, which is the very definition of a go-cue. I would suggest that the authors add a statement in the paragraph highlighted below to clarify this.

We thank the reviewer for this additional remark and followed the suggestions to add a statement that highlight this consideration. We agree that an instruction to initiate a

movement can be considered a go-cue by itself. We clarified this matter in the revised version of the manuscript.

Page 29, Line 7-11:

"Furthermore, while we did not employ a characteristic go-cue (i.e. sudden appearance of a sensory go-cue) in our experimental paradigm, the moment at which the bar reaches the lower limit (cf. Fig. 1a) still resembles a go-cue and could potentially trigger condition-invariant activity changes."

Comment #2:

Our response in revision #3:

"We respectfully disagree. First, the change in activity upon the lower limit should not be specific to the 0% condition, but should instead be observable across all conditions (0%, 25% & 75% likelihood of stopping)."

Yes, agreed, and I didn't mean to imply otherwise.

We thank the reviewer for the approval.

With one caveat: POINT 1. It should be observable across all conditions, so long as the go-cue triggers the movement. In this task, many (most?) of the movements were not triggered by the HLL (in particular for the 0% condition). To assess HLL-triggered activity, only HLL-triggered saccades should have been assessed.

We fully concur that this remaining caveat needs to be considered.

First, we specifically want to clarify the reviewer's assertions that "only HLL-triggered saccades" should be analyzed, which constitutes a misunderstanding. The participants were not asked to make a saccade in this task, but instead, pressed a button to start the trial. We only analyzed trials where a button release occurred after the HLL.

This has been clarified in the methods.

Page 37, Line 1-2:

"Collectively, we excluded trials in which participants released the button prior to the lower limit."

The caveat that the reviewer is referring to is that in the current task design it is challenging to disentangle movement planning from execution. A clear separation was not possible in the current study due to a lack of additional objective measurements of muscle activity by electromyography to determine the precise movement onset. We addressed this limitation in the revised version of the manuscript.

Page 25, Line 6-12:

"However, we cannot completely dismiss the possibility that action-related information preceding the lower limit might merely reflect different mental states, such as discrete phases of movement (planning vs. execution). Therefore, future studies should simultaneously record from both (sub)cortical regions and electromyography to fully untangle the spatiotemporal gradient

between movement planning and movement execution from higher-order association to sensorimotor areas.”

Comment #3:

Our response in revision #3:

“However, as can be clearly seen in Figure 2a, the rise in activity for trials with a 25% and 75% likelihood of stopping does not coincide with the lower limit, but occurs before that.”

Indeed, Figure 2a shows that around 300ms before the HLL there is an increase in activity for the 75% condition. This observation, however, is consistent with an HLL-triggered change in activity, since there could simultaneously exist pre-HLL and HLL-triggered changes in activity.

We thank the reviewer for this comment. We fully concur that activity ramps up prior to the HLL (as quantified in detail in Figures 2-3). While we interpret this signal as a context-dependent signal, which is behaviorally-relevant, it *could* also be interpreted as a HLL-triggered activity change. In order to provide a balanced perspective that incorporates both perspectives, we address this as a potential caveat in the revised version of the manuscript.

Page 29, Line 18 – Page 30, Line 4

However, changes in neural activity both before as well as after the lower limit might also be explained by the lower limit itself. Hence, the activity ramp-up in PFC prior to the lower limit might reflect a mixture of coexisting context- and go-cue-dependent activity. Future studies might resolve this limitation by simultaneously recording both brain and muscle activity. However, it is also possible that distinguishing context- and movement-dependent activity in the PFC cannot be fully disentangled given the involvement of the PFC in multiple operations. Future studies that employ experimental designs that are geared towards disentangling context- and go-cues are necessary to separate purely cognitive representation from motor preparation and execution.

Comment #4:

Our response in revision #3:

Furthermore, we computed the average HFA activity before and after the lower limit for the 0% condition to test whether the HFA activity significantly differed between the two time periods. This revealed that HFA activity did not change from before to after the lower limit (Response Fig. 1); hence rendering it very unlikely that the lower limit triggered large activity changes.

As discussed in POINT 1 above, this observation does not address the question, since most of the 0% condition saccades are initiated prior to the HLL.

This concern is addressed in detail in the previous comments (comments #2 & #3). We would further like to highlight that no saccades were made to avoid any misunderstanding about the nature of the task. We modified the manuscript accordingly as outlined in our previous responses.

Comment #5:

Our response in revision #3:

"Instead, this finding supports our statement that the human PFC was not 'recruited' during trials with no uncertainty (in line with no significant changes in the ramping slope; cf. Figure 3a)."

I think this result is solid.

We thank the reviewer for this positive assessment.

Comment #6:

As discussed above, it could be argued that the 0% condition is the least important condition to look at for this analysis, since subjects are not waiting for the HLL (go-cue) in this condition to initiate their movements (as revealed by their extremely fast RTs - page 6, line 4: mean RT was 43.9ms**). Normally, RTs are >150ms after a go-cue in non-predictive movements. That being said, I presume that in a subset of 0% trials the subjects did indeed wait for the go-cue (I don't have a sense of this since the authors have not provided a distribution of RTs for each condition, but rather only provided a distribution across all conditions – response to reviewers – and a distribution of subject averages – Fig 1c).

We thank the reviewer for this comment and would like to clarify that the 'subset' of trials that the reviewer is referring to were in fact almost all 0% trials, since the participants were tasked to only release the button after HLL. This experimental manipulation worked as indicated by the high accuracy in the 0% condition (>> 90%; c.f., Fig. 1c center panel).

We apologize for the confusion regarding the reaction times. The following sentence "*We found that reaction times (RT) gradually increased as a function of uncertainty (Fig. 1c; +43.9ms ± 19.8ms, mean ± SD [...])*" indicates that there is an increase in reaction time of (plus) +43.9ms from the condition with a 0% to the condition with a 75% likelihood of stopping. We fully agree that the sentence is confusing and leads to misinterpretation. Therefore, we now provide the mean ± SD for each condition (both for accuracy and reaction time).

Page 6, Line 6-11 :

"We found that reaction times (RT) gradually increased as a function of uncertainty (Fig. 1c; 0% = $79.92 \pm 19.66\text{ms}$; 25% = $94.98 \pm 23.86\text{ms}$; 75% = $123.81 \pm 29.34\text{ms}$; mean \pm SD; $F(2,36) = 58.99$, $p < 0.001$, $\eta_p^2 = 0.77$; one-way RM-ANOVA). Participants were also less accurate in trials with high uncertainty (Fig. 1c; 0% = $92.08 \pm 6.99\%$; 25% = $81.1 \pm 8.33\%$; 75% = $71.39 \pm 5.46\%$; mean \pm SD; $F(2,36) = 81.53$, $p < 0.001$, $\eta_p^2 = 0.82$)."

The reviewer is correct that we failed to show RT distributions across all conditions, which are now displayed below.

Response Figure 1. The distribution depicts the movement onset with respect to the lower limit (HLL) for the different contexts (green = 0%; orange = 25%; red = 75%).

If that is the case, then some HLL-triggered activity could be observed. Based on Figure 2 (copied below) there does appear to be an early peak in the green trace (0%). Response Figure 1 above likely missed this because it used an average activity for the post- HLL of 0-300, which is too large for the potential HLL-triggered activity (which appear to occur around 50-200ms after the go-cue).

This comment again refers to the HLL and its interpretation as outlined in detail in comments #2 and #3. As clarified in the previous comment, the participants were tasked to only release the lever after HLL and so no trials with premature button releases were included. The limitations have been addressed in detail in previous revisions and can be found in the discussion.

Moreover, we repeated and expanded the analyses based on the updated time window (50-200ms) as suggested. This query also pertains the next comment and the control analysis is reported below.

I think the most obvious place to look at pre- vs post- HLL HFA should be made in the most uncertain condition, which is the condition in which subjects are most likely to wait for the HLL to initiate their movement. Based on Fig 2a, copied below, the conclusions would be quite different:

**** although see comment below regarding the inconsistency of this RT value and those shown in the plot**

We thank the reviewer for this remark and directly tested for pre-vs post-HLL HFA differences as requested (updated time window as suggested ranging from time window from 50-200ms).

However, we need to highlight that the results from this test can be directly inferred from the univariate results that were reported in Figure 2/3. In this panel, we report that HFA activity is ramping up (which encompasses the HLL), hence, the analysis of averaged HFA necessarily needs to show a difference between pre- and post-HLL episodes only for the 25% and 75% condition (**Response Fig. 2**).

We thereby confirmed our initial analysis that there is no significant change from pre to post lower-limit for condition with a 0% likelihood of stop. We observed an increase in HFA from pre to post lower-limit for trials with a 25% and 75% likelihood of stop as the direct consequence of the ramping activity prior to the lower limit (as extensively quantified in Figs. 2a/3a).

Response Figure 2. We computed the average HFA before (-0.5 to -0.01s) and after (0.05 to 0.2s) the lower limit for all contexts separately. **a** Change in activity before and after the lower limit for trials with a 0% likelihood of stop ($p = 0.255$; Wilcoxon signed-rank test; two-tailed). **b** Same as (a), but for trials with a 25% likelihood of stop ($p = 0.026$). **c** Same as (a), but for trials with a 75% likelihood of stop ($p = 0.03$).

These additional findings indicate that there was no significant activity change triggered by the lower-limit for the condition with no uncertainty. However, as remarked by the reviewer in previous comments and fully acknowledged by us, we cannot fully rule out that activity changes prior to the lower limit partially reflect changes directly triggered by the lower limit. We have addressed this in detail in previous comments. We fully acknowledge that the precise interpretation constitutes the source of all queries regarding the HLL, hence we very carefully nuanced our language throughout the manuscript to appropriately and adequately describe our results.

In sum, we concur that it is conceivable that context- and movement-dependent activity preceding the lower limit can coexist. We incorporated this into the manuscript, the changes were described in response to comments #2 and #3.

Comment #7:

I thank the authors for the effort placed in this answer.

We appreciate the reviewers' approval.

However, it is a whole set of analyses, for which there aren't very many details to be able to thoroughly assess.

For instance, which trials were included? (only 0% or all?).

We apologize for the lack of detail. In the previous version we included all trials.

Was Levene's test done on the normalized values? Or the second plot (un-normalized?). Given the large variability in the values of the different subjects (y-axis in single-subjects plots ranging from 1-200V), I would hope the test was done on the normalized values (although based on the description it appears that it was done on the un-normalized values instead).

Levene's test was performed on unnormalized and normalized values, which in both cases, did not yield significant difference in variance. We report the results for the Levene's test on normalized values below (after the next query), given the overlap between both queries.

Further, while interesting to see the single-subject plots, these are noisy and likely underpowered for statistical assessments, so I am not sure that we can use the lack of stats significance observed in the majority as a strong argument.

We concur with the reviewer, since the single subject plots were used for illustration and not for statistical inference.

To address previous queries, we recomputed the ERPs and performed the Levene's test separately for the three conditions after normalization (subtraction of the mean). This revealed that there was no statistically significant increase in variance from pre- to post-lower limit (**Response Fig. 3**). Hence, we conclude that there is no increase in variance upon the lower limit.

Response Figure 3. Event-related potentials (ERP) locked to the lower-limit for the different conditions. **a** ERP across all conditions. The variance did not increase from pre- to post-lower-limit ($p = 0.166$; Levene's test). **b** Same as **(a)**, but for trials with a 0% likelihood of stop (variance pre vs. post: $p = 0.076$). **c** Same as **(a)**, but for trials with a 25% likelihood

of stop (variance pre vs. post: $p = 0.153$). **d** Same as **(a)**, but for trials with a 75% likelihood of stop (variance pre vs. post: $p = 0.113$).

Comment #8:

As discussed above, I have my reservations regarding response figure 1. The observation that the 25% and 75% uncertainty already ramps up does not imply that there is no HLL-triggered change in activity. They can both co-exist.

We thank the reviewer for the comment and fully concur that most likely, both scenarios are not mutually exclusive but may as well coexist. We acknowledge this consideration in the discussion and in several previous queries. We made several modifications in the manuscript to provide a balanced perspective, including the statements that were reported for comment #3.

And indeed, single trial dynamics should not be used as evidence in this case (I am sure you can find single-trial examples in your data that would support any claim you'd like to make.

As stated above (comment #7), we fully concur and single trials should not be used instead of statistical inference. Therefore, we report the outcome of the statistical tests at group level.

Comment #9:

Our response in revision #3:

"We followed the reviewer's suggestion and performed the analysis now on multiple PCs (1 to 5). None of the PCs showed an activity change upon the lower limit (Supplementary Fig. 6)."

Again, this is useful, but with limited description of the analysis it is hard to interpret... were all conditions included? Were all RTs included? PC₃ looks like it contains a sharp change after HLL. How is this changed deemed non-significant?

We thank the reviewer for the comment and apologize for the lack of detail, which we clarify here.

1. The analysis included all trials. As outlined in comment #6, the reason behind using all trials/conditions was to test whether there is a condition-invariant change upon the lower limit.
2. Also, all reaction times were included.
3. The reviewer is referring to a sharp change in the first derivative of PC₃ (PC₃ itself shows a ramp). The sharp change is in fact a negative deflection that indicates that the ramping signal reached a plateau and drops again after the HLL, but it does not signify a sharp activity increase. In the time domain, PC₃ (analogously to the reports in the main manuscript and PC₁) exhibits a ramp that starts prior the HLL. In response to the previous comment, we did not perform statistical testing on the traces, given that ramping activity will necessarily be reflected in an activity difference (cf. comment #6) and is therefore, uninformative regarding the question if there was an isolated sharp peak after the event.

The nature of the ramping activity has been quantified in detail in the main manuscript Figures 2-4.

Comment #9:

I thank the authors for their thorough responses. To be honest, I am torn here. I do appreciate the limitations of human studies (and the enormous benefits as well). I appreciate the considerable effort placed in addressing the concerns raised. However, I am still worried that these analyses may be mixing different cognitive operations within the same analysis, and assigning the found differences to “contextual differences”, when in fact they may reflect something else (i.e. motor preparation for long RT trials vs motor execution for short RT trials).

We appreciate the thorough response and we fully acknowledge the concerns. We extensively consider the option of motor preparation during the pre-HLL epoch; which we incorporated into the manuscript as outlined in response to the previous comments.

Even if it is true that the HLL does not trigger an activity change (which, as discussed above, is not clear), the concern I raised originally would stand... given the differences in RT across conditions, aligning responses to the HLL will inevitably lead to analyses that match different mental states (i.e. pre and post movement).

We would like to thank the reviewer for the constructive feedback. We concur that alignment to HLL has its inherent limitations, which is acknowledged and discussed in many instances in the manuscript. We would also like to highlight that we additionally report results that were aligned to the BR (button release; in Figure 1d, Figure 5 and 6 and Supplementary Figures 8-10), which provides a complementary perspective to mitigate concerns regarding the HLL. As outlined in previous comments, the interpretational issues around HLL and the BR cannot fully be resolved, hence, we are very careful with all interpretations and nuanced the wording in statements in several instances (outlined in previous comments), for example in the results, where we state:

Page 16, Line 16-21:

"We time-locked the population response to the timing of movement execution to mitigate the impact of distinct cognitive states at the HLL during different contexts. Thus, neural activity that dissociates reaction times prior to movement execution (and that is not primarily driven by early context-integration signals) likely reflects the internal transition from the planning to the execution of a movement."

To give a very specific example, let's consider the -0.2 time point in Fig2a:

As seen in Fig1c, for 0% uncertainty trials (green) subjects were very fast (average RTs <100ms), which means that the HFA data at -0.2 includes a majority of trials in which the subjects are preparing the movement execution. On the other hand, for 75% uncertain trials subjects were comparatively slow (RTs >100ms), which means that the HFA data at -0.2 includes a majority of trials in which the subjects have not prepared the movement execution, but rather are preparing a potential movement pending the HLL. So this (and future plots) are comparing very different cognitive states as if they were comparable.

We understand the reviewers' concern regarding the precise timing effects around the HLL. This inherent limitation is mitigated in all analyses where we time-locked to the button release (BR). We used this analysis to complement the analyses relative to the HLL. When time-locked to BR, this is per definition time locked to the very same 'mental state'.

Moreover, as outlined above and extensively in the previous revision, is that the notion that contextual differences can be fully attributed to "*motor preparation for long RT trials vs motor execution for short RT trials*" is not in accordance with several results. However, we concur that motor preparation could potentially contribute to the signal. Here we briefly summarize a few key points that support our interpretation; however, as stated in the previous responses, we provide a balanced perspective in the discussion:

- a) Participants could only execute the movement plan once the bar reached the lower limit, otherwise the trial was invalid and not included in the analysis.
- b) We do not find evidence for differences in motor execution in motor cortex. if this context-dependency would fully correspond to a motor component, then this should also be directly reflected in motor cortex itself.
- c) We were able to decode both context and action in PFC, whereas only action could be decoded in motor cortex (cf. Fig. 5g; Supplementary Fig. 9). Hence, if the neural computation that we capture were only related to 'low-level' movement preparation, then we should not be able to dissociate context and action in motor cortex.

We now explicitly discuss the concern of mixing distinct cognitive operations in the revised version of the manuscript (see response to comment #2).

2) Throughout the review process, we realized that the usage of the word 'context' can be misleading as was vaguely defined. In the revised version of the manuscript, we now clearly define the word 'context' early on.

Introduction: Page 4, Line 4:

[...] contextual information is rule-based and not sensory-driven.

Introduction: Page 4, Line 16-18:

Here, we defined context as the currently active rule, which exhibited predictive information about a subsequent action.

Results: Page 5, Line 11-14:

A predictive cue defined the context for the current trial by signaling the likelihood of a stop trial (green circle = 0%, orange circle = 25%, red circle = 75%). We refer to the stop likelihood as behavioral uncertainty or predictive context and use these terms interchangeably.

Comment #10:

Our response in revision #3:

"The reviewer correctly remarks that there are "many trials in which subjects responded faster than 100ms after the HLL" and that these reaction times are "too fast to be triggered by the go-cue" and that these reaction times are "predictive in nature". At this point, we would like to emphasize that this is a key finding of our study and the main reason that the experiment was designed as such. [...] hence resulting in very fast reaction times which are, as correctly remarked, "predictive in nature"."

I understand that the task was designed to have predictive movements in the zero-uncertainty condition. However, this is not a finding per-se, since this could be expected based on the task.

We thank the reviewer for this remark. We need to respectfully disagree with this statement. Based on a hypothesis, we designed a task, which yielded a significant manipulation that we quantify and link to neural correlates. Hence, we do not understand how this can be viewed as 'not a finding per-se'. To avoid any misunderstanding, in the revised manuscript, we highlight that those trials where participants performed a premature movement before the HLL were excluded.

Page 37, Line 1-2:

"Collectively, we excluded trials in which participants released the button prior to the lower limit."

It seems to me that it is trivial that RTs can be "decoded", given that for slow RTs the subjects are not executing the motor commands prior to the HLL, whereas for fast RTs they are.

We thank the reviewer for this remark and would like to clarify a key misunderstanding. A key aspect of our analysis is not to decode if they are faster, but (a) when can a classifier distinguish different RTs and (b) where is this information represented neurally. Moreover, the decoding approach enables dissociating the variable that we termed context from the overt action. Critically, we observed a clear temporal evolution where context and action do not arise simultaneously. Moreover, we do not find any evidence for contextual coding in motor cortex; thus, rendering it unlikely (albeit, not impossible) that this only reflects movement preparation. We added a sentence to the manuscript to reflect this consideration:

Page 16, Line 13-16:

Note that we were not primarily interested in distinguishing fast vs. slow RTs, but employed the classifier as a tool to trace the temporal evolution of the representation of the overt behavioral response.

Comment #11:

Our response in revision #3:

"We agree that the decoding might capture direct action, rather than solely action-planning processes. However, this is also not critical for the main message that the current paper conveys;"

I think this is my issue with this figure: why is it included if it is not critical to the main message? I am still not sure what we learned with this additional figure.

We apologize for the confusing statement that in hindsight, was unnecessary. Supplementary Figure 7 shows that there are action-related signals prior to the lower limit. Hence, this suggests that we likely capture action-planning, rather than action execution processes (as trials were invalid when the button was released prematurely). The figure was added in response to a query in a previous version to a specific point, namely whether we can dissociate pure action planning from movement execution. As discussed previously, and acknowledged in several queries (e.g., #2 and #3), the current task design does not enable to fully untangle planning and execution processes. This is acknowledged in the discussion:

Page 29, Line 5-7:

"Second, based on the current experimental design, we cannot fully dissociate between preparatory- and movement-related computations."

Comment #12:

Our response in revision #3:

"[...] namely that contextual information is integrated in the human prefrontal cortex by means of a ramping processing mode whereas the inter-areal communication to guide context-dependent action is facilitated by means of an oscillatory communication mode."

The previous figures attempt to make these points (and we discuss them separately in elsewhere), but this decoding figure does not contribute, as far as I can tell, to either of these conclusions.

We respectfully disagree, since the decoding figure provides important insight into the timing of the effects. However, we concur, that preferences are subjective and therefore, we added the following statement:

Page 16, Line 3-6:

"In order to better visualize and contextualize the univariate results (Figures 1-4) and to identify and to identify behaviorally-relevant factors in the state space, we a multivariate approach to trace the temporal evolution of different behaviorally-relevant factors along the prefrontal-motor hierarchy."

Comment #13:

Our response in revision #3:

In the current study, we won't be able to fully disentangle preparatory signatures from direct motor commands as the transition between the two reflects a continuum.

Not necessarily, as the work of Churchland, Shenoy, Svoboda and others shows. There may be an abrupt transition from preparing a movement to executing it, and this abrupt transition occurs around ~100ms prior to the movement execution.

Thanks. We fully agree. We therefore removed this sentence in the revised version of the manuscript and added the following statement:

Page 26, Line 13-18:

"Neural activity in motor cortex exhibited large, context-independent activity changes preceding movement initiation, a pattern that is consistent with prior studies reporting abrupt shifts in activity shortly before movement onset¹. This context-independent activity, previously also referred to as a condition-invariant signal (CIS)², typically reflects the largest response component in motor cortex²."

Comment #14:

Our response in revision #3:

However, we would like to emphasize that the classifier was trained to discriminate fast, medium and slow reaction times. If the classifier would only decode the execution of very fast movements (33% of all reaction times), then the classifier should fail to predict reaction times above chance.

I don't understand the point being made here. The LDA classifier dissociates 3 categories (fast, mid, slow). How could the classifier only decode fast movements?

We apologize for the lack of clarity. The reviewer initially (revision #3) was concerned that the decoding of reaction time is simply triggered by the "*very fast movements*". However, if the classifier would only be able to distinguish fast vs. slow/medium reaction times, then the classifier should drop to chance level when distinguish slow from medium RT trials. This was not the case in here – instead the decoding of action followed a clear build-up over time. This has similarly been observed in prior studies using human iEEG³.

Comment #15:

Unfortunately, you have not shown the RT distributions of any of the conditions, so it is hard for me to evaluate this claim.

We now provide the reaction time distribution separately per condition in **Response Fig. 1**.

Comment #16:

Our response in revision #3:

Moreover, if the action decoder would be fully driven by simply capturing 'different time periods of the task', then the same decoding traces should have been emerged for context-decoding since reaction times strongly differ between context conditions. This is not the case (cf. Fig. 5g; Supplementary Fig. 9) and has also been previously addressed in the first round of revision (reviewer #2, comment #6).

This claim rests on the orthogonalization method employed. The authors selected an unconventional method, which is to carry out LDA separately on different PCs, and use the highest decoding performance for RT and context as the coding dimension. This analysis ignores the potentially important information found in the non-top dimensions. A more conventional approach would be to use the full PC space to run the LDA and impose an orthogonality constraint on the hyperplanes. Alternatively, they could have run an LDA for RT on the full space, and then a second LDA for context in the residuals (and vice versa). It is perfectly possible that the analyses that I outline above would lead to context signals closer to HLL.

We thank the reviewer for this comment. In addition to the 'unconventional' analysis, for which we have previously explained the motivation in detail (a fact that was previously acknowledged and approved; see reviewer comment #14 in revision #3), we also employed the decoding analysis in high-dimensional space without orthogonality constraints (as suggested by the reviewer in revision #2). This yielded the highly comparable results for the decoding of context and action (cf. Supplementary Fig. 9). Hence, it is unlikely that the decoding of context early on was a consequence of the employed method.

Comment #17:

Our response in revision #3:

We fully agree that the reviewer's suggestion is an excellent idea. However, this is unfortunately not possible simply due to the fact that reaction times strongly differ between context conditions (the main hypothesis of the manuscript) and the resulting number of trials that could be used for the analysis would be too low. This is a direct consequence of clinical recording settings and an inherent limitation of human intracranial recordings. We have added this limitation into the main text:

This is fine. However, if there are limitations in the methods, the conclusions should be consistent with these limitations.

We thank the reviewer for this remark. We nuanced our language to be more descriptive throughout the manuscript, including changes in the title, abstract and throughout the main text, which we highlighted with tracked changes.

Comment #18:

Given the considerations above, unfortunately I remain unconvinced, and my concern stands.

1- BR-aligned analyses may suffer from confounds of HLL-triggered activity (I am willing to suspend my disbelief here, but even if there are no HLL-triggered changes in activity, points 2 and 3 stand).

We concur that this *may* constitute a confound and we provided substantial additional analyses to substantiate our interpretations. Therefore, as outlined above, we actually provide HLL and BR-locked responses throughout the manuscript.

During the review process it became evident that this point cannot be fully disentangled and led to substantial debate. We have acknowledged and discussed the shortcomings in detail throughout the revision process. We briefly outline several points that led us to conclude that potentially coexisting changes cannot fully explain our results and provide a balanced discussion.

- (1) Activity in PFC revealed a significant ramp-up preceding the lower limit and was context-dependent (cf. Fig. 2a; Fig. 3a; Supplementary Fig. 5). This contrasts with the condition-invariant signal observed in motor and somatosensory regions, as seen in previous research in animal models. However, it is important to acknowledge that the activity preceding the lower limit may still partially encompass dynamics associated with movement. We changed the manuscript accordingly as outlined in comment #3.
- (2) We did not find evidence for a change in activity from pre- to post-lower-limit in trials with no uncertainty. We acknowledge that the rise in activity after reaching the lower limit in trials with moderate and high uncertainty, might, to some degree, be attributed to changes triggered by the go-cue. Nevertheless, in support of the idea of concurrent dynamics, the increased HFA observed in these trials likely also signifies context-dependent activity prior to the lower limit, as extensively quantified in Fig. 2a and Fig. 3a.
- (3) Although we did not find any indications of alterations in activity at the lower limit in trials without uncertainty, in accordance with the time window recommended by the reviewer (Response Fig. 1), we acknowledge the rise in activity after reaching the lower limit in trials with moderate and high uncertainty (see point (2)).
- (4) We did not find evidence for a strong phase-locked response upon the lower limit (cf. Response Figure 3).
- (5) Throughout the first 5 principal components in PFC, we did not find any that revealed a strong change in activity upon the go-cue as previously observed by Kaufman et al. 2016 in NHP motor cortex (i.e. compare Supplementary Figure 6 in our manuscript to Figure 3 in Kaufman et al. 2016). This could either reflect species differences and/or differences between motor cortex and higher order areas, such as the prefrontal cortex.

We made several changes in the manuscript to incorporate these considerations, which are highlighted with tracked changes and not listed here, given that they pertain multiple paragraphs.

2- HLL-aligned analyses suffer from confounds of comparing different cognitive processes (movement planning vs movement execution).

We discussed this query in detail in comments #2 and #3. This concern has also been addressed in response to query #9.

3- The decoding analysis does not add much to the main conclusions of the paper.

We respectfully disagree and would rather argue that the multivariate analyses (1) replicate the univariate analyses and (2) extent their scope given that they enable tracing information representation along the PFC-motor cortex hierarchy. Moreover, in comment #23 the reviewer states '**which now accurately reflect the conclusions of these analyses**', referring to our description of the information flow from PFC to motor cortex, which was the direct result of the multivariate approach.

Thus, the multivariate analyses allowed us to better understand the succession of different events along the prefrontal-motor hierarchy. This is highlighted in the manuscript:

Page 28, Line 11-16:

"We found that PFC settled into a low-energy state (smaller magnitude, only covering a limited subspace of the entire state space) during states of high predictability. Critically, these patterns could only be observed using multivariate analysis strategies (Fig. 5; cf. Fig 2 for the univariate approach) that take coordinated variability across different recording sites into account."

Comment #19:

Our response in revision #3:

As mentioned above, after receiving the contextual cue, participants self-paced the trial start. Therefore, the contextual cue is not necessarily given 560-760ms prior to the lower limit, but as stated above: "Upon receiving the predictive cue, participants were able to start the trial in a self-paced manner by pressing the space bar on the keyboard (average time to start the trial: $1.76s \pm 0.55s$; mean \pm SD)..."

Indeed I did not realize that the bar started moving at a variable time after the context cue. This, however, does not preclude the comment above. Again, I will suspend my disbelief about the HLL-triggered signal for a minute, to highlight that the problem still persists.

We thank the reviewer for acknowledging that there was a misunderstanding regarding the precise task instructions. The remaining problem regarding activity at the HLL has substantially been discussed in previous revisions and comments, but we carefully incorporated responses to remaining queries below.

A subject that views a 0% uncertainty trial, will press the spacebar and see that the bar starts ascending towards the LL. During this period, the subject knows that there will be a movement as soon as the bar reaches the LL. This context, which exists throughout, and is presumably reflected in the activity somewhere in the brain (quite likely in the PFC, as has been observed in numerous previous studies in NHPs), means that subjects will start preparing for an impending movement. None of these signals are identified in the current study, since the context predicting signals only emerge ~300-200ms prior to LL. This is fine, since it may reflect species-, task-, region-, or recording-method differences. Regardless, cognitively, the context is present in the subject.

We fully concur with the statement, which adequately summarizes the task instructions and which is reflected in our description in the Methods.

A similar description applies to the 25% and 75% contexts, the only difference being that the task is “different” for these cases... the subjects have to be more alert, and pay closer attention to the bar movement (this is my guess as to why the uncertain contexts – 25% and 75% - show the ramping activity, while the certain context doesn’t... because the ramping activity reflects task engagement/alertness/selective attention, rather than context per-se. But this is just speculation). Again, here, the task rule (context) should be maintained online somewhere, even early on (after the pressing of the context cue presentation), but this is not observed here.

We concur with the reviewer that this consideration remains speculative, but in order to highlight that ramping activity could also reflect other cognitive variables than context, we added

Page 26, Line 1-7:

“Moreover, it is conceivable that ramping dynamics could reflect other latent variables, such as engagement, alertness or selective attention. This possibility could ideally be tested using behavioral tasks that are designed to isolate the constructs, possibly combined with other physiologic readouts, such as pupil size, skin conductance, electrocardiogram or eye-tracking to quantify their contribution to ramping dynamics.”

Back to the original point... if there were context-specific signals in the brain, they should be present from the moment that the context is shown to the subjects. If the context signals only emerge seconds later, I would argue that these are not context signals, but rather something else that correlates with context (such as attention).

We thank the reviewer for this remark and fully concur that the current study does not report context-dependent signals that are maintained throughout. However, as outlined in paper and as an inherent limitation of intracranial EEG, we did not record from other brain regions that might represent context during the entire task duration. Hence, it is conceivable that context is only represented in the PFC when it is need to guide behavior. But we concur with the reviewer, it is likely that context is represented somewhere in the brain throughout.

Page 24, Line 19-22:

This consideration also possibly explains why we did not observe a continuous representation of the currently active rule in PFC. Future studies that also consider e.g., parietal or medial temporal areas might be able to observe such a sustained response.

Page 28, Line 20 – Page 29, Line 5

"However, it is worth noting that we cannot rule out some alternative hypotheses due to inherent limitations of human intracranial recordings and the currently employed experimental design. First, due to the inherent limited coverage in intracranial recordings, we cannot preclude that specialized sub-regions within the human motor cortex (e.g. anterior or posterior parts of the supplementary motor area, frontal eye fields, premotor cortex) might also encode contextual information. Moreover, we did not record from various other brain regions that might maintain context-dependent representation throughout, e.g., the parietal cortex or hippocampus."

Comment #20:

Our response in revision #3:

First, we do not understand why this depiction should raise any doubts as it is simply a different depiction of Figure 1c. The reviewer asks for an RT distribution for the different contexts. This is what we have depicted in Figure 1c and clearly described in the results section:

Yes, I have no issues with this design. The question I raised is because Response Figure 3 shows the RT distribution. Figure 1c shows the distribution of single subject averages, which is not the same. With the figures shown so far, I don't know how often subjects waited for the HLL (rather than predict its timing) across the different contexts.

We have provided the reaction time distributions in **Response Figure 1**.

Comment #21:

Our response in revision #3:

The reviewer's argument here is that the differences in population activity that we observe between the different context-conditions are driven by the presumed activity change upon the lower limit. [...] If it was true that the results are a simple consequence of activity changes upon the lower limit, one would have expected a temporal lag in the responses shown in Fig. 5c simply due to different RTs between context-conditions...

**Agreed. And it does seem that it is the case (at least with visual inspection):
The deviation from baseline for the green line (0%) deviates from baseline ~100ms after the other 2 conditions.**

We thank the reviewer for this comment and concurring with our interpretation.

However, I will refer here to my previous response highlighting that the concern does not rest on the existence of HLL-triggered signals.

We understand that this is central concern and refer to our detailed responses, esp. in comment #2 and #3. We would like to highlight again that activity change from pre- to

post-lower limit for the condition with no uncertainty speaks against this interpretation. However, as outlined above, we carefully consider this alternative interpretation and discuss it in detail in the discussion.

Comment #22:

Our response in revision #3:

We thank the reviewer for this remark and thank the reviewer for highlighting this inconsistency. Figure 1c shows RTs but it reflects the time between the self-paced trial start and the reaction. We changed Figure 1c to reflect the time between the lower limit and the reaction times.

This is still not consistent with the values described in the text (and the distribution shown in the Response Figure 3. This figure shows subject average values in the order of 80-100ms (depending on condition), but the text says (page 6, line 4) that the mean RT was 43.9ms. And then, response figure 3 shows an average of ~90ms.

This is a misunderstanding. The statement referred to a **difference** in reaction times between condition. We apologize for the confusion. The following sentence "*We found that reaction times (RT) gradually increased as a function of uncertainty (Fig. 1c; +43.9ms ± 19.8ms, mean ± SD [...])*" indicates that there is an increase in reaction time of +43.9ms from the condition with a 0% to the condition with a 75% likelihood of stopping. We fully agree that the sentence is confusing and leads to misinterpretation. Therefore, we now provide the mean ± SD for each condition.

Page 6, Line 6-11:

"We found that reaction times (RT) gradually increased as a function of uncertainty (Fig. 1c; 0% = 79.92 ± 19.66ms; 25% = 94.98 ± 23.86ms; 75% = 123.81 ± 29.34ms; mean ± SD; F(2,36) = 58.99, p < 0.001, $\eta_p^2 = 0.77$; one-way RM-ANOVA). Participants were also less accurate in trials with high uncertainty (Fig. 1c; 0% = 92.08 ± 6.99%; 25% = 81.1 ± 8.33%; 75% = 71.39 ± 5.46%; mean ± SD; F(2,36) = 81.53, p < 0.001, $\eta_p^2 = 0.82$)."

Comment #23:

Our response in revision #3:

(3) We then aimed to test whether there is also an 'hand-off' of action-specific information and therefore employed the cross-regional pattern analysis. This analysis revealed that that a discriminative pattern that was present in the action-specific subspace in PFC was equally present in the action-specific subspace in motor cortex, but with a temporal lag, suggesting that this information initially emerged in PFC and was passed onto motor cortex.

However, we realized that this interpretation is clearly subject to debate and hence nuanced our language and removed claims regarding the 'information hand-off'.

I would like to thank the authors for these changes, which now accurately reflect the conclusions of these analyses

We would like to thank the reviewer the positive assessment of our work.

References

- 1 Churchland, M. M., Cunningham, J. P., Kaufman, M. T., Ryu, S. I. & Shenoy, K. V. Cortical preparatory activity: representation of movement or first cog in a dynamical machine? *Neuron* **68**, 387-400, doi:10.1016/j.neuron.2010.09.015 (2010).
- 2 Kaufman, M. T. *et al.* The Largest Response Component in the Motor Cortex Reflects Movement Timing but Not Movement Type. *eNeuro* **3**, doi:10.1523/ENEURO.0085-16.2016 (2016).
- 3 Ter Wal, M. *et al.* Human stereoEEG recordings reveal network dynamics of decision-making in a rule-switching task. *Nat Commun* **11**, 3075, doi:10.1038/s41467-020-16854-w (2020).